# Assessing the quality of denoising diffusion models in Wasserstein distance: noisy score and optimal bounds

Vahan Arsenyan[*]    Elen Vardanyan[*]    Arnak S. Dalalyan
CREST, ENSAE, Institut Polytechnique de Paris
5 avenue Henry Le Chatelier
91764 Palaiseau, France

## Abstract

Generative modeling aims to produce new random examples from an unknown target distribution, given access to a finite collection of examples. Among the leading approaches, denoising diffusion probabilistic models (DDPMs) construct such examples by mapping a Brownian motion via a diffusion process driven by an estimated score function. In this work, we first provide empirical evidence that DDPMs are robust to constant-variance noise in the score evaluations. We then establish finite-sample guarantees in Wasserstein-2 distance that exhibit two key features: (i) they characterize and quantify the robustness of DDPMs to noisy score estimates, and (ii) they achieve faster convergence rates than previously known results. Furthermore, we observe that the obtained rates match those known in the Gaussian case, implying their optimality.

## 1    Introduction

We study the problem of generative modeling, which aims to construct a mechanism capable of producing synthetic samples that mimic a target distribution $P^*$, given access to independent observations from $P^*$. This fundamental task lies at the core of numerous applications, including image, text, music, and molecule generation. Among the recent advances in this domain, Denoising Diffusion Probabilistic Models (DDPMs), introduced in [HJA20], have emerged as a remarkably effective class of generative models; see, *e.g.*, [CMFW24, YZS+24, TZ25] for comprehensive overviews. In this work, we contribute to the growing theoretical understanding of DDPMs by analyzing several of their key properties and performance guarantees.

The central idea underlying DDPMs is to construct a transport map that transforms a simple source of randomness into a sample from the target distribution $P^*$. More precisely, for any distribution $P^*$, there exists a map defined via a stochastic differential equation (SDE) that takes as input a standard Gaussian vector $\boldsymbol{\xi}_0$ and a standard Brownian motion $\boldsymbol{W}$, and outputs a vector with distribution $P^*$. Importantly, only the drift term of the SDE depends on $P^*$, and this dependence occurs through the score function, that is, the gradient of the log-density of a Gaussian-smoothed version of $P^*$. This formulation reduces the generative modeling task to that of score estimation: one can estimate the score function from data and substitute this estimate into the SDE to approximately sample from $P^*$.

For many commonly used datasets, such as CIFAR-10 and CelebA-HQ considered in Section 6, accurate estimators of the score function are available. Generating a synthetic sample reduces to drawing a Gaussian vector together with the increments of a Brownian motion, and simulating the SDE defined by the pretrained score. This procedure requires multiple evaluations of the score estimator. The first question we address in this paper is: what happens if each evaluation returns a value corrupted by additive centered noise? Such a scenario may arise when the pretrained model is hosted on a remote server and communication introduces random perturbations, or when the score values are compressed using stochastic rounding. Anticipating our main findings, we emphasize that, perhaps counterintuitively, we observe that adding even a constant level of noise to each score evaluation has only a limited effect on the quality of the generated samples; see Figure 1 for an illustration.

---

[*]Equal Contribution

39th Conference on Neural Information Processing Systems (NeurIPS 2025).

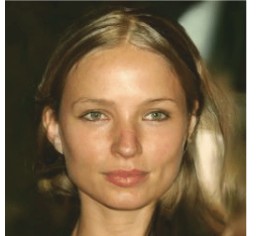 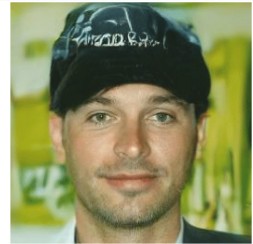 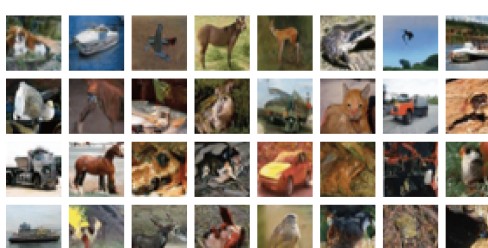

Figure 1: Generated images obtained by DDPM with a constant-level noise added to the estimated score. Left: CelebA-HQ. Right: CIFAR10. The result is visually as good as the noiseless one.

The second question we investigate concerns the accuracy of DDPMs when performance is measured in terms of the Wasserstein distance. A natural criterion in this setting is the number of score function queries $K$ required to achieve a prescribed level of accuracy $\varepsilon$. For the Gaussian target distribution, elementary computations show that $K = \mathcal{O}(\sqrt{D}/\varepsilon)$, where $D$ denotes the ambient dimension. Surprisingly, however, it remains unclear whether DDPMs maintain this level of accuracy for broader classes of distributions beyond the Gaussian case.

**Contributions.** The main contributions of this work can be summarized as follows:

- We provide empirical evidence, based on experiments with the CIFAR-10 and CelebA-HQ datasets, that DDPMs are remarkably robust to noise in the evaluation of the score function.
- We derive non-asymptotic upper bounds on the Wasserstein-2 distance between the target distribution and the distribution induced by the DDPM with noisy score evaluations, thus offering a theoretical explanation for the observed robustness.
- Our bounds match—up to a multiplicative constant—the rate $\sqrt{D}/\varepsilon$ of the case of a Gaussian target. Moreover, our results extend to a significantly broader class of distributions, including compactly supported semi-log-concave measures supported on low-dimensional subspaces.

**Related work** [KFL22] highlighted the connection between DDPMs and the Wasserstein distance. The first quantitative bounds—polynomial in the dimension and valid for a broad class of $P^*$—were established in [CCL$^+$23], covering several metrics. Unlike their result in total variation (TV) distance, their bound in Wasserstein distance has the poor scaling $D^5/\varepsilon^{12}$. Subsequent work significantly improved this rate: [CLL23] achieved $D^4/\varepsilon^2$ under minimal assumptions, while [BZL$^+$23, GNZ25, YY25, SOB$^+$25] reduced it further to $D/\varepsilon^2$, assuming stronger conditions on $P^*$. [SO25] proved the $\sqrt{D}/\varepsilon^2$ rate and our paper closes the loop by proving that the optimal rate $\sqrt{D}/\varepsilon$ is achieved by the standard DDPM procedure. A related result by [GZ24] establishes similar bounds for the probability flow ODE, but under more restrictive assumptions, such as strong log-concavity of $P^*$.

Over the past three years, substantial progress has also been made in establishing guarantees for DDPMs in total variation and Kullback–Leibler divergence under weak assumptions on $P^*$ [CDS25, LJLS25, LY25, BBDD24, LHE$^+$24], including acceleration techniques such as parallel sampling, randomized midpoint, and Runge–Kutta methods [CRYR24, GCC24, WCW24]. In parallel, a growing body of work investigates the statistical optimality of score-based models [OAS23, WWY24, HST25], as well as their ability to adapt to low-dimensional structure [Bor22, TY24, LY24, HWC24, ADR24, PAD24]. Analogous results for flow matching have been established in [KT25].

**Notation** For $D \in \mathbb{N}$, $\mathbf{I}_D$ is the $D \times D$ identity matrix. We use notation $\mathbf{A} \prec \mathbf{B}$, $\mathbf{A} \preccurlyeq \mathbf{B}$, $\mathbf{A} \succ \mathbf{B}$, $\mathbf{A} \succcurlyeq \mathbf{B}$ to design that the matrix $\mathbf{A} - \mathbf{B}$ is, respectively, negative definite, negative semi-definite, positive definite and positive semi-definite. We denote by $\mathcal{N}_D(\boldsymbol{\mu}, \boldsymbol{\Sigma})$ the $D$-dimensional Gaussian distribution with mean $\boldsymbol{\mu}$ and covariance matrix $\boldsymbol{\Sigma}$. Let $\gamma^D$ be the density function of $\mathcal{N}_D(0, \mathbf{I}_D)$. The norm of a vector is always understood as the Euclidean norm, whereas the norm of a matrix is the operator norm (the largest singular value). The independence of random vectors $\boldsymbol{X}$ and $\boldsymbol{Y}$ is denoted by $\boldsymbol{X} \perp\!\!\!\perp \boldsymbol{Y}$. The Wasserstein-$q$ distance between two distributions $P$ and $Q$ is defined by

$$\mathsf{W}_q^q(P, Q) = \inf_{\varrho \in \Gamma(P,Q)} \mathbf{E}_{(\boldsymbol{X},\boldsymbol{Y}) \sim \varrho}[\|\boldsymbol{X} - \boldsymbol{Y}\|^q],$$

where $q \geqslant 1$ and $\Gamma(P, Q)$ is the set of all joint distributions with marginals $P$ and $Q$. For any function $g : [0, T] \times \mathbb{R}^D \to \mathbb{R}$, we will write $\nabla g$ and $\nabla^2 g$ for the gradient and the Hessian of $g$ with respect to its second variable. If $g : [0, T] \times \mathbb{R}^D \to \mathbb{R}^D$, we write $\mathrm{D}g$ for the differential of $g$ with respect to its second variable. For each random vector $\boldsymbol{X}$, we write $\|\boldsymbol{X}\|_{\mathbb{L}_2} = (\mathbf{E}[\|\boldsymbol{X}\|_2^2])^{1/2}$.

## 2 Problem statement and conditions

The goal of this section is to set the framework of denoising diffusion probabilistic models with randomized score estimators and to state the conditions imposed on the unknown target distribution.

**The setting of randomized score estimators** Our setting is a bit more general than those previously studied in the literature. For an unknown distribution $P^*$ on $\mathbb{R}^D$, and for $t > 0$, we define $P_t^*$ as the distribution of $\alpha_t \boldsymbol{X} + \beta_t \boldsymbol{\xi}$, where $(\boldsymbol{X}, \boldsymbol{\xi}) \sim P^* \otimes \gamma^D$, $\alpha_t = e^{-t}$, and $\beta_t = \sqrt{1 - \alpha_t^2}$. The set $(P_t^*)_{t \geqslant 0}$ can be seen as a curve in the space of probability measures interpolating between $P^*$ and $\gamma^D$, since $P_0^* = P^*$ and $P_\infty^* = \gamma^D$. For $t > 0$, $P_t^*$ is absolutely continuous with respect to the Lebesgue measure $\lambda^D$ on $\mathbb{R}^D$ with an infinitely differentiable density. Therefore, we can define the score function $\boldsymbol{s}$ by

$$\pi(t, \boldsymbol{x}) = \frac{\mathrm{d}P_t^*}{\mathrm{d}\lambda^D}(\boldsymbol{x}), \qquad \boldsymbol{s}(t, \boldsymbol{x}) = \nabla \log \pi(t, \boldsymbol{x}). \tag{1}$$

Since $P_t^*$ is unknown, we cannot access $\boldsymbol{s}(t, \boldsymbol{x})$. Instead, we have access to randomized and noisy evaluations of this function: for each query $(t, \boldsymbol{x}) \in [0, \infty) \times \mathbb{R}^D$, we can observe a random vector $\widetilde{\boldsymbol{s}}(t, \boldsymbol{x})$ such that $\|\widetilde{\boldsymbol{s}}(t, \boldsymbol{x}) - \boldsymbol{s}(t, \boldsymbol{x})\|_{\mathbb{L}_2}$ is small. Our goal is to combine independent Gaussian random vectors and queries to the approximate score $\widetilde{s}$ to build a random vector $\boldsymbol{Z}$ in $\mathbb{R}^D$ having a distribution $P_Z$ close to $P^*$. To this end, we focus on the DDPM algorithm presented in Algorithm 1.

---

**Algorithm 1** Generation of $\boldsymbol{Z}$ by the denoising diffusion probabilistic model

**Require:** Sequence $(t_1, \ldots, t_{K+1})$ for some integer $K \geqslant 1$
**Ensure:** Vector $\boldsymbol{Z} = \boldsymbol{Z}_{K+1}$
 1: Set $t_0 = 0$, $T = t_{K+1}$, and $\boldsymbol{Z}_0 \sim \gamma^D$
 2: **for** $k = 0$ **to** $K$ **do**
 3:     Set $h_k = t_{k+1} - t_k$
 4:     Generate $\boldsymbol{\xi}_{k+1} \sim \gamma^D$, independent of all previous randomness
 5:     Query $\tilde{s}$ at $(t_k, \boldsymbol{Z}_k)$
 6:     Set $\boldsymbol{Z}_{k+1} = (1 + h_k)\boldsymbol{Z}_k + 2h_k\tilde{s}(T - t_k, \boldsymbol{Z}_k) + \sqrt{2h_k}\,\boldsymbol{\xi}_{k+1}$
 7: **end for**
 8: **Output** $\boldsymbol{Z}_{K+1}$

---

We postpone the discussion of the origin of this algorithm to Section 3. The main difference between our setting and prior work lies in the randomness of $\widetilde{s}$, which goes beyond the randomness of the training sample. Let us provide concrete examples to illustrate our setting.

**Example 1 (Noisy score estimator).** Assume that an estimator $\widehat{s}$ is available. Due to issues such as communication constraints or privacy concerns, we do not observe $\widehat{s}(t, \boldsymbol{x})$ directly, but rather a noisy version $\widetilde{s}(t, \boldsymbol{x}) = \widehat{s}(t, \boldsymbol{x}) + \boldsymbol{\zeta}$, where $\boldsymbol{\zeta}$ is random, typically with zero mean and bounded variance.

**Example 2 (Compressed score estimator).** Assume again that an estimator $\widehat{s}$ is available, but only one of its coordinates can be queried at a time. At each iteration, we randomly choose $i \in \{1, \ldots, D\}$ uniformly and set $\widetilde{s}(t, \boldsymbol{x}) = D \times (\widehat{s}(t, \boldsymbol{x})^\mathsf{T} \boldsymbol{e}_i)\boldsymbol{e}_i$, where $\boldsymbol{e}_i$ is the $i$-th canonical basis vector.

**Example 3 (Randomized network weights).** The conventional approach fits the weights $\boldsymbol{\theta}$ of a neural net $\boldsymbol{\phi}(t, \boldsymbol{x}; \boldsymbol{\theta})$ to the unknown score $\boldsymbol{s}(t, \boldsymbol{x})$ by minimizing the (estimated) prediction error:

$$\min_{\boldsymbol{\theta} \in \mathbb{R}^p} R(P^*, \boldsymbol{\theta}), \quad \text{where} \quad R(P^*, \boldsymbol{\theta}) := \int_0^T \int_{\mathbb{R}^D} \|\boldsymbol{\phi}(t, \boldsymbol{x}, \boldsymbol{\theta}) - \boldsymbol{s}(t, \boldsymbol{x})\|^2 \pi(t, \boldsymbol{x}) \, \mathrm{d}\boldsymbol{x} \, \mathrm{d}t.$$

One can instead minimize an estimator of the integrated error under a Gaussian prior by solving

$$\widehat{\boldsymbol{\mu}} \in \arg\min_{\boldsymbol{\mu} \in \mathbb{R}^p} \int_{\mathbb{R}^p} R(P^*, \boldsymbol{\mu} + \sigma\boldsymbol{z}) \, \gamma^p(\boldsymbol{z}) \, \mathrm{d}\boldsymbol{z},$$

where $\sigma > 0$ is a hyperparameter. This may lead to a more robust score estimator. In this setting, the randomized estimator of the score at each query point $(t, \boldsymbol{x})$ is $\boldsymbol{\phi}(t, \boldsymbol{x}, \widehat{\boldsymbol{\mu}} + \sigma\boldsymbol{\zeta})$, with $\boldsymbol{\zeta} \sim \gamma^p$ generated independently by the user.

**Conditions on the target distribution**  The guarantees on the precision of the DDPM that we will state in the next section depend on the properties of the target $P^*$. We will express these properties in terms of a function $\varphi$.

**Assumption 1.** For a function $\varphi : \mathbb{R}_{>0} \to \mathbb{R}_{>0}$, we say that $P^*$ or $X$ satisfies Assumption 1 with function $\varphi$ if, for $(X, \boldsymbol{\xi}) \sim P^* \otimes \gamma^D$, it holds that $\mathrm{Var}\left(X \mid X + \sigma\boldsymbol{\xi} = y\right) \preccurlyeq \varphi(\sigma)\,\mathbf{I}_D$ for all $\sigma > 0$.

Many distributions satisfy this assumption (see Appendix A for the proofs):
  (a) If $X$ has compact support $\mathcal{K}$ with $\mathrm{diam}(\mathcal{K}) = 2\mathfrak{D}_X$, Assumption 1 holds with $\varphi(\sigma) \equiv \mathfrak{D}_X^2$;
  (b) Any $m$-strongly log-concave distribution $P^*$ satisfies Assumption 1 with $\varphi(\sigma) = \frac{\sigma^2}{1+m\sigma^2}$;
  (c) If $X$ is semi-log-concave with constant[2] $M \geqslant 0$ and has compact support of diameter $2\mathfrak{D}_X$, then $X$ satisfies Assumption 1 with $\varphi(\sigma) = \mathfrak{D}_X^2 \wedge \frac{\sigma^2}{(1-M\sigma^2)_+}$;
  (d) If $X$ satisfies Assumption 1 with some function $\varphi$, $\mathbf{U}$ is a $D \times D$ orthonormal matrix and $\boldsymbol{b} \in \mathbb{R}^D$, then $\mathbf{U}X + \boldsymbol{b}$ satisfies Assumption 1 with the same $\varphi$;
  (e) If $X$ is obtained by concatenating two independent vectors $X_1$ and $X_2$ satisfying Assumption 1 with the same function $\varphi$, then $X$ satisfies Assumption 1 with $\varphi$.
  (f) If $(W, \boldsymbol{\zeta}) \sim P_0 \otimes \gamma^D$ such that $W$ satisfies Assumption 1 with the function $\varphi_0$, then, $X = W + \tau\boldsymbol{\zeta}$ satisfies Assumption 1 with the function $\varphi_\tau(\sigma) = \frac{\tau^2\sigma^2}{\tau^2+\sigma^2} + \frac{\sigma^4\varphi_0(\sqrt{\tau^2+\sigma^2})}{(\tau^2+\sigma^2)^2}$.
  (g) If $W$ is supported by a compact set of diameter $2\mathfrak{D}$ and $\boldsymbol{\zeta} \perp\!\!\!\perp W$ is $m$-strongly log-concave with an $M$-Lipschitz score function, then $X = W + \boldsymbol{\zeta}$ satisfies Assumption 1 with $\varphi(\sigma) = \frac{\sigma^2}{1+m\sigma^2} + \frac{(M\mathfrak{D}\sigma^2)^2}{(1+M\sigma^2)^2}$.

The main purpose of Assumption 1 is to ensure that the drift coefficient of the backward diffusion process is strongly convex when the noise level is large and semi-log-concave for all noise levels. Moreover, the drift coefficient is always gradient-Lipschitz, with a Lipschitz constant depending on the noise level. These properties are summarized in the following result[3].

**Proposition 1.** *Let $X$ and $\boldsymbol{\xi}$ be random vectors in $\mathbb{R}^D$ drawn from $P^* \otimes \gamma^D$. For any $\alpha, \beta > 0$, the density $\pi_Y$ of $Y = \alpha X + \beta\boldsymbol{\xi}$ is twice continuously differentiable and satisfies*

$$\nabla^2 \log \pi_Y(y) = \frac{\alpha^2}{\beta^4}\mathrm{Var}(X \mid Y = y) - \frac{1}{\beta^2}\mathbf{I}_D \succcurlyeq -\frac{1}{\beta^2}\mathbf{I}_D, \qquad \text{for all } y \in \mathbb{R}^D.$$

*Thus, Assumption 1 is equivalent to $\nabla^2 \log \pi_Y(y) \preccurlyeq \frac{(\alpha^2\varphi(\beta/\alpha)-\beta^2)}{\beta^4}\mathbf{I}_D$, for all $y \in \mathbb{R}^D$, $\alpha, \beta > 0$.*

The last inequality above implies that if $\varphi(\beta/\alpha) \leqslant (\beta/\alpha)^2$, the distribution of $Y = \alpha X + \beta\boldsymbol{\xi}$ is log-concave, and it is strongly log-concave if the inequality is strict.

**Conditions on the estimated score**  As mentioned in Section 2, we consider randomized estimators $\widetilde{s}$ of the true score function $s$. The mean squared error of such an estimator can be decomposed into a bias and a variance term:

$$\mathbf{E}\big[\,\|\widetilde{s}(t, x) - s(t, x)\|^2\,\big] = \|\mathbf{E}\left[\widetilde{s}(t, x)\right] - s(t, x)\|^2 + \mathbf{E}\big[\,\|\widetilde{s}(t, x) - \mathbf{E}\left[\widetilde{s}(t, x)\right]\|^2\,\big].$$

In what follows, we analyze separately the impact of the bias and the variance on the overall error. As we will see, the variance term has a much weaker influence on the final accuracy than the bias term. To reflect this difference, we introduce the following assumption.

**Assumption 2.** There are constants $\varepsilon^b_{\mathsf{score}}$ and $\varepsilon^v_{\mathsf{score}}$ such that for all $t \in \{t_k : k \leqslant K\}$ of Algorithm 1,

$$\sup_{x \in \mathbb{R}^D} \|\mathbf{E}\left[\widetilde{s}(t, x)\right] - s(t, x)\| \leqslant D^{1/2}\varepsilon^b_{\mathsf{score}}, \qquad \sup_{x \in \mathbb{R}^D} \|\widetilde{s}(t, x) - \mathbf{E}\left[\widetilde{s}(t, x)\right]\|_{\mathbb{L}^2} \leqslant D^{1/2}\varepsilon^v_{\mathsf{score}}.$$

Assumption 2 imposes uniformity over all $x \in \mathbb{R}^D$ and $t \in t_k : k \leqslant K$ and, therefore, is a stronger condition than the one used in previous work [CLL23]. The latter considers $\mathbb{L}_2$-norm with respect to $P^*_t$, rather than a supremum, and involves a weighted average over $t$. While it may be possible to relax the requirement involving the maximum over the time grid, the uniformity with respect to $x$ appears to be more difficult to replace by the $\mathbb{L}_2$-norm wrt $P^*_t$. It is important to note, however, that for our proof needs only an $\mathbb{L}_2$ bound with respect to the distribution of the DDPM output at time $t$.

---

[2]We recall that $X$ is semi-log-concave [Cla83] with constant $M \in \mathbb{R}$ if $X$ has a density $\pi_X$ wrt the Lebesgue measure and $-\log \pi_X(x) + \frac{M}{2}\|x\|^2$ is convex; see [VCK25] for an application in sampling.

[3]The formula relating the Hessian of the log-density to the conditional variance, stated in Proposition 1 is often referred to as the second-order Tweedie formula.

# 3 Score-Based Generative Modeling: preliminary considerations

The starting point of a DDPM is the forward process given as a solution to a stochastic differential equation (SDE). The simplest and the most widespread choice is the Ornstein–Uhlenbeck process

$$\mathrm{d}\boldsymbol{X}_t = -\boldsymbol{X}_t \,\mathrm{d}t + \sqrt{2}\,\mathrm{d}\boldsymbol{B}_t, \qquad t \geqslant 0, \quad \boldsymbol{X}_0 \sim P^*, \qquad (\boldsymbol{B}_t)_{t\geqslant 0} \perp\!\!\!\perp \boldsymbol{X}_0, \qquad (2)$$

where $(\boldsymbol{B}_t)_{t\geqslant 0}$ is a standard Brownian motion in $\mathbb{R}^D$. The Ornstein–Uhlenbeck process is a time-homogeneous Markov process which is also a Gaussian process, with stationary distribution equal to the standard Gaussian distribution $\gamma^D$ on $\mathbb{R}^D$. The forward process has the interpretation of transforming samples from the data generating distribution $P^*$ into the latent distribution. From the classical theory of Markov diffusions, it is known that $P_t^* := \mathrm{law}(\boldsymbol{X}_t)$ converges to $\gamma^D$ exponentially fast in various divergences and metrics such as the 2-Wasserstein metric $\mathsf{W}_2$: $\mathsf{W}_2(P_t^*;\gamma^D) \leqslant e^{-t}\mathsf{W}_2(P_0;\gamma^D)$, see for instance [Vil08].

## 3.1 Reverse Process: continuous-time and time-discretized versions

If we reverse the forward process in time, we obtain a process that transforms the latent distribution into the target distribution $P^*$, which is the aim of generative modeling. Fix some large time horizon $T > 0$ and set $\boldsymbol{Y}_t := \boldsymbol{X}_{T-t}$, then $\mathrm{law}(\boldsymbol{Y}_0) = \mathrm{law}(\boldsymbol{X}_T)$ is close to the Gaussian distribution $\gamma^D$. Notably, the dynamics of the reverse process can also be described by a stochastic differential equation, as stated in the next result.

**Theorem 1** ([And82]). *If $(\boldsymbol{X}_t)_{t\geqslant 0}$ is a solution to (2) and $\boldsymbol{Y}_t = \boldsymbol{X}_{T-t}$, then there exists a Brownian Motion $(\widetilde{\boldsymbol{B}}_t)_{t\geqslant 0} \perp\!\!\!\perp \boldsymbol{Y}_0$ such that*

$$\mathrm{d}\boldsymbol{Y}_t = (\boldsymbol{Y}_t + 2\nabla \log \pi(T - t, \boldsymbol{Y}_t))\,\mathrm{d}t + \sqrt{2}\,\mathrm{d}\widetilde{\boldsymbol{B}}_t, \quad 0 \leqslant t \leqslant T, \qquad (3)$$

*where $\pi(t, \boldsymbol{x}) \propto \int_{\mathbb{R}^D} \gamma^D\big((\boldsymbol{x} - \alpha_t \boldsymbol{y})/\beta_t\big) P^*(\mathrm{d}\boldsymbol{y})$, $\alpha_t = e^{-t}$ and $\beta_t^2 = 1 - e^{-2t}$.*

Note that $\pi(t, \boldsymbol{x})$ in this theorem coincides with the one defined in (1) and $\nabla \log \pi(T - t, \boldsymbol{Y}_t)$ is the score function $\boldsymbol{s}$ evaluated at scale $T - t$ and state $\boldsymbol{Y}_t$.

The forward process transforms a data point $\boldsymbol{X}_0$ drawn from $P^*$ into a point which is very close to being drawn from the latent distribution. The reverse process aims to transform a point $\boldsymbol{Y}_0$ drawn from the latent distribution into a point drawn from $P^*$. To this end, we replace the unknown score function by its estimate $\widetilde{\boldsymbol{s}}$ based on a training sample $\boldsymbol{X}_1, \ldots, \boldsymbol{X}_n \sim P^*$. The resulting process is defined as the solution to the SDE

$$\mathrm{d}\widetilde{\boldsymbol{Y}}_t = (\widetilde{\boldsymbol{Y}}_t + 2\widetilde{\boldsymbol{s}}(T - t, \widetilde{\boldsymbol{Y}}_t))\,\mathrm{d}t + \sqrt{2}\,\mathrm{d}\widetilde{\boldsymbol{B}}_t, \qquad \widetilde{\boldsymbol{Y}}_0 \sim \gamma^D, \qquad t \in [0, T]. \qquad (4)$$

Both $\widetilde{\boldsymbol{Y}}$ and $\boldsymbol{Y}$ are processes on the space $\mathbb{C}([0, T], \mathbb{R}^D)$, differing in their initial conditions and drift terms. We wish to assess the distance between the distributions of their states at time $T$.

To efficiently sample the final state of the reverse process, we have to discretize SDE (4). To this end, we introduce a sequence $(h_k)_{k\in\mathbb{N}}$ of positive numbers and set[4] $t_k = h_0 + \ldots + h_{k-1}$. We then define

$$\boldsymbol{Z}_{k+1} = (1 + h_k)\boldsymbol{Z}_k + 2h_k\widetilde{\boldsymbol{s}}(T - t_k, \boldsymbol{Z}_k) + \sqrt{2h_k}\,\boldsymbol{\xi}_{k+1}; \qquad \boldsymbol{Z}_0 \sim \gamma^D, \qquad (5)$$

where $(\boldsymbol{\xi}_k)_{k\in\mathbb{N}}$ is a sequence of independent standard Gaussian random variables. The rationale behind this definition is that $\boldsymbol{Z}_k$ has approximately the same law as $\widetilde{\boldsymbol{Y}}_{t_k}$, for every $k$.

**Definition 1.** The denoising diffusion probabilistic model is the distribution $P^{\mathsf{DDPM}}$ of the random vector $\boldsymbol{Z}_{K+1}$ defined by (5). It requires the choice of $K \in \mathbb{N}$, the sequence $(t_1, \ldots, t_{K+1})$ and the score estimators $\big(\widetilde{\boldsymbol{s}}(T - t_k, \cdot)\big)_{k=0,\ldots,K}$.

In this paper, we are interested in quantifying the accuracy of the denoising diffusion generative model when the error is measured in terms of the Wasserstein distance, that is to upper bound $\mathsf{W}_2(P^*, P^{\mathsf{DDPM}})$. In the rest of this section, we motivate the choice of the Wasserstein distance and discuss the challenges related to it in the framework of denoising diffusions.

---

[4]By convention, $t_0 = 0$.

## 3.2 Relevance of the Wasserstein distance

Recent work on assessing denoising diffusion models mainly focuses on accuracy measured by the total variation distance and the Kullback-Leibler divergence. However, we believe that for statistical purposes, measuring the quality of a generative model in the Wasserstein distance is highly appealing.

To justify this point of view, remind that the closeness of two distributions in TV-distance or KL-divergence does not guarantee the closeness of their means or their covariance matrices. In sharp contrast, the Wasserstein-2 distance offers such a guarantee, since it holds that

$$\|\mathbf{E}_P[\boldsymbol{X}] - \mathbf{E}_Q[\boldsymbol{X}]\| \leqslant \mathsf{W}_2(P, Q); \quad |(\mathbf{E}_P[\boldsymbol{X}^{\mathsf{T}}\mathbf{A}\boldsymbol{X}])^{1/2} - (\mathbf{E}_Q[\boldsymbol{X}^{\mathsf{T}}\mathbf{A}\boldsymbol{X}])^{1/2}| \leqslant \mathsf{W}_2(P, Q),$$

for any matrix $\mathbf{A}$ satisfying $0 \preccurlyeq \mathbf{A} \preccurlyeq \mathbf{I}$. The fact that the TV-distance and the KL-divergence are not suitable for controlling the moments of distributions can be demonstrated by the following example. Let $P$ be the exponential distribution with parameter 1 and, for every $n \in \mathbb{N}$, set $P_n = (1 - \delta_n)P + \delta_n Q_n$, where $\delta_n = 1/\sqrt{n}$ and $Q_n$ is the uniform distribution on $[n, n+2]$. One can easily check that $P_n$ is very close to $P$ both in the TV-distance and in the KL-divergence:

$$\mathsf{d}_{\mathsf{TV}}(P_n; P) \leqslant \delta_n = n^{-1/2}; \qquad \mathsf{d}_{\mathsf{KL}}(P\|P_n) = -\log(1 - \delta_n) \leqslant 2n^{-1/2}, \quad n \geqslant 2.$$

Therefore, one could expect that $P_n$ is an excellent generative model for the target $P$. However, the generated examples will have a mean and variance that explode as $n \to \infty$, and will be infinitely far away from the mean and the variance of the target, since $\mathbf{E}_{P_n}[X] = 1 + n\delta_n \geqslant n^{1/2}$ and $\mathbf{E}_{P_n}[X^2] \geqslant 2(1 - \delta_n) + \delta_n n^2 \geqslant n^{3/2}$.

## 3.3 Challenges inherent to Wasserstein distance

When the distance $\mathsf{W}_q$ is employed to assess the quality of a DDPM, a mathematical challenge arises in quantifying the error due to the absence of the data-processing inequality for $\mathsf{W}_q$-distance. Let us clarify this point. Consider a forward mechanism $\mathscr{M}_{\rightarrow}$ that transforms the target $P^*$ into a distribution $P_1^*$ which is close to an easy-to-sample-from latent distribution $Q_0$: $P_1^* := \mathscr{M}_{\rightarrow}(P^*) \approx Q_0$. Furthermore, assume we have knowledge of the "inverse" forward mechanism, termed backward mechanism, which maps $P_1^*$ back to $P^*$: $\mathscr{M}_{\leftarrow}(P_1^*) = P^*$. The forward-backward methods of generative modeling then define the generative model as $Q_1 = \bar{\mathscr{M}}_{\leftarrow}(Q_0)$, where $\bar{\mathscr{M}}_{\leftarrow}$ represents a suitably regularized estimator of $\mathscr{M}_{\leftarrow}$. In DDPM, $\mathscr{M}_{\leftarrow}$ and $\bar{\mathscr{M}}_{\leftarrow}$ are specified by Markov kernels.

In this context, denoting $\mathsf{d}_F$ as the $F$-divergence for some $F$, the following relationship holds:

$$\mathsf{d}_F(Q_1\|P^*) = \mathsf{d}_F\big(\bar{\mathscr{M}}_{\leftarrow}(Q_0)\big\|P^*\big) \approx \mathsf{d}_F\big(\mathscr{M}_{\leftarrow}(Q_0)\big\|P^*\big)$$

$$= \mathsf{d}_F\big(\mathscr{M}_{\leftarrow}(Q_0)\big\|\mathscr{M}_{\leftarrow}(P_1^*)\big) \overset{\text{DPI}}{\leqslant} \mathsf{d}_F(Q_0\|P_1^*),$$

where the final equality derives from the data-processing inequality. Thus, the error of the generative distribution is dominated by how well the forward mechanism approximates the latent distribution, provided that the error of $\mathscr{M}_{\leftarrow}$ approximation is suitably small. These arguments were central in prior work[5] establishing bounds on the error of denoising diffusion models measured in TV-distance and KL-divergence. However, this approach breaks down for the Wasserstein distance $\mathsf{W}_q$, for which no suitable equivalent of the data processing inequality exists.

In the case of denoising diffusion models, the qualitative difference between the Wasserstein distance and $F$-divergences (such as TV-distances and KL-divergence) can be formally demonstrated even when the backward kernel is known. This is illustrated in the following lemma.

**Lemma 1.** *For any $T > 0$, let $Q_1^{T,\boldsymbol{s}}$ be the distribution of the backward process* (4) *at time $T$ with $\widetilde{\boldsymbol{s}}$ replaced by the true score $\boldsymbol{s}$. Let $\mathcal{N}$ be the set of all the Gaussian distributions. It then holds that*

$$\sup_{P^* \in \mathcal{N}} \frac{\mathsf{d}_{\mathsf{TV}}^2(Q_1^{T,\boldsymbol{s}}; P^*)}{\mathsf{d}_{\mathsf{TV}}^2(P^*; \gamma^D)} \bigvee \frac{\mathsf{d}_{\mathsf{KL}}(Q_1^{T,\boldsymbol{s}}\|P^*)}{\mathsf{d}_{\mathsf{KL}}(P^*\|\gamma^D)} \leqslant e^{-2T}; \qquad \sup_{P^* \in \mathcal{N}} \frac{\mathsf{W}_2(Q_1^{T,\boldsymbol{s}}; P^*)}{\mathsf{W}_2(P^*; \gamma^D)} = 1.$$

This lemma reveals that when assessing accuracy through the rate of improvement in Wasserstein distance, the choice of parameter $T$ must be carefully tailored to the target distribution $P^*$. This might be less important in the case of the TV-distance and the KL-divergence.

---

[5]See [CCL⁺23, BBDD24, HWC24, CDS25] and the references therein

# 4 Main results: bounds on the error in various settings

In this section, we upper bound the Wasserstein-2 distance between DDPM (see Algorithm 1) and the target $P^*$. Similar to [CLL23, BBDD24], we employ a discretization scheme composed of two regimes: an arithmetic grid in the first half and a geometric grid in the second half; see Algorithm 2.

---

**Algorithm 2** Definition of the discretization time steps

**Require:** $\delta, a, T_1 > 0$, and $K_0 \in \mathbb{N}_{>1}$
**Ensure:** Sequence $t_0 < t_1 < \ldots < t_{K+1}$
1: Set $t_0 = 0$, $K = 2K_0$, $t_{K+1} = T_1 + \frac{1}{2}\log(6a)$
2: **for** $k = 1$ **to** $K_0$ **do**
3:    Set $t_k = (T_1/K_0)\,k$                          {arithmetic grid}
4:    Set $t_{K_0+k} = T_1 + \frac{\log(6a)}{2}\big[1 - \big(\frac{2\delta}{\log(6a)}\big)^{k/K_0}\big]$.       {geometric grid}
5: **end for**
6: **Output** $(t_0, \ldots, t_{K+1})$

---

## 4.1 Strongly log-concave distributions convolved with a distribution with compact support

In this section, we consider the case of a distribution $P^*$ satisfying Assumption 1 with a function $\varphi$ that has the following form: for some constants $m, M, b \geqslant 0$,

$$\varphi(\sigma) = \frac{\sigma^2}{1 + m\sigma^2} + \frac{bM^2\sigma^4}{(1 + M\sigma^2)^2}, \qquad \forall \sigma > 0. \tag{6}$$

If $P^*$ is $m$-strongly log-concave, as discussed in Section 2, then (6) holds with $b = 0$ and any $M > 0$. Another class of distributions satisfying (6) consists of convolutions $P^* = P_{\mathsf{slc}} \star P_{\mathsf{cmpct}}$, where $P_{\mathsf{slc}}$ is $m$-strongly log-concave with an $M$-Lipschitz score, and $P_{\mathsf{cmpct}}$ is supported on a compact set of diameter $2\mathfrak{D}$, for some $M \geqslant m > 0$ and $\mathfrak{D} \geqslant 0$. In this case, (6) holds with $b = \mathfrak{D}^2$.

Finally, there are distributions satisfying Assumption 1 with $\varphi$ given by (6) that are not absolutely continuous with respect to the Lebesgue measure on $\mathbb{R}^D$. For example, if $P^*$ is supported on a linear subspace $\mathcal{S}$ of $\mathbb{R}^D$, and its restriction to $\mathcal{S}$, viewed as a distribution on $\mathbb{R}^d$ for some $d \in 1, \ldots, D$, satisfies Assumption 1 with $\varphi$ given by (6), then $P^*$ also satisfies the assumption with the same $\varphi$. This is a consequence of properties (d) and (e) presented in Section 2.

**Theorem 2.** *Let the target distribution $P^*$ satisfy $\mathbf{E}[\|\boldsymbol{X}\|_2^2] \leqslant D$ and Assumption 1 with function $\varphi$ given by (6) for some $m, M, b \geqslant 0$. Let us choose $T_1 > 0$,*

$$a = \tfrac{1}{m} + b, \qquad K_0 \geqslant 7T_1\log(6a) + 4\log(6a)\log\log(6a) \qquad \delta = 0.5e^{-2T_1},$$

*and define the sequence $(t_k)_{0 \leqslant k \leqslant K+1}$ by Algorithm 2. Let $\widetilde{\boldsymbol{s}}$ be a randomized estimator of the score satisfying Assumption 2. Then, the distribution $P^{\mathsf{DDPM}}$ of the output of Algorithm 1 based on $2K_0$ queries to the score estimator $\widetilde{\boldsymbol{s}}$ satisfies*

$$\mathsf{W}_2(P^*, P^{\mathsf{DDPM}}) \leqslant e^{(4/3)bM}\Big\{2e^{-T_1} + 7\sqrt{6a}\,h_{\max} + 4\sqrt{6a}\big(2\varepsilon_{\mathsf{score}}^b + h_{\max}^{1/2}\,\varepsilon_{\mathsf{score}}^v\big)\Big\}\sqrt{D}, \tag{7}$$

*with $h_{\max} = \max_k(t_{k+1} - t_k) \leqslant \frac{\log(6a)(\log\log(6a) + 2T_1)}{K_0}$.*

There are several notable features in the upper bound stated in Theorem 2, when we compare it to the previously known results.

**Remark 1** (Optimality)**.** The dependence of the discretization error (the second term in (7)) on the step size $h_{\max}$ is linear, whereas it was of order $h_{\max}^{1/12}$ in [CCL$^+$23, Cor. 6], $h_{\max}^{1/4}$ in [CLL23, Cor. 2.4], and $h_{\max}^{1/2}$ in [BZL$^+$23, Remark 12], [SOB$^+$25, Cor. 4.3], [SO25, GNZ25, YY25]. Moreover, [GNZ25] establishes that the lower bound on the Wasserstein-2 error, achieved by the Gaussian distribution, scales as $\sqrt{D}\,h_{\max}$, thereby implying the optimality of the bound in Theorem 2.

**Remark 2** (Conditions)**.** Assumptions on $P^*$ in Theorem 2 are less stringent than those in earlier works [BZL$^+$23, YY25, GNZ25]. In particular, for $m$-strongly log-concave $P^*$, we do not assume that the Hessian of the log-density is bounded from below. Furthermore, Theorem 2 covers the class

of distributions obtained as convolutions of a compactly supported distribution and a Gaussian, a framework not addressed in previous studies achieving a discretization error of $h_{\max}^{1/2}$. However, our conditions may be regarded as stronger than those of [CLL23, Cor. 2.4] providing the discretization error of order $h_{\max}^{1/4}$. These stronger assumptions are typically necessary for attaining faster rates of convergence. In conclusion, our conditions are weaker than those previously associated with the $h_{\max}^{1/2}$ rate, while enabling the faster convergence rate of $h_{\max}$.

**Remark 3** (Impact of noise). All previously known bounds are proportional to $\|(\widetilde{s} - s)(\tau, X)\|_{\mathbb{L}_2}$, where the proportionality factor is often logarithmic in the number of queries, and the $\mathbb{L}_2$-norm can take different forms—the weakest being the case where $\tau \sim \mathrm{Unif}([0, T])$ and the law of $X$ given $\tau = t$ is $P_t^*$. If $\widetilde{s}(t, x) = \widehat{s}(t, x) + \zeta$, with $\|\zeta\|_{\mathbb{L}_2}^2 = \sigma_\zeta^2 D$ as in Example 1 of Section 2, then $\|\widetilde{s} - s\|_{\mathbb{L}_2}^2 \geqslant \sigma_\zeta^2 D$. Thus, all known bounds include a term of constant order, independent of the number of queries. In contrast, the corresponding term in the bound of Theorem 2 is $O(\sqrt{D\,h_{\max}}\,\varepsilon_{\mathsf{score}}^v)$, which scales as $\sigma_\zeta \sqrt{DT_1/K}$ and thus vanishes as $K$, the number of queries, grows large.

**Remark 4** (Informal statement). To facilitate comparison with existing results, let us consider the strongly log-concave case $b = 0$ and denote by $L := a$ the surrogate of the Lipschitz norm of the score of $P^*$. For $T_1 = \log(K_0)$, our result implies that, after $K$ queries to the score estimator,

$$\mathsf{W}_2(P^*, P^{\mathsf{DDPM}}) \lesssim \sqrt{LD}\Big\{ \frac{\log L \log K}{K} + \varepsilon_{\mathsf{score}}^b + \frac{\sqrt{\log L \log K}}{\sqrt{K}}\,\varepsilon_{\mathsf{score}}^v \Big\}.$$

In particular, $\mathsf{W}_2(P^*, P^{\mathsf{DDPM}}) \lesssim \sqrt{LD}\,\varepsilon_{\mathsf{score}}^b$, provided that the number of queries satisfies

$$\frac{K}{\log K} \geqslant \Big\{ \frac{1}{\varepsilon_{\mathsf{score}}^b} \bigvee \Big(\frac{\varepsilon_{\mathsf{score}}^v}{\varepsilon_{\mathsf{score}}^b}\Big)^2 \Big\}\,\log L.$$

As mentioned in Remark 3, this improves on [BZL$^+$23, YY25, GNZ25, SO25], which require $K \gtrsim (\log L)/(\varepsilon_{\mathsf{score}}^b)^2$ and $\varepsilon_{\mathsf{score}}^v \lesssim \varepsilon_{\mathsf{score}}^b$ to achieve $\mathsf{W}_2(P^*, P^{\mathsf{DDPM}}) \lesssim \sqrt{LD}\,\varepsilon_{\mathsf{score}}^b$.

## 4.2 Semi log-concave distributions with compact support

In this section, we consider the case of a distribution $P^*$ satisfying Assumption 1 with a function $\varphi$ that has the following form: for some constants $b, M \geqslant 0$,

$$\varphi(\sigma) = b \wedge \frac{\sigma^2}{(1 - M\sigma^2)_+}, \qquad \forall \sigma > 0. \tag{8}$$

The typical example of $P^*$ satisfying this assumption is a distribution on a compact set $\mathcal{K}$ included in a linear subspace of $\mathbb{R}^D$, if in addition the log-density wrt to the Lebesgue measure on the subspace has a Hessian $\preccurlyeq M\mathbf{I}$. It then follows from claims (c), (d), and (e) of Section 2 that $P^*$ satisfies Assumption 1 with $\varphi$ as in (8) with $b = \mathfrak{D}_X^2$.

**Theorem 3.** *Let the target distribution $P^*$ satisfy $\mathbf{E}[\|X\|_2^2] \leqslant D$ and Assumption 1 with function $\varphi$ given by (8) for some $b, M \geqslant 0$. Let us choose $T_1 > 0$,*

$$a = b \vee 1, \qquad K_0 \geqslant 7T_1 \log(6a) + 4\log(6a)\log\log(6a) \qquad \delta = 0.5e^{-2T_1},$$

*and define the sequence $(t_k)_{0 \leqslant k \leqslant K+1}$ by Algorithm 2. Let $\widetilde{s}$ be a randomized estimator of the score satisfying Assumption 2. Then, the distribution $P^{\mathsf{DDPM}}$ of the output of Algorithm 1 based on $2K_0$ queries to the score estimator $\widetilde{s}$ satisfies*

$$\mathsf{W}_2(P^*, P^{\mathsf{DDPM}}) \leqslant e^{2bM+1}\Big\{ 2e^{-T_1} + 7\sqrt{6a}\,h_{\max} + 4\sqrt{6a}\big(2\varepsilon_{\mathsf{score}}^b + h_{\max}^{1/2}\varepsilon_{\mathsf{score}}^v\big)\Big\}\sqrt{D}, \tag{9}$$

*with $h_{\max} = \max_k(t_{k+1} - t_k) \leqslant \frac{\log(6a)(\log\log(6a) + 2T_1)}{K_0}$.*

Since the conclusions of this theorem closely mirror those of Theorem 2, the remarks provided after the latter apply here as well and will not be repeated. We merely emphasize two points. First, $P^*$ is not assumed to have a density wrt the Lebesgue measure on $\mathbb{R}^D$. Second, the number $K$ of queries to the score estimator required to achieve $\mathsf{W}_2$ error $\varepsilon$ scales as $1/\varepsilon$, up to a factor that grows at most logarithmically in $1/\varepsilon$. The exponential terms in (7) and (9) depend on the parameters of the target distribution. The independent work [SO25] employs a different proof technique yet exhibits a similar exponential dependence, suggesting that this behavior is intrinsic to bounding the Wasserstein distance in DDPMs. For a log-concave distribution supported on a compact domain, we have $(M, b) = (0, \mathfrak{D}_X^2)$, so the exponential factor in the bound (9) becomes a universal constant. This complements the result obtained in the strongly log-concave setting from Theorem 2.

# 5 Relation to prior work: extended discussion

Given the wealth of work on Langevin algorithms and score-based generative models, it would be infeasible to provide an exhaustive account of existing results. Instead, this section offers a selective overview of prior work, to situate our contributions within the broader landscape.

Theoretical guarantees for DDPMs have been inspired by techniques from the sampling literature, particularly those used for Langevin Monte Carlo and its variants; see the overview in [Che24]. Prior work can be grouped into three categories based on the underlying proof strategies.

The first category, represented by [CCL$^+$23, CLL23, BBDD24, CDS25, LY25, LJLS25, LHE$^+$24], includes works that build on the approach initiated in [DT12, Dal17b], combining the **Pinsker inequality with the Girsanov formula** to derive bounds in TV. Its key strengths are:

- it requires only a bound on the mean integrated squared error (MISE) of the score estimator—one of the weakest conditions in this framework;

- it relies on mild assumptions on the data-generating distribution $P^*$.

As noted in [CCL$^+$23, CLL23], TV-distance bounds can be converted into Wasserstein bounds under additional assumptions, such as compact support or light-tailed $P^*$. If the support lies in a ball, one can project the generated sample onto this ball and use that $W_2^2$ is bounded by the radius of the ball times the TV distance. By the data-processing inequality, this projection does not increase the TV-error.

However, this versatility comes at a price. Let $K_{\mathsf{TV}}(\widetilde{\varepsilon})$ be the number of steps required to achieve an error smaller than $\widetilde{\varepsilon}$ in TV-distance. Then, to achieve $W_2$-error $\varepsilon$, one needs a TV-error $\widetilde{\varepsilon} = \varepsilon^2/R^2$, leading to a number of steps at least $K_{\mathsf{TV}}(\varepsilon^2/R^2)$. As a result, the rates derived from this strategy are suboptimal: $O(D^4/\varepsilon^2)$ in [CLL23], $O(D/\varepsilon^4)$ in [BBDD24, CDS25], and $O(D^3/\varepsilon^2)$ in [LHE$^+$24], ignoring log-factors. Another limitation of this approach is that the resulting upper bound on the $W_2$ distance scales as the square root of the error of estimation of the score. Hence, to guarantee an error $\varepsilon$ in $W_2$, one needs the score estimation error $\varepsilon_{\mathsf{score}}$ of order $O(\varepsilon^2)$. Our results, as well as those of the third category below, typically require the weaker condition $\varepsilon_{\mathsf{score}} = O(\varepsilon)$.

The second category comprises results that exploit the interpretation of Langevin dynamics as a **gradient flow in the space of probability measures**. This perspective was initiated in [Wib18, Ber18] and further developed in [CB18, DMM19, VW19]. Interestingly, the first polynomial-in-dimension guarantees for DDPM fall within this framework, as shown in [LLT22, YW22]. These works evaluate the error in terms of $f$-divergences such as total variation, KL, or $\chi^2$ divergence. However, when translated to bounds in the $W_2$ distance, they suffer from the same limitations as the TV-based approaches discussed above. Moreover, this line of work typically relies on strong structural assumptions on the target distribution $P^*$, notably the satisfaction of a log-Sobolev inequality. Another limitation, shared with our own analysis, is that the score estimation error is measured in the uniform norm. We believe, however, that this requirement could be relaxed, both in the gradient-flow framework and in the recursive method developed in our work.

The third category comprises works using the **recursive approach** to bound the error of iterative algorithms such as LMC or DDPM. This method, widely used in optimization theory, was shown to yield strong guarantees for sampling in [Dal17a, DM17, DM19, DK19]. For DDPM, it underlies the analyses in [BZL$^+$23, GNZ25, SOB$^+$25, YY25], which establish a $W_2$-error rate of order $D/\varepsilon^2$—an improvement over the bounds derived or derivable from the first two categories. However, despite having all the necessary ingredients, these works do not reach the faster rate $\sqrt{D}/\varepsilon$. This is somewhat surprising, especially since their assumptions on $P^*$ are often quite strong, such as strong log-concavity. We believe this gap arises from not fully exploiting the smoothness of the score of the distribution obtained from $P^*$ by convolving with a Gaussian. Technically, their recursive bounds relate the error at iteration $k$ to that at iteration $k-1$ via triangle inequalities, which can be loose when the two terms involved are weakly correlated. As we show, applying the recursive approach to the **squared** Wasserstein distance yields significantly tighter control and leads to optimal rates. We believe that this improvement can be further exploited to get even faster rates using the randomized midpoint discretization [SL19, HBE20, YKD24, YY25] or to get a faster algorithm exploiting parallelization [CRYR24, ACV24, GCC24, YD25].

# 6 Numerical experiments

We supplement our theoretical results with a small-scale empirical study on CIFAR-10 [KH09], CelebA-HQ [KALL18], and LSUN-Church [YZS⁺15], evaluating the robustness of DDPMs to noise in the estimated score[6].

**Setup.** We use pretrained DDPM models from the publicly available checkpoints `google/ddpm-cifar10-32`, `google/ddpm-celebahq-256`, and `google/ddpm-church-256`, all licensed under Apache license 2.0 and hosted on HuggingFace. For each model, we follow the standard DDPM sampling procedure, and then repeat the generation process while injecting noise into the score network $s_\theta$ at every denoising step. Specifically, we replace the score function with a perturbed version $\widetilde{s}_\theta(t, \boldsymbol{x}) = s_\theta(t, \boldsymbol{x}) + \boldsymbol{\zeta}$, where $\boldsymbol{\zeta}$ is a $D$-dimensional noise vector with independent and identically distributed components. We consider 4 noise distributions: centered `Uniform`, `Gaussian`, `Laplace`, and `Student's-t` with 3 degrees of freedom. For each noise type, we evaluate 6 values for the noise scale, $\sigma \in \{0.25, 0.5, 1, 2, 3, 4\}$. All other elements of the generation pipeline—including the variance schedule, guidance scale, and number of sampling steps—are left unchanged. For each experimental setting, we generate 8192 CIFAR-10 images and 8192 CelebA-HQ images. Additional implementation details can be found in Appendix E.

**Qualitative results.** Figure 1 shows random generations for standard normal noise. We observe that injecting noise with constant variance into the score network has a negligible impact on the visual quality of the generated samples. As expected, the quality gradually degrades as the noise level increases. Additional qualitative results illustrating this phenomenon are provided in Appendix E.

**FID sensitivity.** The Fréchet Inception Distance (FID) is a widely used metric for assessing the quality of generative image models. In Figure 2, we plot the FID as a function of the noise scale $\sigma$. On CelebA-HQ, the FID increases only moderately up to $\sigma \approx 1$, while CIFAR-10 exhibits robustness up to $\sigma \approx 2$. In agreement with our theoretical findings, the shape of the noise distribution has negligible impact, only its scale matters. We also observe a sharp degradation in quality beyond a certain noise threshold, a phenomenon not accounted for by our theoretical analysis.

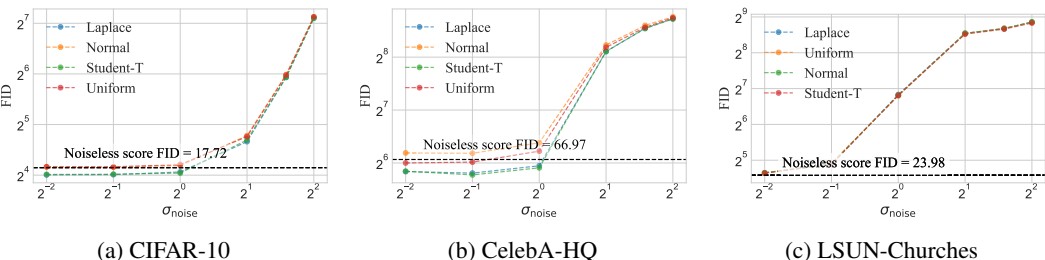

|  |  |  |
|---|---|---|
| (a) CIFAR-10 | (b) CelebA-HQ | (c) LSUN-Churches |

Figure 2: FID as a function of noise level for four distributions and different standard deviations.

# 7 Conclusion

In this paper, we provide a refined theoretical analysis of denoising diffusion probabilistic models (DDPMs), revealing two important features. First, we show that DDPMs exhibit robustness to noise in the estimated score function. Second, we establish that, when the true data-generating distribution belongs to a broad class—significantly larger than the class of log-concave distributions—DDPMs achieve fast convergence rates in the Wasserstein distance.

Our findings open several avenues for future research. One direction is the adaptation of our techniques to the analysis of kinetic Langevin diffusion-based DDPMs. It remains an open question whether such an extension would improve the dependence of the error bounds on the discretization step size. Additionally, the convergence rates we derive include terms that scale exponentially with certain parameters, such as the diameter of the support in the case of semi-log-concave targets. It is unclear whether this dependence is intrinsic to the problem or an artifact of our analysis. Finally, it would be of interest to assess the potential benefits of incorporating estimators of the Hessian of the log-density into the DDPM framework.

---

[6]Code is available at https://github.com/VahanArsenian/DiffusionWasserstein

# Acknowledgements

This work was supported by Hi! PARIS and received government funding managed by the Agence Nationale de la Recherche under the France 2030 program, references (ANR-23-IACL-0005), (ANR-23-PEIA-0004). This work was granted access to the HPC resources of IDRIS under the allocation 2025-AD011016491 made by GENCI. The work was partially supported by ERC grant SAGMOS (grant agreement No. 101201229).

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

# Appendix

## Table of Contents

# A    Classes of distributions satisfying Assumption 1

Throughout the paper we make use of Tweedie's formula [Efr11, Eq. 1.4] which takes the following form using our notation: Let $\pi_{\boldsymbol{Y}}$ be the probability density function of $\boldsymbol{Y} = \alpha\boldsymbol{X} + \beta\boldsymbol{\xi}$ where $(\boldsymbol{X}, \boldsymbol{\xi}) \sim P^* \otimes \gamma^D$, then

$$\nabla \log \pi_{\boldsymbol{Y}}(\boldsymbol{y}) = \frac{\alpha}{\beta^2}\mathbf{E}\left[\boldsymbol{X} \mid \boldsymbol{Y} = \boldsymbol{y}\right] - \frac{\boldsymbol{y}}{\beta^2}, \qquad \forall \boldsymbol{y} \in \mathbb{R}^D. \tag{10}$$

This section shows that distributions mentioned in Section 2 satisfy Assumption 1.

## A.1    Compactly supported distributions: property (a)

**Lemma 2.** *Let $P_{\boldsymbol{X},\boldsymbol{Y}}$ be a probability measure defined on $\mathcal{X} \times \mathcal{Y}$, $P_{\boldsymbol{X}}$ and $P_{\boldsymbol{X}\mid\boldsymbol{Y}=\boldsymbol{y}}$ be the marginal and the conditional distributions of $\boldsymbol{X}$. Then*

$$\mathrm{supp}(P_{\boldsymbol{X}\mid\boldsymbol{Y}=\boldsymbol{y}}) \subset \mathrm{supp}(P_{\boldsymbol{X}}).$$

*Proof.* Let $S_{\boldsymbol{X}} := \mathrm{supp}(P_{\boldsymbol{X}})$. Then by the definition of the marginal probability measure:

$$P_{\boldsymbol{X}}(S_{\boldsymbol{X}}) = P_{\boldsymbol{X},\boldsymbol{Y}}(S_{\boldsymbol{X}} \times \mathcal{Y}) = 1.$$

On the other hand, by Bayes' theorem:

$$P_{\boldsymbol{X},\boldsymbol{Y}}(S_{\boldsymbol{X}} \times \mathcal{Y}) = P_{\boldsymbol{X}\mid\boldsymbol{Y}=\boldsymbol{y}}(S_{\boldsymbol{X}})P_{\boldsymbol{Y}}(\mathcal{Y}), \tag{11}$$

where $P_{\boldsymbol{Y}}$ is the marginal probability measure of $\boldsymbol{Y}$. The proof is completed by noting that (11) yields $P_{\boldsymbol{X}\mid\boldsymbol{Y}=\boldsymbol{y}}(S_{\boldsymbol{X}}) = 1$. □

A simple consequence of Lemma 2 is that if $\mathrm{diam}(\mathrm{supp}(P_{\boldsymbol{X}})) \leqslant C$ then $\mathrm{diam}(\mathrm{supp}(P_{\boldsymbol{X}\mid\boldsymbol{Y}=\boldsymbol{y}})) \leqslant C$. Using this result, we show that a random vector $\boldsymbol{X}$ with support diameter $2\mathfrak{D}_{\boldsymbol{X}}$ satisfies Assumption 1 with $\varphi(\sigma) = \mathfrak{D}_{\boldsymbol{X}}^2$.

**Lemma 3** (Property (a) in Section 2). *Let $\boldsymbol{X} \sim P$ such that $\mathrm{diam}(\mathrm{supp}(P)) \leqslant 2\mathfrak{D}_{\boldsymbol{X}}$ and let $\boldsymbol{Y}$ be any random variable defined on the same probability space. Then*

$$\mathrm{Var}\left(\boldsymbol{X} \mid \boldsymbol{Y} = \boldsymbol{y}\right) \preccurlyeq \mathfrak{D}_{\boldsymbol{X}}^2 \mathbf{I}_D.$$

*Proof.* We need to prove that for any $\boldsymbol{v} \in \mathbb{R}^D$:

$$\boldsymbol{v}^\mathsf{T}\, \mathrm{Var}(\boldsymbol{X} \mid \boldsymbol{Y} = \boldsymbol{y})\boldsymbol{v} \leqslant \boldsymbol{v}^\mathsf{T}\left(\mathfrak{D}_{\boldsymbol{X}}^2\mathbf{I}_D\right)\boldsymbol{v},$$

which can be rewritten as:

$$\mathrm{Var}(\boldsymbol{v}^\mathsf{T}\boldsymbol{X} \mid \boldsymbol{Y} = \boldsymbol{y}) \leqslant \|\boldsymbol{v}\|^2 \mathfrak{D}_{\boldsymbol{X}}^2.$$

By dividing both sides by $\|\boldsymbol{v}\|^2$, we can rewrite the target inequality with respect to a unit vector $\boldsymbol{u} \in \mathbb{R}^D$:

$$\mathrm{Var}(\boldsymbol{u}^\mathsf{T}\boldsymbol{X} \mid \boldsymbol{Y} = \boldsymbol{y}) \leqslant \mathfrak{D}_{\boldsymbol{X}}^2.$$

Denote $Z = \boldsymbol{u}^\mathsf{T}\boldsymbol{X}$. The $\mathrm{supp}(P_Z)$ is contained in the set $\{\boldsymbol{u}^\mathsf{T}\boldsymbol{x} \mid \boldsymbol{x} \in \mathrm{supp}(P_{\boldsymbol{X}\mid\boldsymbol{Y}=\boldsymbol{y}})\}$. By Lemma 2, the $\mathrm{diam}(\mathrm{supp}(P_{\boldsymbol{X}\mid\boldsymbol{Y}=\boldsymbol{y}})) \leqslant 2\mathfrak{D}_{\boldsymbol{X}}$. Let $z_1 = \boldsymbol{u}^\mathsf{T}\boldsymbol{x}_1$ and $z_2 = \boldsymbol{u}^\mathsf{T}\boldsymbol{x}_2$ for arbitrary $\boldsymbol{x}_1, \boldsymbol{x}_2 \in \mathrm{supp}(P_{\boldsymbol{X}\mid\boldsymbol{Y}=\boldsymbol{y}})$. The distance between them is:

$$|z_1 - z_2| = |\boldsymbol{u}^\mathsf{T}\boldsymbol{x}_1 - \boldsymbol{u}^\mathsf{T}\boldsymbol{x}_2| = |\boldsymbol{u}^\mathsf{T}(\boldsymbol{x}_1 - \boldsymbol{x}_2)|.$$

By the Cauchy-Schwarz inequality:

$$|\boldsymbol{u}^\mathsf{T}(\boldsymbol{x}_1 - \boldsymbol{x}_2)| \leqslant \|\boldsymbol{u}\|_2\|\boldsymbol{x}_1 - \boldsymbol{x}_2\|_2.$$

Since $\|\boldsymbol{u}\|_2 = 1$, we write $|z_1 - z_2| \leqslant \|\boldsymbol{x}_1 - \boldsymbol{x}_2\|_2$. The maximum possible value for $\|\boldsymbol{x}_1 - \boldsymbol{x}_2\|_2$ is the diameter $2\mathfrak{D}_{\boldsymbol{X}}$. Therefore, $|z_1 - z_2| \leqslant 2\mathfrak{D}_{\boldsymbol{X}}$ for all $z_1, z_2$ in the support of $Z$. This implies that the support of $Z$ is contained within an interval $[a, b]$ such that the length of the interval $b - a \leqslant 2\mathfrak{D}_{\boldsymbol{X}}$. We now apply Popoviciu's inequality on variances [SGK10], which yields that:

$$\mathrm{Var}(Z \mid \boldsymbol{Y} = \boldsymbol{y}) \leqslant \tfrac{1}{4}\left(b - a\right)^2 \leqslant \mathfrak{D}_{\boldsymbol{X}}^2.$$

□

## A.2 Log-concave and semi-log-concave distributions: properties (b) and (c)

Random vectors with $m$-strongly log-concave densities also satisfy Assumption 1, as shown in the lemma below.

**Lemma 4** (Property (b) in Section 2)**.** *Let $(\boldsymbol{X}, \boldsymbol{\xi}) \sim P \otimes \gamma^D$, where the density of $P$, denoted as $\pi(\boldsymbol{x})$, is $m$-strongly log-concave. Then*

$$\mathrm{Var}\left(\boldsymbol{X} \mid \boldsymbol{X} + \sigma\boldsymbol{\xi} = \boldsymbol{y}\right) \preccurlyeq \frac{\sigma^2}{1 + m\sigma^2}\mathbf{I}_D.$$

*In addition, if $\boldsymbol{x} \mapsto \nabla \log \pi(\boldsymbol{x})$ is $M$-Lipschitz for some $M > 0$, then*

$$\mathrm{Var}\left(\boldsymbol{X} \mid \boldsymbol{X} + \sigma\boldsymbol{\xi} = \boldsymbol{y}\right) \succcurlyeq \frac{\sigma^2}{1 + M\sigma^2}\mathbf{I}_D.$$

*Proof.* By applying the preservation of strong log-concavity [SW14], we obtain that $\pi_{\boldsymbol{X}+\sigma\boldsymbol{\xi}}(\boldsymbol{y})$ is $\frac{m}{1+m\sigma^2}$-strongly log-concave. We then invoke Proposition 1 with parameters $\alpha = 1$ and $\beta = \sigma$, which yields

$$\frac{1}{\sigma^4} \mathrm{Var}(\boldsymbol{X} \mid \boldsymbol{Y} = \boldsymbol{y}) - \frac{1}{\sigma^2}\mathbf{I}_D \preccurlyeq -\frac{m}{1 + m\sigma^2}\mathbf{I}_D,$$

for $\boldsymbol{Y} = \boldsymbol{X} + \sigma\boldsymbol{\xi}$, from which the first desired result follows.

For the second claim, set $\boldsymbol{Y} = \boldsymbol{X} + \sigma\boldsymbol{\xi}$. The definition of semi-log-concavity yields

$$0 \preccurlyeq -\nabla^2 \log \pi(\boldsymbol{x}) \preccurlyeq M\mathbf{I}_D.$$

The conditional density of $\boldsymbol{X}$ given $\boldsymbol{Y}$ satisfies

$$\pi_{\boldsymbol{X} \mid \boldsymbol{Y} = \boldsymbol{y}}(\boldsymbol{x}) \propto \pi_{\boldsymbol{X}}(\boldsymbol{x})\pi_{\boldsymbol{Y} \mid \boldsymbol{X} = \boldsymbol{x}}(\boldsymbol{y})$$

with $\pi_{\boldsymbol{Y} \mid \boldsymbol{X} = \boldsymbol{x}}(\boldsymbol{y}) \propto \exp(-\frac{\|\boldsymbol{y} - \boldsymbol{x}\|^2}{2\sigma^2})$. Hence, the Hessian of $\pi_{\boldsymbol{X} \mid \boldsymbol{Y} = \boldsymbol{y}}(\boldsymbol{x})$ is equal to:

$$\nabla^2 \log \pi_{\boldsymbol{X} \mid \boldsymbol{Y} = \boldsymbol{y}}(\boldsymbol{x}) = \nabla^2 \log \pi_{\boldsymbol{X}}(\boldsymbol{x}) - \frac{1}{\sigma^2}\mathbf{I}_D \succcurlyeq \left[-M - \frac{1}{\sigma^2}\right]\mathbf{I}_D = -\frac{1 + M\sigma^2}{\sigma^2}\,\mathbf{I}_D.$$

The Cramer-Rao inequality implies that

$$\mathrm{Var}(\boldsymbol{X} \mid \boldsymbol{Y} = \boldsymbol{y}) \succcurlyeq -\left(\mathbf{E}[\nabla^2 \log \pi_{\boldsymbol{X} \mid \boldsymbol{Y} = \boldsymbol{y}}(\boldsymbol{X}) \mid \boldsymbol{Y} = \boldsymbol{y}]\right)^{-1} \succcurlyeq \frac{\sigma^2}{1 + M\sigma^2}\,\mathbf{I}_D$$

and the claim of the lemma follows. $\square$

Similar results hold for for semi-log-concave distributions with a compact support.

**Lemma 5** (Property (c) in Section 2)**.** *Let $(\boldsymbol{X}, \boldsymbol{\xi}) \sim P \otimes \gamma^D$ where $P$ has a density w.r.t. Lebesgue measure denoted as $\pi(\boldsymbol{x})$ and $\mathrm{diam}(\mathrm{supp}(P)) \leqslant 2\mathfrak{D}_{\boldsymbol{X}}$. If $\pi(x)$ is $M$-semi-log-concave for $M \geqslant 0$, then:*

$$\mathrm{Var}\left(\boldsymbol{X} \mid \boldsymbol{X} + \sigma\boldsymbol{\xi} = \boldsymbol{y}\right) \preccurlyeq \mathfrak{D}_{\boldsymbol{X}}^2 \wedge \frac{\sigma^2}{(1 - M\sigma^2)_+}\mathbf{I}_D.$$

*Proof.* Denote $\boldsymbol{Y} = \boldsymbol{X} + \sigma\boldsymbol{\xi}$. We obtain from the definition of semi-log-concavity that:

$$\nabla^2 \log \pi(\boldsymbol{x}) \preccurlyeq M\mathbf{I}_D.$$

The posterior of $\boldsymbol{X}$ given $\boldsymbol{Y}$ is proportional to the joint:

$$\pi(\boldsymbol{x} \mid \boldsymbol{y}) \propto \pi(\boldsymbol{x})\pi(\boldsymbol{y} \mid \boldsymbol{x})$$

with $\pi(\boldsymbol{y} \mid \boldsymbol{x}) \propto \exp(-\frac{\|\boldsymbol{y} - \boldsymbol{x}\|^2}{2\sigma^2})$. Hence, the Hessian of $\log \pi(\boldsymbol{x} \mid \boldsymbol{y})$ is equal to:

$$\nabla^2 \log \pi(\boldsymbol{x} \mid \boldsymbol{y}) = \nabla^2 \log \pi(\boldsymbol{x}) - \frac{1}{\sigma^2}\mathbf{I}_D \preccurlyeq \left[M - \frac{1}{\sigma^2}\right]\mathbf{I}_D,$$

where the last inequality follows from the semi-log-concavity of $\pi(\boldsymbol{x})$. By Brascamp-Lieb inequality [BL76], we have that:

$$\mathrm{Var}(\boldsymbol{X} \mid \boldsymbol{Y} = \boldsymbol{y}) \preccurlyeq \frac{\sigma^2}{1 - M\sigma^2}\mathbf{I}_D \tag{12}$$

whenever $M\sigma^2 \leqslant 1$. The conditional variance of $\boldsymbol{X}$ can be bounded via Lemma 3, as $P$ has a compact support:

$$\mathrm{Var}(\boldsymbol{X} \mid \boldsymbol{Y} = \boldsymbol{y}) \preccurlyeq \mathfrak{D}_{\boldsymbol{X}}^2 \mathbf{I}_D.$$

Combined with (12), we write:

$$\mathrm{Var}\left(\boldsymbol{X} \mid \boldsymbol{X} + \sigma\boldsymbol{\xi} = \boldsymbol{y}\right) \preccurlyeq \mathfrak{D}_{\boldsymbol{X}}^2 \wedge \frac{\sigma^2}{(1 - M\sigma^2)_+}\mathbf{I}_D.$$

This completes the proof of the lemma. $\qquad\square$

## A.3 Stability by orthogonal transform and concatenation: properties (d) and (e)

Afterwards, we prove that if $\boldsymbol{X}$ satisfies Assumption 1 then its rotation also satisfies Assumption 1 with the same $\varphi(\sigma)$.

**Lemma 6** (Property (d) in Section 2). *Let $(\boldsymbol{X}, \boldsymbol{\xi}) \sim P \otimes \gamma^D$ and*
$$\mathrm{Var}\left(\boldsymbol{X} \mid \boldsymbol{X} + \sigma\boldsymbol{\xi} = \boldsymbol{y}\right) \preccurlyeq \varphi(\sigma)\,\mathbf{I}_D, \qquad \forall \sigma > 0.$$

*Then for any orthonormal matrix $\mathbf{U}$, we have that:*
$$\mathrm{Var}\left(\mathbf{U}\boldsymbol{X} \mid \mathbf{U}\boldsymbol{X} + \sigma\boldsymbol{\xi}' = \boldsymbol{y}'\right) \preccurlyeq \varphi(\sigma)\,\mathbf{I}_D, \qquad \forall \sigma > 0.$$

*for $\boldsymbol{\xi}' \sim \gamma^D$ and $\boldsymbol{\xi}' \perp\!\!\!\perp \mathbf{U}\boldsymbol{X}$.*

*Proof.* Consider $\mathrm{Var}\left(\mathbf{U}\boldsymbol{X} \mid \mathbf{U}\boldsymbol{X} + \sigma\boldsymbol{\xi}' = \boldsymbol{y}\right)$. We rewrite it as:
$$\mathrm{Var}\left(\mathbf{U}\boldsymbol{X} \mid \mathbf{U}\boldsymbol{X} + \sigma\boldsymbol{\xi}' = \boldsymbol{y}'\right) = \mathbf{U}\,\mathrm{Var}\left(\boldsymbol{X} \mid \mathbf{U}\boldsymbol{X} + \sigma\boldsymbol{\xi}' = \boldsymbol{y}'\right)\mathbf{U}^\mathsf{T}$$
$$= \mathbf{U}\,\mathrm{Var}\left(\boldsymbol{X} \mid \mathbf{U}^\mathsf{T}\mathbf{U}\boldsymbol{X} + \sigma\mathbf{U}^\mathsf{T}\boldsymbol{\xi}' = \mathbf{U}^\mathsf{T}\boldsymbol{y}'\right)\mathbf{U}^\mathsf{T}$$

Let $\boldsymbol{y} := \mathbf{U}^\mathsf{T}\boldsymbol{y}'$ and $\boldsymbol{\xi} := \mathbf{U}^\mathsf{T}\boldsymbol{\xi}'$. By using the properties that $\mathbf{U}^\mathsf{T}\mathbf{U} = \mathbf{I}_D$ as $\mathbf{U}$ is orthonormal and that $\boldsymbol{\xi} \sim \gamma^D$ independently from $\boldsymbol{X}$ as $\boldsymbol{\xi}' \sim \gamma^D$ and $\boldsymbol{\xi}' \perp\!\!\!\perp \mathbf{U}\boldsymbol{X}$, we write:
$$\mathrm{Var}\left(\mathbf{U}\boldsymbol{X} \mid \mathbf{U}\boldsymbol{X} + \sigma\boldsymbol{\xi}' = \boldsymbol{y}'\right) = \mathbf{U}\,\mathrm{Var}\left(\boldsymbol{X} \mid \boldsymbol{X} + \sigma\boldsymbol{\xi} = \boldsymbol{y}\right)\mathbf{U}^\mathsf{T} \preccurlyeq \varphi(\sigma)\,\mathbf{I}_D,$$
and the claim of the lemma follows. $\qquad\square$

We now show that the concatenation of two independent random vectors satisfying Assumption 1 also satisfies Assumption 1.

**Lemma 7** (Property (e) in Section 2). *Let $(\boldsymbol{X}_1, \boldsymbol{X}_2) \sim P_1 \otimes P_2$, where $P_1$ and $P_2$ satisfy Assumption 1 for some $\varphi$. Then the concatenation of $\boldsymbol{X}_1$ and $\boldsymbol{X}_2$, denoted as $\boldsymbol{X}_1 \oplus \boldsymbol{X}_2$ also satisfies Assumption 1 for the same $\varphi$.*

*Proof.* Let $\boldsymbol{X}_1$ be $d_1$-dimensional, $\boldsymbol{X}_2$ be $d_2$-dimensional, and $D = d_1 + d_2$. Consider $\boldsymbol{\xi} \sim \gamma^D$ and independent of $(\boldsymbol{X}_1, \boldsymbol{X}_2)$. We may write
$$\boldsymbol{Y} = \boldsymbol{X}_1 \oplus \boldsymbol{X}_2 + \sigma\boldsymbol{\xi} = \boldsymbol{X}_1 \oplus \boldsymbol{X}_2 + \sigma\left(\boldsymbol{\xi}_1 \oplus \boldsymbol{\xi}_2\right) = \underbrace{\left[\boldsymbol{X}_1 + \sigma\boldsymbol{\xi}_1\right]}_{:= \boldsymbol{Y}_1} \oplus \underbrace{\left[\boldsymbol{X}_2 + \sigma\boldsymbol{\xi}_2\right]}_{:= \boldsymbol{Y}_2}.$$

We have that $(\boldsymbol{X}_1, \boldsymbol{X}_2, \boldsymbol{\xi}_1, \boldsymbol{\xi}_2)$ are mutually independent as $(\boldsymbol{X}_1, \boldsymbol{X}_2, \boldsymbol{\xi})$ are mutually independent and $\boldsymbol{\xi}_1$ and $\boldsymbol{\xi}_2$ are uncorrelated. From $(\boldsymbol{X}_1, \boldsymbol{\xi}_1) \perp\!\!\!\perp (\boldsymbol{X}_2, \boldsymbol{\xi}_2)$ we get that $(\boldsymbol{X}_1, \boldsymbol{Y}_1) \perp\!\!\!\perp (\boldsymbol{X}_2, \boldsymbol{Y}_2)$. Applying the weak union property of the conditional independence twice we get:
$$(\boldsymbol{X}_1, \boldsymbol{Y}_1) \perp\!\!\!\perp (\boldsymbol{X}_2, \boldsymbol{Y}_2) \Rightarrow \boldsymbol{X}_1 \perp\!\!\!\perp (\boldsymbol{X}_2, \boldsymbol{Y}_2) \mid \boldsymbol{Y}_1 \Rightarrow \boldsymbol{X}_1 \perp\!\!\!\perp \boldsymbol{X}_2 \mid (\boldsymbol{Y}_1, \boldsymbol{Y}_2).$$

Hence the covariance of $\boldsymbol{X}_1$ and $\boldsymbol{X}_2$ given $(\boldsymbol{Y}_1, \boldsymbol{Y}_2)$ is $\mathbf{0}$. Finally,
$$\mathrm{Var}(\boldsymbol{X}_1 \oplus \boldsymbol{X}_2 \mid \boldsymbol{Y} = \boldsymbol{y}) = \mathrm{Var}(\boldsymbol{X}_1 \oplus \boldsymbol{X}_2 \mid \boldsymbol{Y}_1 = \boldsymbol{y}_1, \boldsymbol{Y}_2 = \boldsymbol{y}_2)$$
$$= \begin{bmatrix} \mathrm{Var}(\boldsymbol{X}_1 \mid \boldsymbol{Y}_1 = \boldsymbol{y}_1, \boldsymbol{Y}_2 = \boldsymbol{y}_2) & \mathrm{Cov}(\boldsymbol{X}_1, \boldsymbol{X}_2 \mid \boldsymbol{Y}_1 = \boldsymbol{y}_1, \boldsymbol{Y}_2 = \boldsymbol{y}_2) \\ \mathrm{Cov}(\boldsymbol{X}_1, \boldsymbol{X}_2 \mid \boldsymbol{Y}_1 = \boldsymbol{y}_1, \boldsymbol{Y}_2 = \boldsymbol{y}_2) & \mathrm{Var}(\boldsymbol{X}_2 \mid \boldsymbol{Y}_1 = \boldsymbol{y}_1, \boldsymbol{Y}_2 = \boldsymbol{y}_2) \end{bmatrix}$$
$$= \begin{bmatrix} \mathrm{Var}(\boldsymbol{X}_1 \mid \boldsymbol{Y}_1 = \boldsymbol{y}_1) & \mathbf{0} \\ \mathbf{0} & \mathrm{Var}(\boldsymbol{X}_2 \mid \boldsymbol{Y}_2 = \boldsymbol{y}_2) \end{bmatrix} \preccurlyeq \varphi(\sigma)\,\mathbf{I}_D$$

where the last inequality is due to $P_1$ and $P_2$ satisfying Assumption 1. $\qquad\square$

### A.4 Convolution with a spherical Gaussian: property (f)

**Lemma 8** (Property (f) in Section 2). *Let $(\boldsymbol{W}, \boldsymbol{\zeta}) \sim P_0 \otimes \gamma^D$. If $\boldsymbol{W}$ satisfies Assumption 1 with the function $\varphi_0$, then, for every $\tau > 0$, $\boldsymbol{X} = \boldsymbol{W} + \tau\boldsymbol{\zeta}$ satisfies Assumption 1 with the function*

$$\varphi_\tau(\sigma) = \frac{\tau^2\sigma^2}{\tau^2 + \sigma^2} + \frac{\sigma^4 \varphi_0(\sqrt{\tau^2 + \sigma^2})}{(\tau^2 + \sigma^2)^2}, \qquad \forall \sigma > 0.$$

*Proof.* Let us define $\boldsymbol{Y} = \boldsymbol{X} + \sigma\boldsymbol{\xi} = \boldsymbol{W} + \tau\boldsymbol{\zeta} + \sigma\boldsymbol{\xi}$ and $\boldsymbol{\eta} := \tau\boldsymbol{\zeta} + \sigma\boldsymbol{\xi}$. Since $\boldsymbol{\xi}, \boldsymbol{\zeta} \overset{\text{i.i.d.}}{\sim} \gamma^D$ are independent of $\boldsymbol{W}$, we have $\boldsymbol{\eta} \sim \gamma^D$ with covariance $(\tau^2 + \sigma^2)\mathbf{I}_D$ and $\boldsymbol{Y} = \boldsymbol{W} + \boldsymbol{\eta}$. Equivalently,

$$\boldsymbol{Y} = \boldsymbol{W} + \sqrt{\tau^2 + \sigma^2}\,\boldsymbol{\xi}', \quad \boldsymbol{\xi}' \sim \gamma^D, \boldsymbol{\xi}' \perp\!\!\!\perp \boldsymbol{W}.$$

Using Assumption 1 with noise level $\sqrt{\tau^2 + \sigma^2}$ leads to

$$\mathrm{Var}(\boldsymbol{W} \mid \boldsymbol{Y} = \boldsymbol{y}) \preccurlyeq \varphi_0(\sqrt{\tau^2 + \sigma^2})\,\mathbf{I}_D.$$

To ease notation, we write $\mathbf{E}_{\boldsymbol{y}}$ and $\mathrm{Var}_{\boldsymbol{y}}$ to refer to the conditional expectation and conditional variance given $\boldsymbol{Y} = \boldsymbol{y}$, respectively. By the law of total variance, we have

$$\mathrm{Var}_{\boldsymbol{y}}(\boldsymbol{X}) = \mathbf{E}_{\boldsymbol{y}}\left[\mathrm{Var}(\boldsymbol{X} \mid \boldsymbol{Y} = \boldsymbol{y}, \boldsymbol{W})\right] + \mathrm{Var}_{\boldsymbol{y}}\left(\mathbf{E}[\boldsymbol{X} \mid \boldsymbol{Y} = \boldsymbol{y}, \boldsymbol{W}]\right). \tag{13}$$

We know that $\tau\boldsymbol{\zeta}$ and $\boldsymbol{\eta} = \tau\boldsymbol{\zeta} + \sigma\boldsymbol{\xi}$ are linear transforms of two independent standard Gaussians. Hence, the standard covariance calculation gives us

$$\mathrm{Var}(\tau\boldsymbol{\zeta} \mid \boldsymbol{\eta}) = \tau^2\mathbf{I}_D - \tau^2\mathbf{I}_D(\tau^2 + \sigma^2)^{-1}\mathbf{I}_D\tau^2\mathbf{I}_D$$

$$= \frac{\tau^2\sigma^2}{\tau^2 + \sigma^2}\mathbf{I}_D.$$

And since $\mathrm{Var}(\boldsymbol{X} \mid \boldsymbol{Y} = \boldsymbol{y}, \boldsymbol{W}) = \mathrm{Var}(\tau\boldsymbol{\zeta} \mid \boldsymbol{\eta})$, we get the first part of (13) equal to

$$\mathbf{E}_{\boldsymbol{y}}\left[\mathrm{Var}(\boldsymbol{X} \mid \boldsymbol{Y} = \boldsymbol{y}, \boldsymbol{W})\right] = \frac{\tau^2\sigma^2}{\tau^2 + \sigma^2}\mathbf{I}_D.$$

For the second term, since $\boldsymbol{\zeta}, \boldsymbol{\xi} \overset{\text{i.i.d.}}{\sim} \mathcal{N}(\mathbf{0}, \mathbf{I}_D)$, then the corresponding $2D$-dimensional vector

$$\begin{pmatrix} \tau\boldsymbol{\zeta} \\ \boldsymbol{\eta} \end{pmatrix} \sim \mathcal{N}(\mathbf{0}, \boldsymbol{\Sigma}), \quad \text{with } \boldsymbol{\Sigma} = \begin{pmatrix} \mathrm{Var}(\tau\boldsymbol{\zeta}) & \mathrm{Cov}(\tau\boldsymbol{\zeta}, \boldsymbol{\eta}) \\ \mathrm{Cov}(\boldsymbol{\eta}, \tau\boldsymbol{\zeta}) & \mathrm{Var}(\boldsymbol{\eta}) \end{pmatrix} = \begin{pmatrix} \tau^2\mathbf{I}_D & \tau^2\mathbf{I}_D \\ \tau^2\mathbf{I}_D & (\tau^2 + \sigma^2)\mathbf{I}_D \end{pmatrix}.$$

So, the conditional expectation that we are interested in will be equal to

$$\mathbf{E}[\tau\boldsymbol{\zeta} \mid \boldsymbol{\eta}] = \tau^2\mathbf{I}_D(\tau^2 + \sigma^2)^{-1}\mathbf{I}_D\boldsymbol{\eta} = \frac{\tau^2}{\tau^2 + \sigma^2}\boldsymbol{\eta}.$$

Under the conditioning on both $\boldsymbol{Y}$ and $\boldsymbol{W}$, the quantity $\boldsymbol{\eta} = \boldsymbol{Y} - \boldsymbol{W}$ is deterministic. Therefore,

$$\mathbb{E}[\boldsymbol{X} \mid \boldsymbol{Y} = \boldsymbol{y}, \boldsymbol{W}] = \boldsymbol{W} + \mathbb{E}[\tau\boldsymbol{\zeta} \mid \boldsymbol{\eta} = \boldsymbol{y} - \boldsymbol{W}]$$

$$= \boldsymbol{W} + \frac{\tau^2}{\tau^2 + \sigma^2}(\boldsymbol{y} - \boldsymbol{W})$$

$$= \frac{\sigma^2}{\tau^2 + \sigma^2}\boldsymbol{W} + \frac{\tau^2}{\tau^2 + \sigma^2}\boldsymbol{y}.$$

Given $\boldsymbol{Y} = \boldsymbol{y}$, the second term is deterministic, so

$$\mathrm{Var}_{\boldsymbol{y}}\left(\mathbf{E}[\boldsymbol{X} \mid \boldsymbol{Y} = \boldsymbol{y}, \boldsymbol{W}]\right) = \left(\frac{\sigma^2}{\tau^2 + \sigma^2}\right)^2 \mathrm{Var}(\boldsymbol{W} \mid \boldsymbol{Y} = \boldsymbol{y}) \preccurlyeq \frac{\sigma^4 \varphi_0(\sqrt{\tau^2 + \sigma^2})}{(\tau^2 + \sigma^2)^2}\mathbf{I}_D.$$

Adding the two components gives us

$$\mathrm{Var}(\boldsymbol{X} \mid \boldsymbol{Y} = \boldsymbol{y}) \preccurlyeq \left(\frac{\tau^2\sigma^2}{\tau^2 + \sigma^2} + \frac{\sigma^4 \varphi_0(\sqrt{\tau^2 + \sigma^2})}{(\tau^2 + \sigma^2)^2}\right)\mathbf{I}_D,$$

which proves the lemma with $\varphi_\tau(\sigma) = \frac{\tau^2\sigma^2}{\tau^2 + \sigma^2} + \frac{\sigma^4 \varphi_0(\sqrt{\tau^2 + \sigma^2})}{(\tau^2 + \sigma^2)^2}$. $\qquad\square$

### A.5 Convolution of a semi-log-concave and a compactly supported distribution: property (g)

**Lemma 9** (Property (g) in Section 2). *If $P^* = P_{\mathsf{slc}} \star P_{\mathsf{cmpct}}$, where $P_{\mathsf{slc}}$ is an $m$-strongly log-concave distribution with an $M$-Lipschitz score function, and $P_{\mathsf{cmpct}}$ is supported on a compact set with diameter $2\mathfrak{D}$, then $P^*$ satisfies Assumption 1 with*

$$\varphi(\sigma) = \frac{\sigma^2}{1 + m\sigma^2} + \frac{\mathfrak{D}^2 M^2 \sigma^4}{(1 + M\sigma^2)^2}, \qquad \forall \sigma > 0,$$

*Proof.* Let $\boldsymbol{W} \sim P_{\mathsf{cmpct}}$ and $\boldsymbol{\zeta} \sim P_{\mathsf{slc}}$ be two independent random vectors so that $\boldsymbol{X} = \boldsymbol{W} + \boldsymbol{\zeta} \sim P^*$. This means that for some compact set $\mathcal{K}$ with diameter $2\mathfrak{D}$, we have $\operatorname{Var}(\boldsymbol{W}) \leqslant 4\mathfrak{D}^2$, and that the density $\pi_{\boldsymbol{\zeta}}$ is continuously differentiable with a score function $\boldsymbol{s}_{\boldsymbol{\zeta}}$ satisfying

$$m\|\boldsymbol{x} - \boldsymbol{x}'\|^2 \leqslant (\boldsymbol{x} - \boldsymbol{x}')^{\mathsf{T}} \big( \boldsymbol{s}_{\boldsymbol{\zeta}}(\boldsymbol{x}) - \boldsymbol{s}_{\boldsymbol{\zeta}}(\boldsymbol{x}') \big) \leqslant M \|\boldsymbol{x} - \boldsymbol{x}'\|^2.$$

For $\boldsymbol{\xi} \perp\!\!\!\perp (\boldsymbol{W}, \boldsymbol{\zeta})$ such that $\boldsymbol{\xi} \sim \gamma^D$, and for $\boldsymbol{Y} = \boldsymbol{X} + \sigma \boldsymbol{\xi}$, we have to prove that

$$\operatorname{Var}(\boldsymbol{X} \mid \boldsymbol{Y} = \boldsymbol{y}) \preccurlyeq \Big( \frac{\sigma^2}{1 + m\sigma^2} + \frac{\mathfrak{D}^2 M^2 \sigma^4}{(1 + M\sigma^2)^2} \Big) \mathbf{I}_D.$$

As before, to ease notation, we write $\mathbf{E}_{\boldsymbol{y}}$ and $\operatorname{Var}_{\boldsymbol{y}}$ to refer to the conditional expectation and conditional variance given $\boldsymbol{Y} = \boldsymbol{y}$, respectively. By the law of total variance, we have

$$\operatorname{Var}_{\boldsymbol{y}}(\boldsymbol{X}) = \mathbf{E}_{\boldsymbol{y}} \left[ \operatorname{Var}(\boldsymbol{X} \mid \boldsymbol{Y} = \boldsymbol{y}, \boldsymbol{W}) \right] + \operatorname{Var}_{\boldsymbol{y}} \left( \mathbf{E}[\boldsymbol{X} \mid \boldsymbol{Y} = \boldsymbol{y}, \boldsymbol{W}] \right). \tag{14}$$

Since the random vector $\boldsymbol{\zeta}$ is $m$-strongly log-concave, it follows from Lemma 4 that

$$\operatorname{Var}(\boldsymbol{\zeta} \mid \boldsymbol{\zeta} + \sigma \boldsymbol{\xi} = \boldsymbol{y}') \leqslant \frac{\sigma^2}{1 + m\sigma^2}, \qquad \forall \boldsymbol{y}' \in \mathbb{R}^D.$$

Therefore,

$$\operatorname{Var}(\boldsymbol{X} \mid \boldsymbol{Y} = \boldsymbol{y}, \boldsymbol{W} = \boldsymbol{w}) = \operatorname{Var}(\boldsymbol{\zeta} \mid \boldsymbol{\zeta} + \sigma \boldsymbol{\xi} = \boldsymbol{y} - \boldsymbol{w}) \leqslant \frac{\sigma^2}{1 + m\sigma^2}, \qquad \forall \boldsymbol{y}, \boldsymbol{w} \in \mathbb{R}^D.$$

Hence, $\operatorname{Var}(\boldsymbol{X} \mid \boldsymbol{Y} = \boldsymbol{y}, \boldsymbol{W}) \leqslant \frac{\sigma^2}{1 + m\sigma^2}$ almost surely. This implies that

$$\mathbf{E}_{\boldsymbol{y}} \left[ \operatorname{Var}(\boldsymbol{X} \mid \boldsymbol{Y} = \boldsymbol{y}, \boldsymbol{W}) \right] \preccurlyeq \frac{\sigma^2}{1 + m\sigma^2} \mathbf{I}_D.$$

We switch to assessing the second term in (14). It holds that

$$
\begin{aligned}
\mathbf{E}[\boldsymbol{X} \mid \boldsymbol{Y} = \boldsymbol{y}, \boldsymbol{W} = \boldsymbol{w}] &\overset{\text{①}}{=} \boldsymbol{w} + \mathbf{E}[\boldsymbol{\zeta} \mid \boldsymbol{\zeta} + \sigma \boldsymbol{\xi} = \boldsymbol{y} - \boldsymbol{w}, \boldsymbol{W} = \boldsymbol{w}] \\
&\overset{\text{②}}{=} \boldsymbol{w} + \mathbf{E}[\boldsymbol{\zeta} \mid \boldsymbol{\zeta} + \sigma \boldsymbol{\xi} = \boldsymbol{y} - \boldsymbol{w}] \\
&\overset{\text{③}}{=} \boldsymbol{w} + \sigma^2 \nabla \log \pi_{\boldsymbol{\zeta} + \sigma \boldsymbol{\xi}}(\boldsymbol{y} - \boldsymbol{w}) + \boldsymbol{y} - \boldsymbol{w} \\
&= \boldsymbol{y} + \sigma^2 \nabla \log \pi_{\boldsymbol{\zeta} + \sigma \boldsymbol{\xi}}(\boldsymbol{y} - \boldsymbol{w}),
\end{aligned}
$$

where ① is a consequence of $\boldsymbol{X} = \boldsymbol{W} + \boldsymbol{\zeta}$, ② follows from the independence of $\boldsymbol{\zeta}$ and $\boldsymbol{W}$, ③ is obtained by the Tweedie formula recalled in (10). Let us set $\psi(\boldsymbol{w}) = \nabla \log \pi_{\boldsymbol{\zeta} + \sigma \boldsymbol{\xi}}(\boldsymbol{y} - \boldsymbol{w})$. The second claim of Lemma 4 combined with Proposition 1 implies that $\psi$ is Lipschitz-continuous with the constant $M/(1 + M\sigma^2)$. Therefore,

$$\operatorname{Var}_{\boldsymbol{y}}(\mathbf{E}[\boldsymbol{X} \mid \boldsymbol{Y} = \boldsymbol{y}, \boldsymbol{W}]) = \sigma^4 \operatorname{Var}_{\boldsymbol{y}} \left( \psi(\boldsymbol{W}) \right) \preccurlyeq \frac{M^2 \sigma^4}{(1 + M\sigma^2)^2} \operatorname{Var}_{\boldsymbol{y}} \left( \boldsymbol{W} \right) \leqslant \frac{M^2 \sigma^4 \mathfrak{D}^2}{(1 + M\sigma^2)^2} \mathbf{I}_D,$$

where in the last step we used Lemma 3. $\qquad\square$

# B  Proof of Lemma 1

We start by first proving that:

$$\sup_{P^* \in \mathcal{N}} \frac{\mathsf{d}^2_{\mathsf{TV}}(Q_D^{T,\boldsymbol{s}^*}; P^*)}{\mathsf{d}^2_{\mathsf{TV}}(\gamma^D; P^*)} \bigvee \frac{\mathsf{d}_{\mathsf{KL}}(Q_D^{T,\boldsymbol{s}^*} \| P^*)}{\mathsf{d}_{\mathsf{KL}}(\gamma^D \| P^*)} \leqslant e^{-2T}$$

The data processing inequality [PW17] states that:

$$\mathsf{d}_{\mathsf{TV}}(Q_D^{T,\boldsymbol{s}^*}; P^*) \leqslant \mathsf{d}_{\mathsf{TV}}(\gamma^D; P_T^*); \qquad \mathsf{d}_{\mathsf{KL}}(Q_D^{T,\boldsymbol{s}^*}; P^*) \leqslant \mathsf{d}_{\mathsf{KL}}(\gamma^D; P_T^*).$$

Combined with the concentration property of Ornstein–Uhlenbeck process [GZ24, EGZ19]:

$$\mathsf{d}_{\mathsf{TV}}(\gamma^D; P_T^*) \leqslant \mathsf{d}_{\mathsf{TV}}(\gamma^D; P^*)e^{-T}; \qquad \mathsf{d}_{\mathsf{KL}}(\gamma^D; P_T^*) \leqslant \mathsf{d}_{\mathsf{KL}}(\gamma^D; P^*)e^{-2T}.$$

gives the desired result.

We now focus on a subset of $\mathcal{N}' \subset \mathcal{N}$ that contains $D$ dimensional Gaussian distributions with mean $\boldsymbol{0}$ and $(1 + \sigma^2)\mathbf{I}_D$ covariance matrix with $\sigma > 0$. Clearly

$$\sup_{P^* \in \mathcal{N}'} \frac{\mathsf{W}_2(Q_D^{T,\boldsymbol{s}^*}; P^*)}{\mathsf{W}_2(P^*; \gamma^D)} \leqslant \sup_{P^* \in \mathcal{N}} \frac{\mathsf{W}_2(Q_D^{T,\boldsymbol{s}^*}; P^*)}{\mathsf{W}_2(P^*; \gamma^D)}$$

Let $\boldsymbol{X}_t$ be defined by Equation (2), then the distribution of $\boldsymbol{X}_t$ is $\mathcal{N}(\boldsymbol{0}, (e^{-2t}\sigma^2 + 1)\,\mathbf{I}_D)$. Hence, the true score function is

$$\widetilde{\boldsymbol{s}}(\boldsymbol{x}) = -\boldsymbol{x}/\sigma^2(t),$$

where $\sigma^2(t) = e^{-2t}\sigma^2 + 1$. Equation (4) obtains the following form under this score function:

$$\mathrm{d}\widetilde{\boldsymbol{Y}}_t = \left[\widetilde{\boldsymbol{Y}}_t\left(1 - \frac{2}{\sigma^2(T-t)}\right)\right]\mathrm{d}t + \sqrt{2}\,\mathrm{d}\widetilde{\boldsymbol{B}}_t.$$

The integrating factor for the SDE is:

$$
\begin{aligned}
I(t) &= \exp\left(-\int_0^t 1 - \frac{2}{\sigma^2(T-u)}\,\mathrm{d}u\right) \\
&= \exp\left(-t + \int_0^t \frac{2}{\exp(2(u-T))\sigma^2 + 1}\,\mathrm{d}u\right) \\
&= \exp\left(-t + 2t + \log\left(\frac{\sigma^2 + e^{2T}}{\sigma^2 e^{2t} + e^{2T}}\right)\right) \\
&= e^t \frac{\sigma^2 + e^{2T}}{\sigma^2 e^{2t} + e^{2T}}.
\end{aligned}
$$

From Itô's product rule applied to $I(t)\widetilde{\boldsymbol{Y}}_t$, we get:

$$\mathrm{d}I(t)\widetilde{\boldsymbol{Y}}_t = I(t)\left[f(t)\widetilde{\boldsymbol{Y}}_t\,\mathrm{d}t + \sqrt{2}\,\mathrm{d}\widetilde{\boldsymbol{B}}_t\right] - I(t)f(t)\widetilde{\boldsymbol{Y}}_t\,\mathrm{d}t = \sqrt{2}I(t)\,\mathrm{d}\widetilde{\boldsymbol{B}}_t, \tag{15}$$

where we have used the fact that $\mathrm{d}I(t) = -I(t)\left(1 - \frac{2}{\sigma^2(T-t)}\right)\mathrm{d}t$.

Integrating both sides of (15) from 0 to $t$:

$$I(t)\widetilde{\boldsymbol{Y}}_t = \widetilde{\boldsymbol{Y}}_0 + \sqrt{2}\int_0^t I(u)\,\mathrm{d}\widetilde{\boldsymbol{B}}_u$$

from which:

$$\widetilde{\boldsymbol{Y}}_t = \frac{\widetilde{\boldsymbol{Y}}_0 + \sqrt{2}\int_0^t I(u)\,\mathrm{d}\widetilde{\boldsymbol{B}}_u}{I(t)}. \tag{16}$$

Note that $\widetilde{Y}_0 \sim \gamma^D$. Combined with the fact that $I(t)$ is a deterministic function, we infer from (16) that $\widetilde{Y}_t$ is a zero mean Gaussian random variable. So the Wasserstein distance between $\gamma^D$ and the distribution of $\widetilde{Y}_t$ depends only on the covariance matrices:

$$\mathsf{W}_2(Q_D^{T,\boldsymbol{s}^*}; P^*) = \|\sigma_{\widetilde{\boldsymbol{Y}}_t}\mathbf{I}_D - \sqrt{\sigma^2+1}\mathbf{I}_D\|_F = \big|\sigma_{\widetilde{\boldsymbol{Y}}_t} - \sqrt{\sigma^2+1}\big|\sqrt{D}, \tag{17}$$

where $\sigma_{\widetilde{\boldsymbol{Y}}_t}^2\mathbf{I}_D$ is the covariance of $\widetilde{\boldsymbol{Y}}_t$.

Let $\boldsymbol{Z}_t := \sqrt{2}\int_0^t I(u)\,\mathrm{d}\widetilde{\boldsymbol{B}}_u$. Hence, $\boldsymbol{Z}_t \sim \mathcal{N}(\boldsymbol{0}, 2\int_0^t I^2(u)\,\mathrm{d}u\,\mathbf{I}_D)$ and it is independent of $\widetilde{\boldsymbol{Y}}_0$. The variance of $\boldsymbol{Z}_t$ is:

$$\sigma_{\boldsymbol{z}_t}^2 = 2\int_0^t I^2(u)du = \frac{(e^{2t}-1)(e^{2T}+\sigma^2)}{e^{2T}+\sigma^2 e^{2t}} \quad \text{and} \quad \sigma_{\boldsymbol{z}_T}^2 = \frac{(1-e^{-2T})(e^{2T}+\sigma^2)}{\sigma^2+1}$$

The variance of $\widetilde{\boldsymbol{Y}}_T$ can be computed from Equation (16):

$$\sigma_{\widetilde{\boldsymbol{Y}}_T}^2 = \frac{1+\sigma_{\boldsymbol{Z}_T}^2}{I^2(T)} = \frac{(\sigma^2+1)(2\sigma^2 e^{2T} - \sigma^2 + e^{4T})}{(\sigma^2+e^{2T})^2} = (\sigma^2+1)\Big[1 - \frac{\sigma^2(\sigma^2+1)}{(\sigma^2+e^{2T})^2}\Big].$$

Plugging in the value of $\sigma_{\widetilde{\boldsymbol{Y}}_T}$ into (17) we get:

$$\mathsf{W}_2(Q_D^{T,\boldsymbol{s}^*}; P^*) = \left(1 - \Big\{1 - \frac{\sigma^2(\sigma^2+1)}{(\sigma^2+e^{2T})^2}\Big\}^{1/2}\right)\sqrt{(\sigma^2+1)D} \overset{\sigma\to\infty}{\sim} \sigma\sqrt{D}.$$

We note that $\mathsf{W}_2(P^*; \gamma^D) = |\sqrt{\sigma^2+1}-1|\sqrt{D} \overset{\sigma\to\infty}{\sim} \sigma\sqrt{D}$, so we have

$$r(\sigma) = \frac{\mathsf{W}_2(Q_D^{T,\boldsymbol{s}^*}; P^*)}{\mathsf{W}_2(P^*; \gamma^D)} \xrightarrow{\sigma\to\infty} 1.$$

Hence,

$$\sup_{P^*\in\mathcal{N}'} \frac{\mathsf{W}_2(Q_D^{T,\boldsymbol{s}^*}; P^*)}{\mathsf{W}_2(P^*; \gamma^D)} \geqslant 1.$$

When combined with the established contraction behavior of the backward diffusion—operating with the true score function—in the 2-Wasserstein metric for Gaussian distributions [EGZ19], we get:

$$1 \leqslant \sup_{P^*\in\mathcal{N}} \frac{\mathsf{W}_2(Q_D^{T,\boldsymbol{s}^*}; P^*)}{\mathsf{W}_2(P^*; \gamma^D)} \leqslant 1.$$

## C   Proofs of the main results

We recall that $P^*$ is the target distribution and $P_t^* = \alpha_t P^* + \beta_t \gamma^D$ is the distribution of the forward process at time $t > 0$, with $\alpha_t = e^{-t} = \sqrt{1-\beta_t^2}$. We also fix some $T > 0$ and define $\boldsymbol{Y}_t = \boldsymbol{X}_{T-t}$ and $Q_t^* = \mathrm{Law}(\boldsymbol{Y}_t)$; $\boldsymbol{Y}_t$ is the state of the backward process (3). We set $\widetilde{P}_k$ to be the law of $\boldsymbol{Z}_k$ defined by (5) so that $P^{\mathsf{DDPM}} = \widetilde{P}_{K+1}$. Throughout this proof, we will repeatedly use the following notation:

$$\bar{m}_2 = 1 \vee (\|\boldsymbol{X}\|_{\mathbb{L}_2}/\sqrt{D}),$$

$$\varepsilon_k^b = \|\mathbf{E}[\widetilde{\boldsymbol{s}}(T-t_k, \boldsymbol{Z}_k)\,|\,\mathcal{F}_k] - \boldsymbol{s}(T-t_k, \boldsymbol{Z}_k)\|_{\mathbb{L}_2}, \quad \varepsilon^b = \max_k \varepsilon_k^b$$

$$\varepsilon_k^v = \|\widetilde{\boldsymbol{s}}(T-t_k, \boldsymbol{Z}_k) - \mathbf{E}[\widetilde{\boldsymbol{s}}(T-t_k, \boldsymbol{Z}_k)\,|\,\mathcal{F}_k]\|_{\mathbb{L}_2}, \quad \varepsilon^v = \max_k \varepsilon_k^v.$$

### C.1   Main recursion

We set $T = t_{K+1}$ and consider a version of the continuous-time process $(\boldsymbol{Y}_t)_{0\leqslant t\leqslant T}$ and the discrete-time process $(\boldsymbol{Z}_k)_{0\leqslant k\leqslant K+1}$ defined on the same probability space and coupled by the relation $\boldsymbol{\xi}_{k+1} = (\widetilde{\boldsymbol{B}}_{t_{k+1}} - \widetilde{\boldsymbol{B}}_{t_k})/\sqrt{h_k}$. We then use the definition of the Wasserstein distance to infer that

$$\mathsf{W}_2(P^*, \widetilde{P}_{K+1}) = \mathsf{W}_2(Q_{t_{K+1}}^*, \widetilde{P}_{K+1}) \leqslant \|\boldsymbol{Y}_{t_{K+1}} - \boldsymbol{Z}_{K+1}\|_{\mathbb{L}_2}. \tag{18}$$

Combining (3) and (5), in conjunction with the relation $\sqrt{h_k}\,\boldsymbol{\xi}_{k+1} = (\widetilde{\boldsymbol{B}}_{t_{k+1}} - \widetilde{\boldsymbol{B}}_{t_k})$, we get

$$
\begin{aligned}
\boldsymbol{Y}_{t_{k+1}} - \boldsymbol{Z}_{k+1} = {}& (1 + h_k)(\boldsymbol{Y}_{t_k} - \boldsymbol{Z}_k) + 2h_k\big(\boldsymbol{s}(T - t_k, \boldsymbol{Y}_{t_k}) - \boldsymbol{s}(T - t_k, \boldsymbol{Z}_k)\big) \\
& - 2h_k\big(\widetilde{\boldsymbol{s}}(T - t_k, \boldsymbol{Z}_k) - \boldsymbol{s}(T - t_k, \boldsymbol{Z}_k)\big) \\
& + \int_{t_k}^{t_{k+1}} \Big\{ \boldsymbol{Y}_t - \boldsymbol{Y}_{t_k} + 2\boldsymbol{s}(T - t, \boldsymbol{Y}_t) - 2\boldsymbol{s}(T - t_k, \boldsymbol{Y}_{t_k}) \Big\}\, \mathrm{d}t.
\end{aligned}
\tag{19}
$$

In what follows, we use the notation $\boldsymbol{\Delta}_k = \boldsymbol{Y}_{t_k} - \boldsymbol{Z}_k$ and

$$
\begin{aligned}
\boldsymbol{U}_k &= \boldsymbol{s}(T - t_k, \boldsymbol{Y}_{t_k}) - \boldsymbol{s}(T - t_k, \boldsymbol{Z}_k); \\
\boldsymbol{\zeta}_k &= \widetilde{\boldsymbol{s}}(T - t_k, \boldsymbol{Z}_k) - \boldsymbol{s}(T - t_k, \boldsymbol{Z}_k) \\
\boldsymbol{V}_k &= \int_{t_k}^{t_{k+1}} \Big\{ \boldsymbol{Y}_t - \boldsymbol{Y}_{t_k} + 2\boldsymbol{s}(T - t, \boldsymbol{Y}_t) - 2\boldsymbol{s}(T - t_k, \boldsymbol{Y}_{t_k}) \Big\}\, \mathrm{d}t.
\end{aligned}
\tag{20}
$$

This allows us to rewrite (19) as follows

$$
\boldsymbol{\Delta}_{k+1} = (1 + h_k)\boldsymbol{\Delta}_k + 2h_k\boldsymbol{U}_k - 2h_k\boldsymbol{\zeta}_k + \boldsymbol{V}_k.
\tag{21}
$$

In view of (18), we are interested in bounding the term

$$
x_K := \|\boldsymbol{\Delta}_K\|_{\mathbb{L}_2}.
$$

We will proceed by establishing a recursive inequality upper bounding $x_{k+1}$ by a simple expression involving $x_k$, and then by unfolding this recursive inequality.

Let us introduce the filtration $(\mathcal{F}_k)_{k \in \mathbb{N}}$. The first element of this sequence is the $\sigma$-algebra generated by $\boldsymbol{Y}_0$ and $\boldsymbol{Z}_0$. Then, each $\mathcal{F}_{k+1}$ is obtained by extending $\mathcal{F}_k$ to the smallest $\sigma$-algebra for which both $\boldsymbol{\zeta}_k$ and the process $(\tilde{\boldsymbol{B}}_t - \tilde{\boldsymbol{B}}_{t_k})_{t \in [t_k; t_{k+1}]}$ are measurable. Note that $Z_k$ is necessarily $\mathcal{F}_k$-measurable, but the same is not true for $\boldsymbol{\zeta}_k$. Indeed, the estimator $\tilde{s}(T - t_k, \cdot)$ may depend on some random variables that are not in $\mathcal{F}_k$.

It is clear that

$$
\begin{aligned}
\mathbf{E}\big[\|\boldsymbol{\Delta}_{k+1}\|^2\big] &= \mathbf{E}\big[\|\mathbf{E}[\boldsymbol{\Delta}_{k+1}\,|\,\mathcal{F}_k]\|^2\big] + \mathbf{E}\big[\|\boldsymbol{\Delta}_{k+1} - \mathbf{E}[\boldsymbol{\Delta}_{k+1}\,|\,\mathcal{F}_k]\|^2\big] \\
&= \|\mathbf{E}[\boldsymbol{\Delta}_{k+1}\,|\,\mathcal{F}_k]\|_{\mathbb{L}_2}^2 + \|\boldsymbol{\Delta}_{k+1} - \mathbf{E}[\boldsymbol{\Delta}_{k+1}\,|\,\mathcal{F}_k]\|_{\mathbb{L}_2}^2.
\end{aligned}
\tag{22}
$$

From (21), by the triangle inequality,

$$
\|\mathbf{E}[\boldsymbol{\Delta}_{k+1}\,|\,\mathcal{F}_k]\|_{\mathbb{L}_2} \leqslant \|(1 + h_k)\boldsymbol{\Delta}_k + 2h_k\boldsymbol{U}_k\|_{\mathbb{L}_2} + 2h_k\|\mathbf{E}[\boldsymbol{\zeta}_k\,|\,\mathcal{F}_k]\|_{\mathbb{L}_2} + \|\mathbf{E}[\boldsymbol{V}_k\,|\,\mathcal{F}_k]\|_{\mathbb{L}_2}.
\tag{23}
$$

Furthermore,

$$
\|\boldsymbol{\Delta}_{k+1} - \mathbf{E}[\boldsymbol{\Delta}_{k+1}\,|\,\mathcal{F}_k]\|_{\mathbb{L}_2} \leqslant 2h_k\|\boldsymbol{\zeta}_k - \mathbf{E}[\boldsymbol{\zeta}_k\,|\,\mathcal{F}_k]\|_{\mathbb{L}_2} + \|\boldsymbol{V}_k - \mathbf{E}[\boldsymbol{V}_k\,|\,\mathcal{F}_k]\|_{\mathbb{L}_2}.
\tag{24}
$$

Combining displays (22), (23) and (24), we arrive at

$$
\begin{aligned}
\mathbf{E}\big[\|\boldsymbol{\Delta}_{k+1}\|^2\big] \leqslant {}& \bigg( \|(1 + h_k)\boldsymbol{\Delta}_k + 2h_k\boldsymbol{U}_k\|_{\mathbb{L}_2} + 2h_k \underbrace{\|\mathbf{E}[\boldsymbol{\zeta}_k\,|\,\mathcal{F}_k]\|_{\mathbb{L}_2}}_{\varepsilon_k^b := \text{bias of estim. score}} + \underbrace{\|\mathbf{E}[\boldsymbol{V}_k\,|\,\mathcal{F}_k]\|_{\mathbb{L}_2}}_{\mathfrak{B}_k := \text{bias of discr. error}} \bigg)^2 \\
& + \bigg( 2h_k \underbrace{\|\boldsymbol{\zeta}_k - \mathbf{E}[\boldsymbol{\zeta}_k\,|\,\mathcal{F}_k]\|_{\mathbb{L}_2}}_{\varepsilon_k^v := \text{variance of estim. score}} + \underbrace{\|\boldsymbol{V}_k - \mathbf{E}[\boldsymbol{V}_k\,|\,\mathcal{F}_k]\|_{\mathbb{L}_2}}_{\mathfrak{V}_k := \text{variance of discr. error}} \bigg)^2.
\end{aligned}
$$

In what follows, it is convenient to use the following notation: for every $k \in \mathbb{N}$, let $\alpha_k = e^{-(T - t_k)}$ and $\beta_k^2 = 1 - \alpha_k^2$.

**Lemma 10.** *If $P^*$ satisfies Assumption 1 with a function $\varphi$ and*

$$
h_k\left(\frac{1 + \alpha_k^2}{1 - \alpha_k^2} + m_k\right) \leqslant 2, \quad \text{for} \quad m_k = 1 + \frac{2\alpha_k^2}{1 - \alpha_k^2}\left(1 - \frac{\varphi(\beta_k/\alpha_k)}{1 - \alpha_k^2}\right)
\tag{25}
$$

*then,*

$$
\|(1 + h_k)\boldsymbol{\Delta}_k + 2h_k\boldsymbol{U}_k\|_{\mathbb{L}_2} \leqslant \big(1 - m_k h_k\big)\|\boldsymbol{\Delta}_k\|_{\mathbb{L}_2}.
\tag{26}
$$

Lemma 10 implies that

$$x_{k+1}^2 \leqslant \left((1 - m_k h_k)x_k + 2h_k \varepsilon_k^b + \mathfrak{B}_k\right)^2 + \left(2h_k \varepsilon_k^v + \mathfrak{V}_k\right)^2. \tag{27}$$

The next lemma which can be easily deduced by induction applying the Minkowski inequality, will be used to derive a global bound on the error $x_K$ from recursive inequalities upper bounding the error $x_{k+1}$ at the $(k+1)$th step by the one of the $k$th step.

**Lemma 11.** *Let $(A_k)_{k\in\mathbb{N}}$, $(B_k)_{k\in\mathbb{N}}$ and $(C_k)_{k\in\mathbb{N}}$ be three sequences of real numbers such that $B_k \geqslant 0$ and $C_k \geqslant 0$ for every $k$. If $(x_k)_{k\in\mathbb{N}}$ satisfies the recursive inequality*

$$x_{k+1}^2 \leqslant (e^{A_k} x_k + B_k)^2 + C_k^2, \qquad \forall k \geqslant 0,$$

*then, for $\bar{A}_k = A_0 + \ldots + A_k$,*

$$x_{k+1} \leqslant e^{\bar{A}_k} x_0 + \sum_{j=0}^{k} e^{\bar{A}_k - \bar{A}_j} B_j + \left( \sum_{j=0}^{k} e^{2(\bar{A}_k - \bar{A}_j)} C_j^2 \right)^{1/2}.$$

For the subsequent steps of the proof, we leverage the properties of discretization. We begin with the portion employing constant step-sizes. This discretization is applied in the time interval where the inequality from (26) yields a near-contraction. This is equivalent to considering the values of $k$ for which $m_k$ in (27) is positive and bounded away from zero.

**Lemma 12.** *If $T$ and $a \geqslant 1$ are real numbers such that $T \geqslant \frac{1}{2}\log(6a)$. Let $K_0 \in \mathbb{N}$ be such that for every $k \in \{0, 1, \ldots, K_0\}$,*

$$0 \leqslant t_k \leqslant T - \tfrac{1}{2}\log(6a), \qquad h_k \leqslant 0.7, \qquad \varphi(\beta_k/\alpha_k) \leqslant a.$$

*Then, for $\alpha_k = e^{-(T-t_k)}$, we have $\alpha_k^2 \leqslant 1/(6a)$ as well as*

$$m_k \geqslant 1 + \frac{2\alpha_k^2}{1 - \alpha_k^2}\left(1 - \frac{a}{1 - \alpha_k^2}\right) \geqslant 1/3, \qquad \text{and} \qquad h_k\left(\frac{1 + \alpha_k^2}{1 - \alpha_k^2} + m_k\right) \leqslant 2,$$

*for all $k = 0, \ldots, K_0$.*

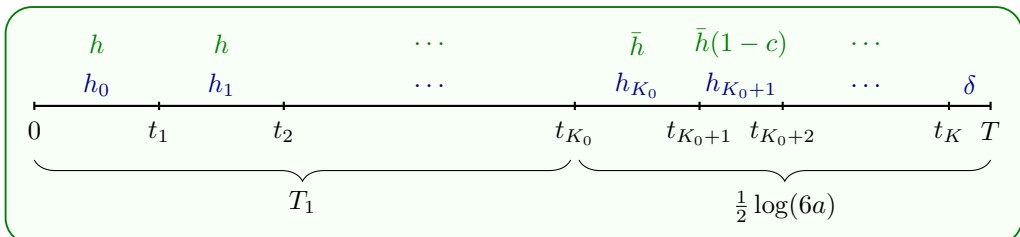

Figure 3: Notations corresponding to the discretization schedule.

We set $h_k = h$ for $k = 0, \ldots, K_0$. Then, (27), Lemma 11 and $1 - h_k m_k \leqslant 1 - h/3 \leqslant e^{-h/3}$ imply that

$$x_{K_0} \overset{①}{\leqslant} e^{-K_0 h/3} x_0 + \sum_{k=0}^{K_0-1} (1 - \tfrac{h}{3})^{K_0-k-1}(2h\varepsilon_k^b + \mathfrak{B}_k)$$

$$+ \left\{ \sum_{k=0}^{K_0-1} (1 - \tfrac{h}{3})^{2(K_0-k-1)}(2h\varepsilon_k^v + \mathfrak{V}_k)^2 \right\}^{1/2}$$

$$\overset{②}{\leqslant} e^{-K_0 h/3} x_0 + \max_{1\leqslant k < K_0}\left[\tfrac{3}{h}(2h\varepsilon_k^b + \mathfrak{B}_k)\right] + \max_{1\leqslant k < K_0}\left[\sqrt{\tfrac{1.7}{h}}(2h\varepsilon_k^v + \mathfrak{V}_k)\right]$$

$$\leqslant e^{-K_0 h/3} x_0 + \max_{1\leqslant k < K_0}\left[6\varepsilon_k^b + 3h^{-1}\mathfrak{B}_k\right] + \max_{1\leqslant k < K_0}\left[1.35h^{-1/2}(2h\varepsilon_k^v + \mathfrak{V}_k)\right], \tag{28}$$

where ① comes from applying Lemma 11 with $e^{A_k} = (1 - m_k h) \leqslant e^{-h/3}$, form which we get

$$e^{\bar{A}_j} = e^{A_0} \cdot e^{A_1} ... e^{A_j} = \prod_{l=0}^{j} (1 - m_l h_l) \leqslant \left(1 - \tfrac{h}{3}\right)^{j+1} \text{ and } e^{\bar{A}_{K_0-1}} = e^{-K_0 h/3}, \text{ and } ② \text{ uses the fact}$$

that $\sum_{k=0}^{K_0-1} (1 - \tfrac{h}{3})^{K_0-k-1} = \tfrac{3}{h}\left[1 - \left(1 - \tfrac{h}{3}\right)^{K_0}\right] \leqslant \tfrac{3}{h}$ and, similarly, $\sum_{k=0}^{K_0-1}(1 - \tfrac{h}{3})^{2(K_0-k-1)} = \tfrac{9}{h(6-h)}\left[1 - \left(1 - \tfrac{h}{3}\right)^{2K_0}\right] \leqslant \tfrac{9}{h(6-h)} \leqslant \tfrac{9}{5.3h} \leqslant \tfrac{1.7}{h}$ since $h \leqslant 0.7$.

The next lemma provides an upper bound for the bias and the variance of the discretization error.

**Lemma 13.** *Assume that for some $a > 0$ and $k \in \{0, \ldots, K\}$, $P^*$ satisfies Assumption 1 with $\varphi$ satisfying $\varphi(\sigma) \leqslant a$ for every $\sigma \in [\beta_{k+1}/\alpha_{k+1}; \beta_k/\alpha_k]$. Assume, in addition, that $\bar{m}_2 = (\mathbf{E}[\|\mathbf{X}\|^2]/D) \vee 1 < \infty$. Then, it holds that*

$$\mathfrak{B}_k \leqslant \tfrac{1}{2}\sqrt{\bar{m}_2 D}\, h_k^2, \tag{29}$$

$$\mathfrak{V}_k \leqslant \tfrac{1}{2}\sqrt{\bar{m}_2 D}\, h_k^2 + \tfrac{4\sqrt{2D}}{3}\, h_k^{3/2}\, \frac{(a\alpha_{k+1}^2) \vee \beta_{k+1}^2}{\beta_{k+1}^4}. \tag{30}$$

*If instead of $\varphi(\sigma) \leqslant a$, we have $\varphi(\sigma) \leqslant \bar{a}\sigma^2$ for some $\bar{a} \geqslant 1$, then (30) can be strengthened as follows*

$$\mathfrak{V}_k \leqslant \tfrac{1}{2}\sqrt{\bar{m}_2 D}\, h_k^2 + \tfrac{4\sqrt{2D}}{3}\, \bar{a}\, \frac{h_k^{3/2}}{\beta_{k+1}^2}. \tag{31}$$

*Finally, under the same condition, the error $\mathfrak{V}_K$ of the last iterate can be bounded by*

$$\mathfrak{V}_K \leqslant \tfrac{1}{2}\sqrt{\bar{m}_2 D}\, h_K^2 + \tfrac{9}{2}\, \bar{a}\sqrt{D\, h_K}. \tag{32}$$

### C.2 Proof of Theorem 2: Strongly log-concave convolved with a compactly supported distribution

We know that

$$\varphi(\sigma) = \frac{\sigma^2}{1 + m\sigma^2} + \frac{bM^2\sigma^4}{(1 + M\sigma^2)^2} \leqslant \left[\frac{1}{m} + b\right] \wedge \left[\sigma^2\left(1 + \frac{bM}{4}\right)\right].$$

Therefore, we can apply Lemma 10, Lemma 12 with $a = 1 \vee [(1/m) + b]$ as well as inequalities (29) and (31) of Lemma 13 with $\bar{a} = 1 + \tfrac{1}{4}bM$. In addition, to bound the last term in (31), we use the fact that

$$\frac{1}{\beta_{k+1}^2} = \frac{1}{1 - e^{2(t_{k+1}-T)}} \leqslant \frac{1}{1 - e^{-\log(6a)}} = \frac{6a}{6a - 1} \leqslant 1.2.$$

Together with (28), this leads to

$$\begin{aligned}
x_{K_0} &\leqslant e^{-K_0 h/3} x_0 + 6\varepsilon^b + 3h^{1/2}\varepsilon^v + \sqrt{\bar{m}_2 D}\left(\tfrac{3}{2} + 1.35 \times (\tfrac{1}{2} + \tfrac{4\sqrt{2}\,\bar{a}}{3} \times 1.2)\right)h \\
&\leqslant e^{-K_0 h/3} x_0 + 6\varepsilon^b + 3h^{1/2}\varepsilon^v + (5.3 + 0.6bM)h\sqrt{\bar{m}_2 D}.
\end{aligned} \tag{33}$$

On the time interval $[T - \tfrac{\log(6a)}{2}; T]$, we use the discretization obtained by geometrically decreasing stepsizes as previously proposed in the literature:

$$h_{K_0+j} = \tfrac{\log(6a)}{2}\, c\,(1-c)^j, \quad j = 0, \ldots, K - K_0 - 1,$$

where $c \leqslant 0.6/\log(6a)$. This implies, in particular, that $c \leqslant 0.6/\log 6 \leqslant 0.4$ and that $\bar{h} := \max_{k\in[K_0,K]} h_k \leqslant 0.3$. The constants $c$ and $K$ are chosen in such a way that $t_K = T - h_K$ for some small $h_K \leqslant \tfrac{\log(6a)}{2}$, and $t_{K+1} = T$. This means that

$$\begin{aligned}
T - h_K &= T - \tfrac{\log(6a)}{2} + \tfrac{\log(6a)}{2} c \sum_{j=0}^{K-K_0-1} (1-c)^j \\
&= T - \tfrac{\log(6a)}{2} + \tfrac{\log(6a)}{2}(1 - (1-c)^{K-K_0}).
\end{aligned}$$

This yields

$$(1-c)^{K-K_0} = \frac{2h_K}{\log(6a)} \qquad \text{and} \qquad K - K_0 = \frac{\log\log(6a) - \log(2h_K)}{-\log(1-c)}$$

$$\leqslant \frac{\log\log(6a) - \log(2h_K)}{c}.$$

For $k \geqslant K_0 + 1$, we will apply Lemma 10. To check that its conditions are fulfilled, note that

$$\frac{1+\alpha_k^2}{1-\alpha_k^2} + m_k \leqslant \frac{1+\alpha_k^2}{1-\alpha_k^2} + 1 + \frac{2\alpha_k^2}{1-\alpha_k^2} \leqslant \frac{4}{1-e^{2(t_k-T)}} \leqslant \frac{4}{1-e^{-1}} \left( \frac{1}{2(T-t_k)} \vee 1 \right).$$

This expression, multiplied by $h_k$, is less than 2 whenever $h_k \leqslant 0.3$. Indeed, on the one hand,

$$\frac{4h_k}{1-e^{-1}} \leqslant \frac{1.2}{1-e^{-1}} \leqslant 2.$$

On the other hand, for $k > K_0$,

$$t_k = T - \frac{\log(6a)}{2} + h_{K_0} + \ldots + h_{k-1} = T - \frac{\log(6a)}{2} + \frac{\log(6a)}{2} c \sum_{j=0}^{k-K_0-1} (1-c)^j$$

$$= T - \frac{\log(6a)}{2} + \frac{\log(6a)}{2} (1 - (1-c)^{k-K_0}) = T - c^{-1}h_k. \tag{34}$$

This implies that

$$\frac{4h_k}{2(1-e^{-1})(T-t_k)} = \frac{2c}{1-e^{-1}} < 2$$

since $c \leqslant 0.6$. In addition, taking $\sigma = \beta_k/\alpha_k$ and using the substitution $\beta_k^2 = 1 - \alpha_k^2$, we have

$$m_k \overset{\text{①}}{=} 1 + \frac{2\alpha_k^2}{1-\alpha_k^2} \left( 1 - \frac{\varphi(\beta_k/\alpha_k)}{1-\alpha_k^2} \right)$$

$$\overset{\text{②}}{=} 1 + \frac{2\alpha_k^2}{1-\alpha_k^2} \left( 1 - \frac{1}{\beta_k^2} \left[ \frac{\sigma^2}{1+m\sigma^2} + \frac{bM^2\sigma^4}{(1+M\sigma^2)^2} \right] \right)$$

$$\overset{\text{③}}{=} 1 + \frac{2\alpha_k^2}{\beta_k^2} \left( 1 - \frac{1}{\alpha_k^2 + m\beta_k^2} - \frac{\beta_k^2 \cdot bM^2}{(\alpha_k^2 + M\beta_k^2)^2} \right)$$

$$= 1 + \frac{2\alpha_k^2}{\beta_k^2} - \frac{2}{\beta_k^2(1+m\sigma^2)} - \frac{2bM^2\alpha_k^2}{(\alpha_k^2 + M(1-\alpha_k^2))^2},$$

where ① comes from the definition of $m_k$ from (25), ① is true for any $\varphi(\sigma)$ satisfying (6). Equality ③ comes from the fact that

$$\frac{1}{\beta_k^2} \cdot \frac{\sigma^2}{1+m\sigma^2} = \frac{1}{\beta_k^2} \cdot \frac{\beta_k^2/\alpha_k^2}{1+m\sigma^2} = \frac{1}{\alpha_k^2(1+m\sigma^2)} = \frac{1}{\alpha_k^2 + m\beta_k^2},$$

and

$$\frac{1}{\beta_k^2} \cdot \frac{bM^2\sigma^4}{(1+M\sigma^2)^2} = \frac{1}{\beta_k^2} \cdot \frac{bM^2 \cdot \beta_k^4/\alpha_k^4}{(1+M\sigma^2)^2} = \frac{\beta_k^2 bM^2}{\alpha_k^4(1+M\sigma^2)^2} = \frac{\beta_k^2 bM^2}{(\alpha_k^2 + M\beta_k^2)^2}.$$

Finally, noting that

$$1 + \frac{2\alpha_k^2}{\beta_k^2} - \frac{2}{\beta_k^2(1+m\sigma^2)} \geqslant 1 + \frac{2\alpha_k^2}{\beta_k^2} - \frac{2}{\beta_k^2} = -1,$$

for any $m, \sigma^2 \geqslant 0$, we arrive at

$$m_k \geqslant -1 - \frac{2bM^2\alpha_k^2}{(\alpha_k^2 + M(1-\alpha_k^2))^2}. \tag{35}$$

Therefore, (27) yields

$$x_{k+1}^2 \leqslant \left(e^{-m_k h_k} x_k + 2h_k \varepsilon_k^b + \mathfrak{B}_k\right)^2 + \left(2h_k \varepsilon_k^v + \mathfrak{V}_k\right)^2.$$

From this recursion and Lemma 11, using the notation $H(k) = -m_{K_0} h_{K_0} - \ldots - m_k h_k$, we infer that

$$x_{K+1} \leqslant e^{H(K)} \left[ x_{K_0} + \sum_{k=K_0}^{K} e^{-H(k)} (2h_k \varepsilon_k^b + \mathfrak{B}_k) + \left\{ \sum_{k=K_0}^{K} e^{-2H(k)} (2h_k \varepsilon_k^v + \mathfrak{V}_k)^2 \right\}^{1/2} \right].$$

Inequality (35) yields

$$H(K) - H(k) \leqslant \sum_{j=k+1}^{K} h_j + 2bM \sum_{j=k+1}^{K} \frac{M h_j \alpha_j^{-2}}{(1 + M(\alpha_j^{-2} - 1))^2}$$

$$\leqslant \frac{1}{2} \log(6a) - \sum_{j=K_0}^{k} h_j + 2bM \sum_{j=K_0}^{K} \frac{M h_j e^{2(T-t_j)}}{(1 + M(e^{2(T-t_j)} - 1))^2}.$$

Let us set $y_j = M(e^{2(T-t_j)} - 1)$. On the one hand, we have

$$H(K) - H(k) \leqslant \frac{1}{2} \log(6a) - \sum_{j=K_0}^{k} h_j + 2bM \sum_{j=K_0}^{K} \frac{h_j(y_j + M)}{(1 + y_j)^2}.$$

On the other hand, since $h_j \leqslant 0.3$, we have $e^{-2h_j} - 1 \leqslant -1.5\, h_j$. Therefore,

$$y_j - y_{j+1} = (y_j + M)(1 - e^{-2h_j}) \geqslant 1.5 h_j(y_j + M).$$

This implies that

$$H(K) - H(k) \leqslant \frac{1}{2} \log(6a) - \sum_{j=K_0}^{k} h_j + bM \sum_{j=K_0}^{K} \frac{4(y_j - y_{j+1})}{3(1 + y_j)^2}$$

$$\leqslant \frac{1}{2} \log(6a) - \sum_{j=K_0}^{k} h_j + bM \int_0^\infty \frac{4}{3(1 + t)^2}\, \mathrm{d}t$$

$$\leqslant \frac{1}{2} \log(6a) - \sum_{j=K_0}^{k} h_j + \frac{4bM}{3}.$$

Using the standard inequalities

$$\sum_{k=K_0}^{K} e^{-u(h_{K_0} + \ldots + h_k)} h_k \leqslant \int_0^\infty e^{-ux}\, \mathrm{d}x = 1/u, \qquad \forall u > 0, \tag{36}$$

we arrive at

$$x_{K+1} \leqslant \sqrt{6a}\, e^{\frac{4}{3} bM} \left( x_{K_0} + 2\varepsilon^b + \max_{K_0 < k < K} h_k^{-1} \mathfrak{B}_k + \max_{K_0 < k < K} \left[ \sqrt{h_k}\, \varepsilon_k^v + \frac{1}{2} h_k^{-1/2} \mathfrak{V}_k \right] \right)$$
$$+ \mathfrak{B}_K + 2h_K \varepsilon^v + \mathfrak{V}_K.$$

We apply then inequalities (29), (31) and (32) of Lemma 13 with $\bar{a} = 1 + \frac{1}{4} bM$. This leads to

$$x_{K+1} \leqslant \sqrt{6a} e^{\frac{4}{3} bM} \left( x_{K_0} + 2\varepsilon^b + \sqrt{\bar{h}}\, \varepsilon^v + \sqrt{D} \left[ \sqrt{\bar{m}_2}\, \bar{h} + \max_{k<K} \frac{\bar{a} h_k}{\beta_{k+1}^2} \right] \right) + 5\bar{a} \sqrt{\bar{m}_2 D} h_K. \tag{37}$$

The stepsizes $h_k$ of the geometric grid are much smaller than the noise levels $\beta_{k+1}^2$, as attested by the following inequality[7]

$$\frac{h_k}{\beta_{k+1}^2} = \frac{h_k}{1 - e^{2(t_{k+1} - T)}} \leqslant \frac{h_k}{1.2(T - t_{k+1}) \wedge 0.5} \leqslant \frac{5h_k}{6(T - t_{k+1})} \vee \frac{5h_k}{3}.$$

---

[7]We use the standard inequality $1 - e^{-x} \geqslant (1 - e^{-1})(x \wedge 1)$ for every $x > 0$.

It follows from (34) that $T - t_{k+1} = c^{-1}h_{k+1} = c^{-1}(1-c)h_k \geqslant \frac{2}{3}c^{-1}h_k$. Hence,

$$\frac{h_k}{\beta_{k+1}^2} \leqslant \frac{5c}{4} \vee \frac{5c\log(6a)}{6} = \frac{5c\log(6a)}{6} = \frac{\bar{h}}{3}. \tag{38}$$

Combining (37) and (38), we arrive at

$$x_{K+1} \leqslant \sqrt{6a}\, e^{\frac{4}{3}bM}\left(x_{K_0} + 2\varepsilon^b + \bar{h}^{1/2}\varepsilon^v + \frac{4}{3}\bar{a}\sqrt{\bar{m}_2 D}\,\bar{h}\right) + 5\bar{a}\sqrt{\bar{m}_2 D}h_K.$$

This inequality, in conjunction with (33), leads to

$$x_{K+1} \leqslant \sqrt{6a}\, e^{\frac{4}{3}bM}\left(x_0 + 8\varepsilon^b + 4h_{\max}^{1/2}\varepsilon^v + 6.7\bar{a}\sqrt{\bar{m}_2 D}\,h_{\max}\right) + 5\bar{a}\sqrt{\bar{m}_2 D}h_K,$$

where $h_{\max} = \max(h, \bar{h})$ is the maximal step size of the entire discretization grid, comprising the parts defined through arithmetic and geometric progressions. These step sizes should satisfy the inequalities

$$h \leqslant \frac{T - \frac{1}{2}\log(6a)}{K_0} \leqslant 0.7 \quad \bar{h} = \frac{c\log(6a)}{2} \leqslant \frac{\log(6a)\big(\log\log(6a) - \log(2h_K)\big)}{K - K_0} \leqslant 0.3.$$

To bound $x_0$, we note that

$$x_0^2 \leqslant \mathbf{E}[\|\alpha_T \boldsymbol{X} + \beta_T \boldsymbol{\xi} - \boldsymbol{\xi}\|^2] = \alpha_T^2 \|\boldsymbol{X}\|_{\mathbb{L}_2}^2 + (1-\beta_T)^2 D \leqslant 1.01\bar{m}_2^2 De^{-2T}.$$

as soon as $T \geqslant \log(6)$. Thus, $x_0 \leqslant 1.01\sqrt{\bar{m}_2 D}e^{-T}$. We set $T = \frac{1}{2}\log(6a) + T_1$ and $h_K = \delta = 0.5e^{-2T_1}$ and $K = 2K_0$. This leads to the claim of the theorem. Indeed, $h \leqslant 0.7$ translates into $K_0 \geqslant (10/7)T_1$ and $\bar{h} \leqslant 0.3$ translates into

$$K_0 \geqslant \frac{10\log(6a)\big(\log\log(6a) + 2T_1\big)}{3}$$

which is satisfied when $K_0 \geqslant 7T_1\log(6a) + 4\log(6a)\log\log(6a)$. Finally, notice that $h \leqslant T_1/K_0$ and

$$\bar{h} \leqslant \frac{\log(6a)\big(\log\log(6a) + 2T_1\big)}{K_0}.$$

These inequalities yield the claimed upper bound on $h_{\max}$.

## C.3   Proof of Theorem 3: Semi log-concave and compactly supported distribution on a subspace

For $P^*$ satisfying Assumption 1 with the function

$$\varphi(\sigma) = b \wedge \frac{\sigma^2}{(1 - M\sigma^2)_+}. \tag{39}$$

we can apply Lemma 12 with $a = b \vee 1$ and Lemma 13 with $\bar{a} = bM + 1$. Similarly to Appendix C.2, the application of Lemma 10 and Lemma 12 yields

$$\begin{aligned}
x_{K_0} &\leqslant e^{-K_0 h/3}x_0 + 6\varepsilon^b + 3h^{1/2}\varepsilon^v + \sqrt{\bar{m}_2 D}\left(\tfrac{3}{2} + 1.35 \times (\tfrac{1}{2} + \tfrac{4\sqrt{2}\,\bar{a}}{3} \times 1.2)\right)h \\
&\leqslant e^{-K_0 h/3}x_0 + 6\varepsilon^b + 3h^{1/2}\varepsilon^v + (2.2 + 3.1\bar{a})h\sqrt{\bar{m}_2 D}.
\end{aligned} \tag{40}$$

We again use the discretization with geometrically decreasing stepsize on the interval $[T - \frac{\log(6a)}{2}; T]$:

$$h_{K_0+j} = \tfrac{\log(6a)}{2}\, c\,(1-c)^j, \quad j = 0, \ldots, K - K_0 - 1,$$

where $c \leqslant 0.6/\log(6a)$. Following the discussion in Appendix C.2, we have that

$$K - K_0 \leqslant \frac{\log\log(6a) - \log(2h_K)}{c},$$

and, for $k > K_0$

$$h_k\left(\frac{1+\alpha_k^2}{1-\alpha_k^2} + m_k\right) \leqslant 2 \quad \text{and} \quad t_k = T - c^{-1}h_k.$$

Combined with (39), we get

$$m_k \geqslant 1 - 2\bar{a}.$$

Hence, 27 yields

$$x_{k+1}^2 \leqslant \left( e^{(2\bar{a}-1)h_k} x_k + 2h_k \varepsilon_k^b + \mathfrak{B}_k \right)^2 + \left( 2h_k \varepsilon_k^v + \mathfrak{V}_k \right)^2.$$

We denote $H_k = (2\bar{a} - 1) \sum_{i=K_0}^{k} h_k$. We note that $H_K \leqslant \frac{2\bar{a}-1}{2} \log(6a)$. Lemma 11 states:

$$x_{K+1} \leqslant e^{H_K} \left[ x_{K_0} + \sum_{k=K_0}^{K} e^{-H_k} (2h_k \varepsilon_k^b + \mathfrak{B}_k) + \left\{ \sum_{k=K_0}^{K} e^{-2H_k} (2h_k \varepsilon_k^v + \mathfrak{V}_k)^2 \right\}^{1/2} \right].$$

As $2\bar{a} - 1 = 2bM + 1$ which is strictly positive, we may apply (36) which results in:

$$x_{K+1} \leqslant \sqrt{6a}\, e^{2\bar{a}-1} \left( x_{K_0} + \frac{2\varepsilon^b + \max_{k<K} h_k^{-1} \mathfrak{B}_k}{2\bar{a}-1} + \frac{1}{\sqrt{2\bar{a}-1}} \max_{k<K} \left\{ \sqrt{h_k}\, \varepsilon_k^v + \frac{\mathfrak{V}_k}{2h_k^{1/2}} \right\} \right)$$
$$+ \mathfrak{B}_K + 2h_K \varepsilon^v + \mathfrak{V}_K.$$

We apply then inequalities (29), (31) and (32) of Lemma 13, which leads to

$$x_{K+1} \leqslant \sqrt{6a}\, e^{(2\bar{a}-1)} \left( x_{K_0} + \frac{2\varepsilon^b}{2\bar{a}-1} + \frac{\sqrt{h}\, \varepsilon^v}{\sqrt{2\bar{a}-1}} + \frac{\sqrt{D}}{\sqrt{2\bar{a}-1}} \left[ \sqrt{\bar{m}_2}\, \bar{h} + \max_{k<K} \frac{\bar{a} h_k}{\beta_{k+1}^2} \right] \right) + 5\bar{a} \sqrt{\bar{m}_2 D h_K}.$$

The above inequality with (38) yields:

$$x_{K+1} \leqslant \sqrt{6a}\, e^{(2\bar{a}-1)} \left( x_{K_0} + \frac{2\varepsilon^b}{2\bar{a}-1} + \sqrt{\frac{\bar{h}}{2\bar{a}-1}}\, \varepsilon^v + \frac{4\bar{a}\bar{h}}{3} \sqrt{\frac{D\bar{m}_2}{2\bar{a}-1}} \right) + 5\bar{a} \sqrt{\bar{m}_2 D h_K}. \quad (41)$$

Combining (41) with (40) and noting that $(2\bar{a} - 1) \geqslant 1$, we get:

$$x_{K+1} \leqslant \sqrt{6a}\, e^{2\bar{a}-1} \left( x_0 + 8\varepsilon^b + 4h_{\max}^{1/2} \varepsilon^v + 6.7\bar{a} \sqrt{\bar{m}_2 D}\, h_{\max} \right) + 5\bar{a} \sqrt{\bar{m}_2 D h_K},$$

Following the discussion of Appendix C.2, we complete the proof by showing that:

$$x_{K+1} \leqslant e^{2\bar{a}-1} \left\{ 2e^{-T_1} + 7\sqrt{6a}\, h_{\max} + 4\sqrt{6a} \left( 2\varepsilon_{\text{score}}^b + h_{\max}^{1/2} \varepsilon_{\text{score}}^v \right) \right\} \sqrt{D}.$$

# D   Proofs of lemmas used in the proofs of main theorems

We collect in this section the proofs of the building blocks of our main results.

## D.1   Proof of Lemma 10: the origin of the contraction/expansion

Since $s$ is continuously differentiable, by the mean-value identity, we have

$$
\begin{aligned}
\boldsymbol{U}_k &= \boldsymbol{s}(T - t_k, \boldsymbol{Y}_{t_k}) - \boldsymbol{s}(T - t_k, \boldsymbol{Z}_k) \\
&= \int_0^1 \mathrm{D}\boldsymbol{s}\left( T - t_k, \boldsymbol{Z}_k + \theta(\boldsymbol{Y}_{t_k} - \boldsymbol{Z}_k) \right) (\boldsymbol{Y}_{t_k} - \boldsymbol{Z}_k)\, \mathrm{d}\theta := \int_0^1 \mathbf{M}_k(\theta) \boldsymbol{\Delta}_k\, \mathrm{d}\theta,
\end{aligned}
$$

where

$$\mathbf{M}_k(\theta) = \mathrm{D}\boldsymbol{s}(T - t_k, \boldsymbol{Z}_k + \theta \boldsymbol{\Delta}_k) = \nabla^2 \log \pi(T - t_k, \boldsymbol{Z}_k + \theta \boldsymbol{\Delta}_k).$$

The matrix $\mathbf{M}_k$ is symmetric, and according to Proposition 1, all its eigenvalues satisfy

$$\lambda_{\min} := -\frac{1}{1 - \alpha_k^2} \leqslant \lambda_j(\mathbf{M}_k(\theta)) \leqslant -\frac{1}{1 - \alpha_k^2} \left( 1 - \frac{\alpha_k^2 \varphi(\beta_k/\alpha_k)}{1 - \alpha_k^2} \right) =: \lambda_{\max}.$$

Since $\boldsymbol{U}_k = \int_0^1 \mathbf{M}_k(\theta) \boldsymbol{\Delta}_k \mathrm{d}\theta$, we get

$$(1 + h_k) \boldsymbol{\Delta}_k + 2h_k \boldsymbol{U}_k = \int_0^1 \left[ (1 + h_k) \mathbf{I}_D + 2h_k \mathbf{M}_k(\theta) \right] \boldsymbol{\Delta}_k \mathrm{d}\theta,$$

and

$$\|(1 + h_k)\,\mathbf{I}_D + 2h_k\mathbf{M}_k(\theta)\| = \max_{\lambda \in [\lambda_{\min}, \lambda_{\max}]} |1 + h_k + 2h_k\lambda|.$$

We assume that $h_k$ is chosen so that

$$\frac{2h_k}{1 - \alpha_k^2} - (1 + h_k) \leqslant (1 + h_k) - \frac{2h_k}{1 - \alpha_k^2}\Big(1 - \frac{\alpha_k^2\varphi(\beta_k/\alpha_k)}{1 - \alpha_k^2}\Big). \tag{42}$$

This is equivalent to

$$\frac{h_k}{1 - \alpha_k^2}\Big(2 - \frac{\alpha_k^2\varphi(\beta_k/\alpha_k)}{1 - \alpha_k^2}\Big) \leqslant (1 + h_k).$$

Regrouping the terms, we get

$$\frac{h_k}{1 - \alpha_k^2}\Big(1 + \alpha_k^2 - \frac{\alpha_k^2\varphi(\beta_k/\alpha_k)}{1 - \alpha_k^2}\Big) \leqslant 1.$$

This inequality can be checked to be the same as (25). Hence, (42) is indeed satisfied and, therefore,

$$\|(1 + h_k)\,\mathbf{I}_D + 2h_k\mathbf{M}_k(\theta)\| \leqslant 1 + h_k - \frac{2h_k}{1 - \alpha_k^2}\Big(1 - \frac{\alpha_k^2\varphi(\beta_k/\alpha_k)}{1 - \alpha_k^2}\Big).$$

Therefore, by the triangle (Minkowski) inequality, we have

$$
\begin{aligned}
\|(1 + h_k)\boldsymbol{\Delta}_k + 2h_k\boldsymbol{U}_k\|_{\mathbb{L}_2} &\leqslant \int_0^1 \big\|\big((1 + h_k)\,\mathbf{I}_D + 2h_k\mathbf{M}_k(\theta)\big)\boldsymbol{\Delta}_k\big\|_{\mathbb{L}_2}\,\mathrm{d}\theta \\
&\leqslant \Big\{1 + h_k - \frac{2h_k}{1 - \alpha_k^2}\Big(1 - \frac{\alpha_k^2\varphi(\beta_k/\alpha_k)}{1 - \alpha_k^2}\Big)\Big\}\|\boldsymbol{\Delta}_k\|_{\mathbb{L}_2} \\
&= \Big\{1 - \frac{2h_k}{1 - \alpha_k^2}\Big(\frac{1 + \alpha_k^2}{2} - \frac{\alpha_k^2\varphi(\beta_k/\alpha_k)}{1 - \alpha_k^2}\Big)\Big\}\|\boldsymbol{\Delta}_k\|_{\mathbb{L}_2} \\
&= (1 - m_kh_k)\|\boldsymbol{\Delta}_k\|_{\mathbb{L}_2}.
\end{aligned}
$$

This completes the proof of Lemma 10.

## D.2 Proof of Lemma 12: strength of the deflation in the contracting regime

First, notice that $\alpha_k$ being an increasing function of $t_k$, we have

$$\alpha_k^2 = e^{2(t_k - T)} \leqslant \exp(-\log(6a)) = 1/(6a).$$

Second, since we assumed $\varphi(\beta_k/\alpha_k) \leqslant a$, we have

$$m_k \geqslant 1 + \frac{2\alpha_k^2}{1 - \alpha_k^2}\Big(1 - \frac{a}{1 - \alpha_k^2}\Big) = \frac{1 - 2a\alpha_k^2 - \alpha_k^4}{(1 - \alpha_k^2)^2}.$$

Since $\alpha_k^2 \leqslant 1/(6a)$ and we assumed $a \geqslant 1$, we have $\alpha_k^2 \leqslant a$. Combining these inequalities with $0 \leqslant 1 - \alpha_k^2 \leqslant 1$, we arrive at

$$m_k \geqslant \frac{1 - (1/3) - \alpha_k^2/(6a)}{(1 - \alpha_k^2)^2} \geqslant \frac{1 - (1/3) - a/(6a)}{(1 - \alpha_k^2)^2} = \frac{1}{2(1 - \alpha_k^2)^2} \geqslant \frac{1}{3}.$$

For the second inequality of the lemma, it suffices to notice that $\varphi(\sigma) \geqslant 0$ and $a \geqslant 1$ imply that

$$\frac{1 + \alpha_k^2}{1 - \alpha_k^2} + m_k \leqslant \frac{1 + \alpha_k^2}{1 - \alpha_k^2} + 1 + \frac{2\alpha_k^2}{1 - \alpha_k^2} = 1 + \frac{1 + 3\alpha_k^2}{1 - \alpha_k^2} \leqslant 1 + \frac{6a + 3}{6a - 1} \leqslant 2.8.$$

Combining with the condition $h_k \leqslant 0.7$, this yields $h_k\big(\frac{1 + \alpha_k^2}{1 - \alpha_k^2} + m_k\big) \leqslant 2$ and completes the proof of the lemma.

### D.3 Proof of Lemma 13: assessing the increments of the drift

Let $\boldsymbol{b}_t = \boldsymbol{Y}_t + 2\boldsymbol{s}(T - t, \boldsymbol{Y}_t)$. To prove the first inequality, we recall that $\boldsymbol{s}(T - t, \boldsymbol{y}) = (\alpha_{T-t}\mathbf{E}[\,\boldsymbol{X}_0 \,|\, \boldsymbol{Y}_t = \boldsymbol{y}] - \boldsymbol{y})/\beta_{T-t}^2$. Therefore,

$$\boldsymbol{b}_t = \frac{2\alpha}{\beta^2}\mathbf{E}[\,\boldsymbol{X}_0 \,|\, \boldsymbol{Y}_t] + \boldsymbol{Y}_t\Big(1 - \frac{2}{\beta^2}\Big).$$

In addition, $\boldsymbol{Y}_t = \alpha\boldsymbol{X}_0 + \beta\boldsymbol{\xi}$ with $\boldsymbol{\xi} \perp\!\!\!\perp \boldsymbol{X}_0$ and $\boldsymbol{\xi} \sim \mathcal{N}_D(0, \mathbf{I}_D)$. It holds that

$$\mathbf{E}[\|\boldsymbol{Y}_t\|^2] = \alpha^2\mathbf{E}[\|\boldsymbol{X}_0\|^2] + \beta^2\mathbf{E}[\|\boldsymbol{\xi}\|^2] = \alpha^2\bar{m}_2 D + \beta^2 D, \tag{43}$$

$$\mathbf{E}[\boldsymbol{X}_0^\mathsf{T}\boldsymbol{Y}_t] = \alpha\mathbf{E}[\|\boldsymbol{X}_0\|^2] + \beta\mathbf{E}[\boldsymbol{X}_0^\mathsf{T}\boldsymbol{\xi}] = \alpha\bar{m}_2 D,$$

since $\boldsymbol{\xi}$ is independent of $\boldsymbol{X}_0$ and has zero mean.

Let us use the "local notation" $\bar{\boldsymbol{s}}(t, \boldsymbol{y}) = \boldsymbol{s}(t, \boldsymbol{y}) + \boldsymbol{y}$ as well as $\mathbf{H}(t, \boldsymbol{y}) = \mathrm{D}\boldsymbol{s}(t, \boldsymbol{y})$. According to [CDS25, Prop. 2], it holds that

$$\mathrm{d}\bar{\boldsymbol{s}}(T - t, \boldsymbol{Y}_t) = \bar{\boldsymbol{s}}(T - t, \boldsymbol{Y}_t)\,\mathrm{d}t + \sqrt{2}\,\mathrm{D}\bar{\boldsymbol{s}}(T - t, \boldsymbol{Y}_t)\,\mathrm{d}\tilde{\boldsymbol{B}}_t.$$

Since $2\bar{\boldsymbol{s}}(T - t, \boldsymbol{Y}_t) = \boldsymbol{b}_t + \boldsymbol{Y}_t$, and $\mathrm{D}\bar{\boldsymbol{s}}(T - t, \boldsymbol{Y}_t) = \mathbf{H}(T - t, \boldsymbol{Y}_t) + \mathbf{I}_D$ we get

$$\begin{aligned}
\mathrm{d}\boldsymbol{b}_t &= -\mathrm{d}\boldsymbol{Y}_t + 2\mathrm{d}\bar{\boldsymbol{s}}(T - t, \boldsymbol{Y}_t) \\
&= -\boldsymbol{b}_t\,\mathrm{d}t - \sqrt{2}\,\mathrm{d}\tilde{\boldsymbol{B}}_t + (\boldsymbol{b}_t + \boldsymbol{Y}_t)\,\mathrm{d}t + 2\sqrt{2}\,(\mathbf{H}(T - t, \boldsymbol{Y}_t) + \mathbf{I}_D)\,\mathrm{d}\tilde{\boldsymbol{B}}_t \\
&= \boldsymbol{Y}_t\,\mathrm{d}t + \sqrt{2}\,(2\mathbf{H}(T - t, \boldsymbol{Y}_t) + \mathbf{I}_D)\,\mathrm{d}\tilde{\boldsymbol{B}}_t.
\end{aligned}$$

Since $\tilde{\boldsymbol{B}}_t - \tilde{\boldsymbol{B}}_{t_k}$ is independent of the $\sigma$-algebra $\mathcal{F}_k$, we get

$$\mathbf{E}[\,\boldsymbol{b}_t - \boldsymbol{b}_{t_k} \,|\, \mathcal{F}_k] = \int_{t_k}^t \mathbf{E}[\,\boldsymbol{Y}_u \,|\, \mathcal{F}_k]\,\mathrm{d}u$$

and, therefore,

$$\begin{aligned}
\|\mathbf{E}[\,\boldsymbol{b}_t - \boldsymbol{b}_{t_k} \,|\, \mathcal{F}_k]\|_{\mathbb{L}_2} &\leqslant \int_{t_k}^t \|\mathbf{E}[\,\boldsymbol{Y}_u \,|\, \mathcal{F}_k]\|_{\mathbb{L}_2}\mathrm{d}u \leqslant \int_{t_k}^t \|\boldsymbol{Y}_u\|_{\mathbb{L}_2}\mathrm{d}u \\
&\leqslant \int_{t_k}^t \sqrt{D(e^{-2(T-u)}\bar{m}_2 + (1 - e^{-2(T-u)}))}\,\mathrm{d}u \\
&\leqslant \sqrt{\bar{m}_2 D}\,(t - t_k).
\end{aligned}$$

The definition of $\boldsymbol{V}_k$ given in (20) implies that $\boldsymbol{V}_k = \int_{t_k}^{t_{k+1}}(\boldsymbol{b}_t - \boldsymbol{b}_{t_k})\,\mathrm{d}t$. This leads to

$$\begin{aligned}
\|\mathbf{E}[\,\boldsymbol{V}_k \,|\, \mathcal{F}_k]\| &\leqslant \int_{t_k}^{t_{k+1}} \|\mathbf{E}[\,\boldsymbol{b}_t - \boldsymbol{b}_{t_k} \,|\, \mathcal{F}_k]\|_{\mathbb{L}_2}\,\mathrm{d}t \\
&\leqslant \sqrt{\bar{m}_2 D}\int_{t_k}^{t_{k+1}}(t - t_k)\,\mathrm{d}t = \tfrac{1}{2}\sqrt{\bar{m}_2 D}\,h_k^2.
\end{aligned}$$

This yields the claim of (29).

We prove now (30). The definition of $\boldsymbol{b}_t = \boldsymbol{Y}_t + 2\boldsymbol{s}(T - t, \boldsymbol{Y}_t)$ leads to

$$\begin{aligned}
\big\|\boldsymbol{b}_t &- \boldsymbol{b}_{t_k} - \mathbf{E}[\,\boldsymbol{b}_t - \boldsymbol{b}_{t_k} \,|\, \mathcal{F}_k]\big\|_{\mathbb{L}_2} \\
&= \bigg\|\int_{t_k}^t (\boldsymbol{Y}_u - \mathbf{E}[\,\boldsymbol{Y}_u \,|\, \mathcal{F}_k])\,\mathrm{d}u + \int_{t_k}^t \sqrt{2}\big(2\mathbf{H}(u) + \mathbf{I}_D\big)\,\mathrm{d}\tilde{\boldsymbol{B}}_u\bigg\|_{\mathbb{L}_2} \\
&\leqslant \int_{t_k}^t \big\|\boldsymbol{Y}_u - \mathbf{E}[\,\boldsymbol{Y}_u \,|\, \mathcal{F}_k]\big\|_{\mathbb{L}_2}\,\mathrm{d}u + \bigg\|\int_{t_k}^t \sqrt{2}\big(2\mathbf{H}(u) + \mathbf{I}_D\big)\,\mathrm{d}\tilde{\boldsymbol{B}}_u\bigg\|_{\mathbb{L}_2}. \tag{44}
\end{aligned}$$

On the one hand, in view of the law of total variance, we have $\big\|\boldsymbol{Y}_u - \mathbf{E}[\,\boldsymbol{Y}_u \,|\, \mathcal{F}_k]\big\|_{\mathbb{L}_2} \leqslant \big\|\boldsymbol{Y}_u\big\|_{\mathbb{L}_2}$. Therefore, using (43), we get

$$\int_{t_k}^t \big\|\boldsymbol{Y}_u - \mathbf{E}[\,\boldsymbol{Y}_u \,|\, \mathcal{F}_k]\big\|_{\mathbb{L}_2}\,\mathrm{d}u \leqslant \int_{t_k}^t \sqrt{\bar{m}_2 D}\,\mathrm{d}u = \sqrt{\bar{m}_2 D}\,(t - t_k). \tag{45}$$

On the other hand, the properties of the stochastic integral imply that

$$\left\|\int_{t_k}^t \sqrt{2}\bigl(2\mathbf{H}(u)+\mathbf{I}_D\bigr)\,\mathrm{d}\tilde{\boldsymbol{B}}_u\right\|_{\mathbb{L}_2}^2 = 2\int_{t_k}^t \mathbf{E}\bigl[\bigl\|2\mathbf{H}(u)+\mathbf{I}_D\bigr\|_F^2\bigr]\,\mathrm{d}u. \tag{46}$$

Combining he definition of $\boldsymbol{V}_k$ given in (20) with (44), (45) and (46), we get

$$\|\boldsymbol{V}_k - \mathbf{E}[\,\boldsymbol{V}_k\,|\,\mathcal{F}_k\,]\|_{\mathbb{L}_2} = \int_{t_k}^{t_{k+1}} \bigl\|\boldsymbol{b}_t - \boldsymbol{b}_{t_k} - \mathbf{E}[\,\boldsymbol{b}_t - \boldsymbol{b}_{t_k}\,|\,\mathcal{F}_k\,]\bigr\|_{\mathbb{L}_2}\mathrm{d}t$$

$$\leqslant \tfrac{1}{2}\sqrt{\bar{m}_2 D}\,h_k^2 + \int_{t_k}^{t_{k+1}} \left\{2\int_{t_k}^t \mathbf{E}\bigl[\bigl\|2\mathbf{H}(u)+\mathbf{I}_D\bigr\|_F^2\bigr]\,\mathrm{d}u\right\}^{1/2}\mathrm{d}t. \tag{47}$$

The integral in (46) can be bounded from above using Proposition 1 and various assumptions of the function $\varphi$ from Assumption 1. Indeed, denoting $\sigma_{T-u} = \beta_{T-u}/\alpha_{T-u}$, we have $\mathbf{H}(u) \preccurlyeq \beta_{T-u}^{-2}(\varphi(\sigma_{T-u})\sigma_{T-u}^{-2}-1)\,\mathbf{I}_D$. Since, in addition $\mathbf{H}(u) \succcurlyeq -\beta_{T-u}^{-2}\mathbf{I}_D$, we get

$$0 \preccurlyeq (2\mathbf{H}(u)+\mathbf{I}_D)^2 \preccurlyeq 4\frac{[\varphi(\sigma_{T-u})/\sigma_{T-u}^2]^2 \vee 1}{\beta_{T-u}^4}\,\mathbf{I}_D. \tag{48}$$

If we assume that $\varphi(\sigma_{T-u}) \leqslant a$, we arrive at

$$\left\{2\int_{t_k}^t \mathbf{E}\bigl[\bigl\|2\mathbf{H}(u)+\mathbf{I}_D\bigr\|_F^2\bigr]\,\mathrm{d}u\right\}^{1/2} \leqslant 2\sqrt{2D(t-t_k)}\,\frac{(a\alpha_{T-t}^2)\vee\beta_{T-t}^2}{\beta_{T-t}^4}.$$

In view of (47), this yields

$$\|\boldsymbol{V}_k - \mathbf{E}(\boldsymbol{V}_k\,|\,\mathcal{F}_k)\|_{\mathbb{L}_2} \leqslant \tfrac{1}{2}\sqrt{\bar{m}_2 D}\,h_k^2 + \tfrac{4\sqrt{2D}}{3}\,h_k^{3/2}\frac{(a\alpha_{T-t_{k+1}}^2)\vee\beta_{T-t_{k+1}}^2}{\beta_{T-t_{k+1}}^4}.$$

This completes the proof of the second claim of the lemma.

If instead of the assumption $\varphi(\sigma) \leqslant a$, we use the assumption $\varphi(\sigma) \leqslant \bar{a}\sigma^2$ with $\bar{a} \geqslant 1$, inequality (48), the fact that $u \mapsto \beta_{T-u}$ is decreasing, and inequality (47) imply that

$$\|\boldsymbol{V}_k - \mathbf{E}(\boldsymbol{V}_k\,|\,\mathcal{F}_k)\|_{\mathbb{L}_2} \leqslant \tfrac{1}{2}\sqrt{\bar{m}_2 D}\,h_k^2 + \tfrac{4\sqrt{2D}}{3}\,h_k^{3/2}\frac{\bar{a}}{\beta_{T-t_{k+1}}^2}.$$

For the last claim, we use (47) and (48) as follows

$$\int_{t_K}^T \left\{\int_{t_K}^t \mathbf{E}\bigl[\bigl\|2\mathbf{H}(u)+\mathbf{I}_D\bigr\|_F^2\bigr]\,\mathrm{d}u\right\}^{1/2}\mathrm{d}t \leqslant \sqrt{D}\int_{t_K}^T\left\{\int_{t_K}^t\frac{4\bar{a}^2}{(1-e^{-2(T-u)})^2}\,\mathrm{d}u\right\}^{1/2}\mathrm{d}t$$

$$\leqslant \sqrt{D}\int_0^{T-t_K}\left\{\int_t^{T-t_K}\frac{4\bar{a}^2}{(1-e^{-2u})^2}\,\mathrm{d}u\right\}^{1/2}\mathrm{d}t$$

$$\leqslant \sqrt{D}\int_0^{T-t_K}\left\{\int_t^{T-t_K}\frac{4\bar{a}^2}{u^2}\,\mathrm{d}u\right\}^{1/2}\mathrm{d}t$$

$$= \sqrt{D}\int_0^{T-t_K}\left\{\frac{4\bar{a}^2(T-t_k-t)}{t(T-t_K)}\right\}^{1/2}\mathrm{d}t$$

$$= \pi\bar{a}\sqrt{D(T-t_K)}.$$

Thus, from (47), we infer that

$$\|\boldsymbol{V}_K - \mathbf{E}(\boldsymbol{V}_K\,|\,\mathcal{F}_K)\|_{\mathbb{L}_2} \leqslant \tfrac{1}{2}\sqrt{\bar{m}_2 D}\,h_K^2 + \tfrac{9}{2}\bar{a}\sqrt{D h_K}.$$

This completes the proof.

# E  Numerical Experiments

Our experiments follow the standard DDPM sampling procedure as described in the original DDPM paper by [HJA20], specifically the pseudocode presented in their Algorithm 2.

### E.1 Implementation Details

For clarity, we re-state their algorithm below.

---

**Algorithm 3** DDPM Sampling [HJA20]

---

1: $\boldsymbol{x}_T \sim \mathcal{N}(\mathbf{0}, \mathbf{I})$
2: **for** $t = T$ **to** 1 **do**
3:     $\boldsymbol{z} \sim \mathcal{N}(\mathbf{0}, \mathbf{I})$ if $t > 1$, else $\boldsymbol{z} = \mathbf{0}$
4:     $\boldsymbol{\mu_\theta}(\boldsymbol{x}_t, t) = \frac{1}{\sqrt{\alpha_t}}\left(\boldsymbol{x}_t - \frac{1-\alpha_t}{\sqrt{1-\bar{\alpha}_t}}\boldsymbol{\epsilon}_\theta(\boldsymbol{x}_t, t)\right)$
5:     $\boldsymbol{x}_{t-1} = \boldsymbol{\mu_\theta}(\boldsymbol{x}_t, t) + \sigma_t \boldsymbol{z}$
6: **end for**
7: **return** $\boldsymbol{x}_0$

---

To better explain the correspondence between notation used in our paper and that of [HJA20], we provide the following table:

| Notation in [HJA20] | Our notation |
|---|---|
| $\boldsymbol{x}_T, \ldots, \boldsymbol{x}_0$ | $\boldsymbol{Z}_0, \ldots, \boldsymbol{Z}_{K+1}$ |
| $\boldsymbol{z}$ | $\boldsymbol{\xi}_{k+1}$ |
| $\sigma_t$ | $\sqrt{2h_k}$ |
| $\alpha_t$ | $(1+h_k)^{-2} \approx e^{-2h_k} \approx 1 - 2h_k$ |
| $\bar{\alpha}_t$ | $\prod_{j=0}^{k}(1+h_k)^{-2} \approx e^{-2t_{K+1}}$ |
| $\dfrac{\boldsymbol{\epsilon}_\theta(\boldsymbol{x}_t, t)}{\sqrt{1-\bar{\alpha}_t}}$ | $-2\widetilde{\boldsymbol{s}}(T - t_k, \boldsymbol{Z}_k)$ |

To evaluate the robustness of the generative process under perturbed score estimates, we had to isolate the score estimation component within the sampling loop. In the formulation of [HJA20], this corresponds to the rescaled neural network output $-0.5\boldsymbol{\epsilon}_\theta(\boldsymbol{x}_t, t)/\sqrt{1-\bar{\alpha}_t}$. In our experiments, we added various forms of noise (Gaussian, Uniform, Laplace, and Student's-$t$) directly to this term, simulating inaccurate or noisy score predictions. This modification allows us to assess the impact of score perturbations on the quality of generated samples, both visually and quantitatively.

We know that in our formulation of the problem, the conditional expectation of the next state given that the current state is $\boldsymbol{x}$ is given by $\boldsymbol{\mu_\theta}(\boldsymbol{x}, t) = (1+h)\boldsymbol{x} + 2\boldsymbol{s}(t, \boldsymbol{x})h$. Therefore, adding $\boldsymbol{\zeta}$ to $\boldsymbol{s}(t, \boldsymbol{x})$ implies adding $2h\boldsymbol{\zeta}$ to $\boldsymbol{\mu_\theta}(\boldsymbol{x}_t, t)$, and thus adding $\dfrac{\sqrt{\alpha_t(1-\bar{\alpha}_t)}}{1-\alpha_t} \times 2h_k\boldsymbol{\zeta} \approx 2\sqrt{1-\bar{\alpha}_t}\,\boldsymbol{\zeta}$ to $\boldsymbol{\epsilon}_\theta(\boldsymbol{x}, t)$.

### E.2 Additional Figures

**Qualitative results.** Figure 6, Figure 7 and Figure 8 extend the main-paper image grids. For each dataset (CIFAR-10, CelebA-HQ, and LSUN-Churches) we display samples generated with Gaussian, Laplace, and Student's-$t$ score noise at two strengths, $\sigma = 0.5$ or $\sigma = 1$ (moderate) and $\sigma = 2$ (severe). Rows share the same latent seed as the baseline to enable direct visual comparison.

**Quantitative trends.** Figure 4 tracks FID on the CIFAR-10 dataset as we truncate the 1 000-step DDPM schedule at $\{250, 500, 750, 1000\}$ steps for the *clean* score and the i.i.d. $\mathcal{N}(\mathbf{0}, \mathbf{I}_D)$ noise contaminated score. We observe that performance increases at a similar rate with the number of steps for both clean and noisy score estimates.

Additionally, Figure 5 illustrates the "deterioration" of three distinct pictures for each of the different models (datasets) that we have — each starting with a fixed random noise, generating the corresponding image after 1000 diffusion steps with the noise contaminated score, as described before, parametrized by different $\sigma$. We observe that datasets with higher-resolution images and, respectively, deeper noise (alternatively, score) predicting neural networks exhibit higher deterioration than those with low-resolution images.

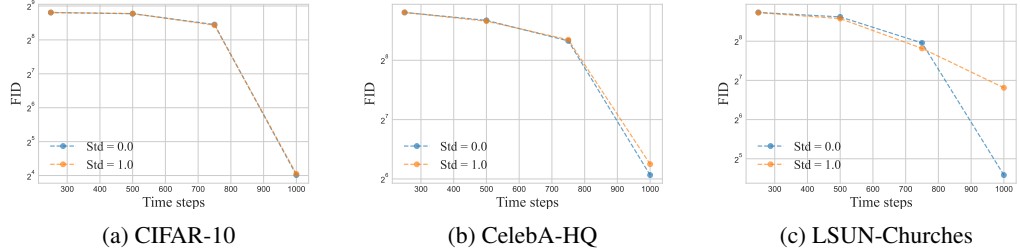

(a) CIFAR-10  (b) CelebA-HQ  (c) LSUN-Churches

Figure 4: FID as a function of time steps. Blue: standard DDPM inference. Orange: same sampler with i.i.d. $\mathcal{N}(\mathbf{0}, \mathbf{I}_D)$ noise added to the score at each step.

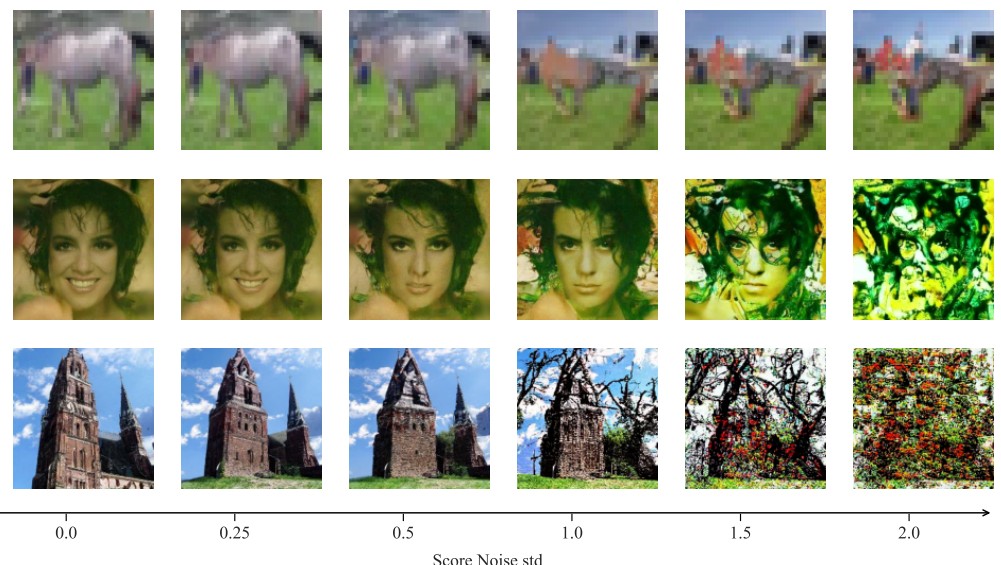

Figure 5: A single example of CIFAR-10 (top), CelebA-HQ (middle) and LSUN-Churches (bottom) generated data, respectively, over different standard deviations.

### E.3 Computational Resources

This project was provided with computer and storage resources by GENCI at IDRIS thanks to the grant 2025-AD011016491 on the supercomputer Jean Zay's A100 partition.

Some of the experiments were run on two additional GPU nodes: one with AMD EPYC 7V12 64-Core Processor, 1TB of RAM, and with 8xA100 40GB VRAM version NVIDIA GPUs. The other one with AMD EPYC 9005 192-Core Processor, 0.5TB of RAM, and with 2xH100 NVIDIA GPUs.

Sampling 8192 CIFAR-10 images or 512 CelebA-HQ or 512 LSUN-Churches images takes 1.5 GPU-hours. FID evaluation for all the scale values of a single noise distribution takes 0.2 GPU-hours.

### E.4 Dataset and Model Licensing

- **CIFAR-10:** Licensed under the MIT License.
- **CelebA-HQ:** Licensed under CC BY-NC 4.0.
- **LSUN-Churches:** Licensed under CC BY-NC 4.0.
- `google/ddpm-cifar10-32`: Apache License, Version 2.0.
- `google/ddpm-celebahq-256`: Apache License, Version 2.0.
- `google/ddpm-church-256`: Apache License, Version 2.0.

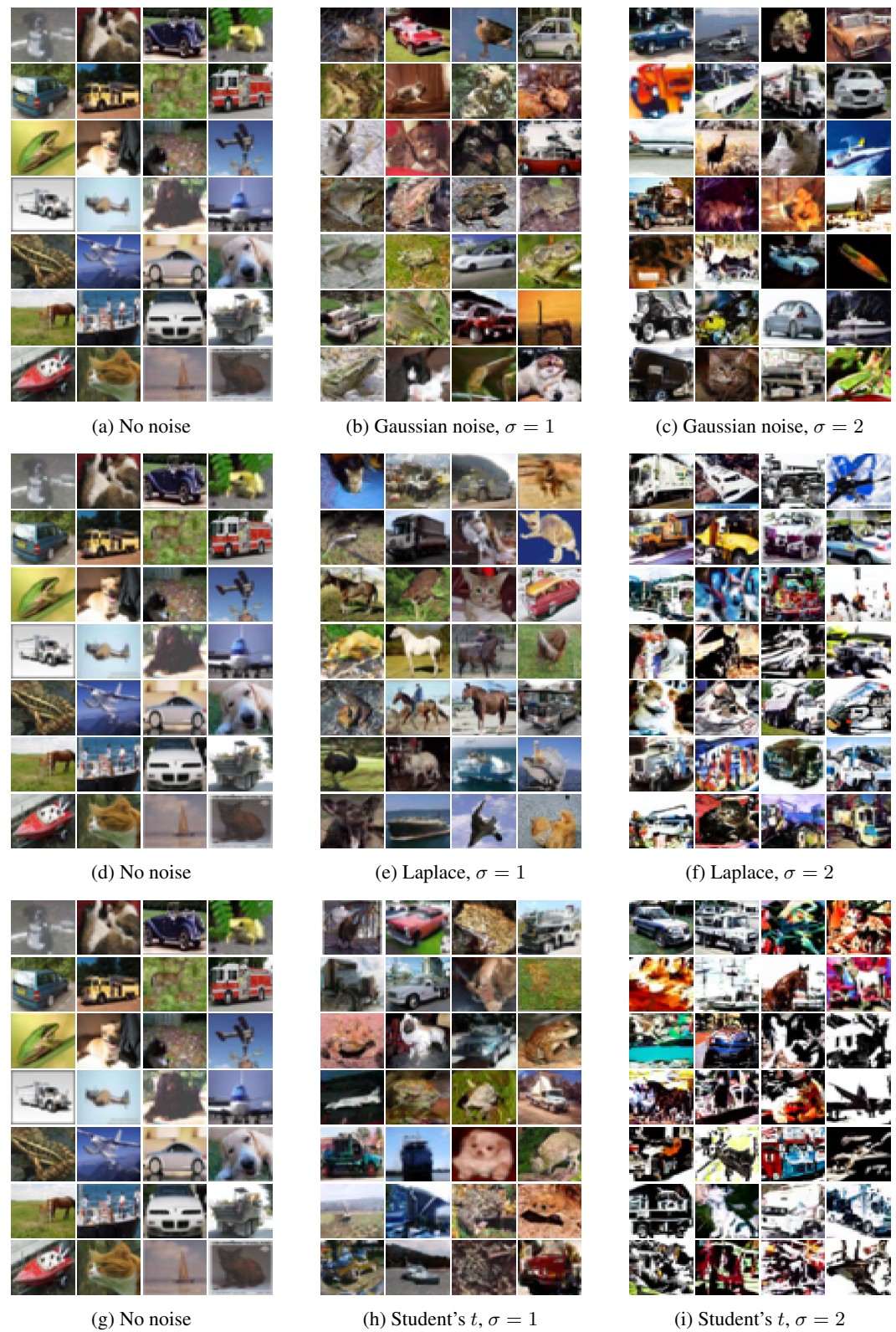

Figure 6: Additional CIFAR-10 generations for 3 noise families (rows) and 2 noise levels (columns).

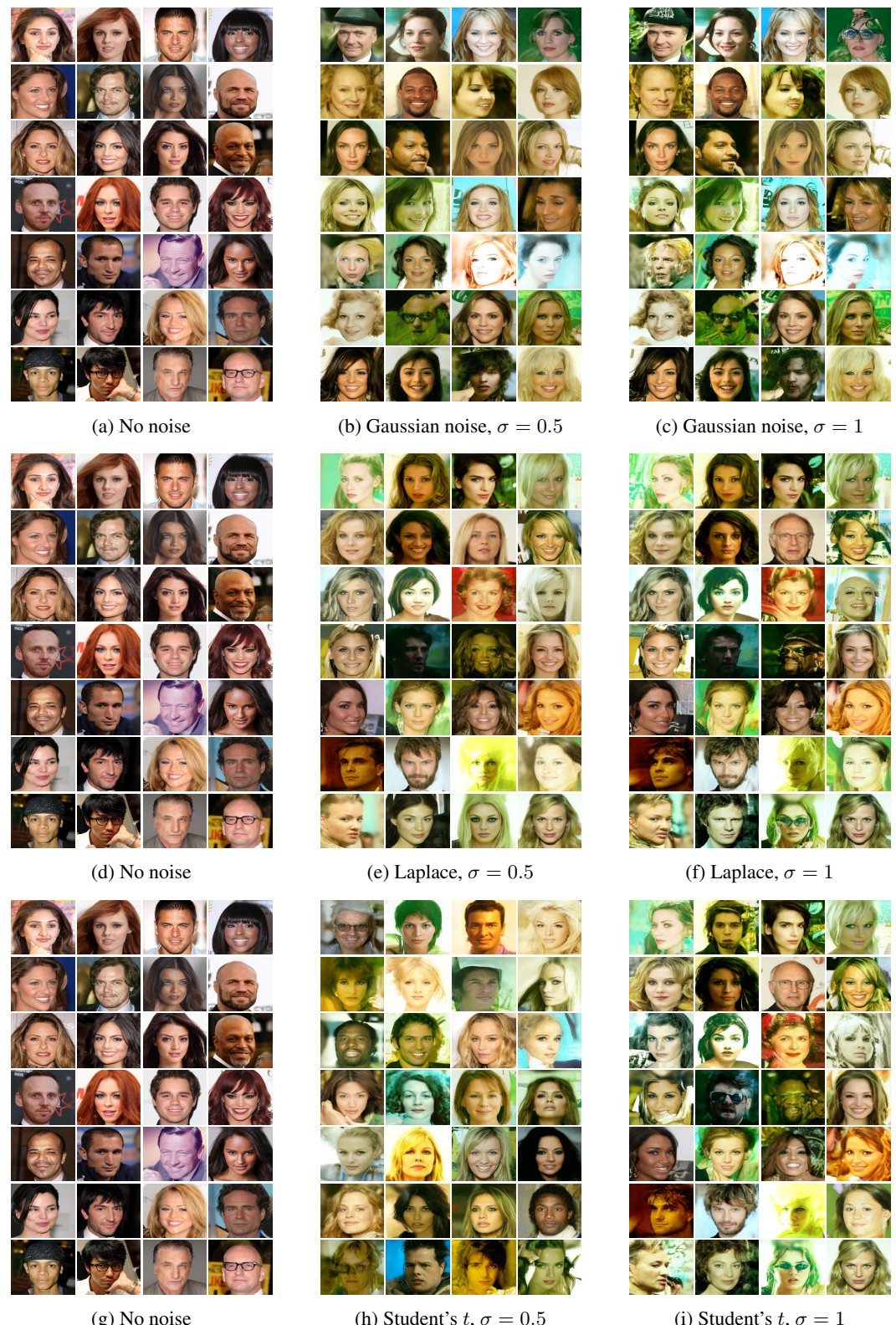

(a) No noise      (b) Gaussian noise, $\sigma = 0.5$      (c) Gaussian noise, $\sigma = 1$

(d) No noise      (e) Laplace, $\sigma = 0.5$      (f) Laplace, $\sigma = 1$

(g) No noise      (h) Student's $t$, $\sigma = 0.5$      (i) Student's $t$, $\sigma = 1$

Figure 7: Additional CelebA-HQ generations for 3 noise families (rows) and 2 noise levels (columns).

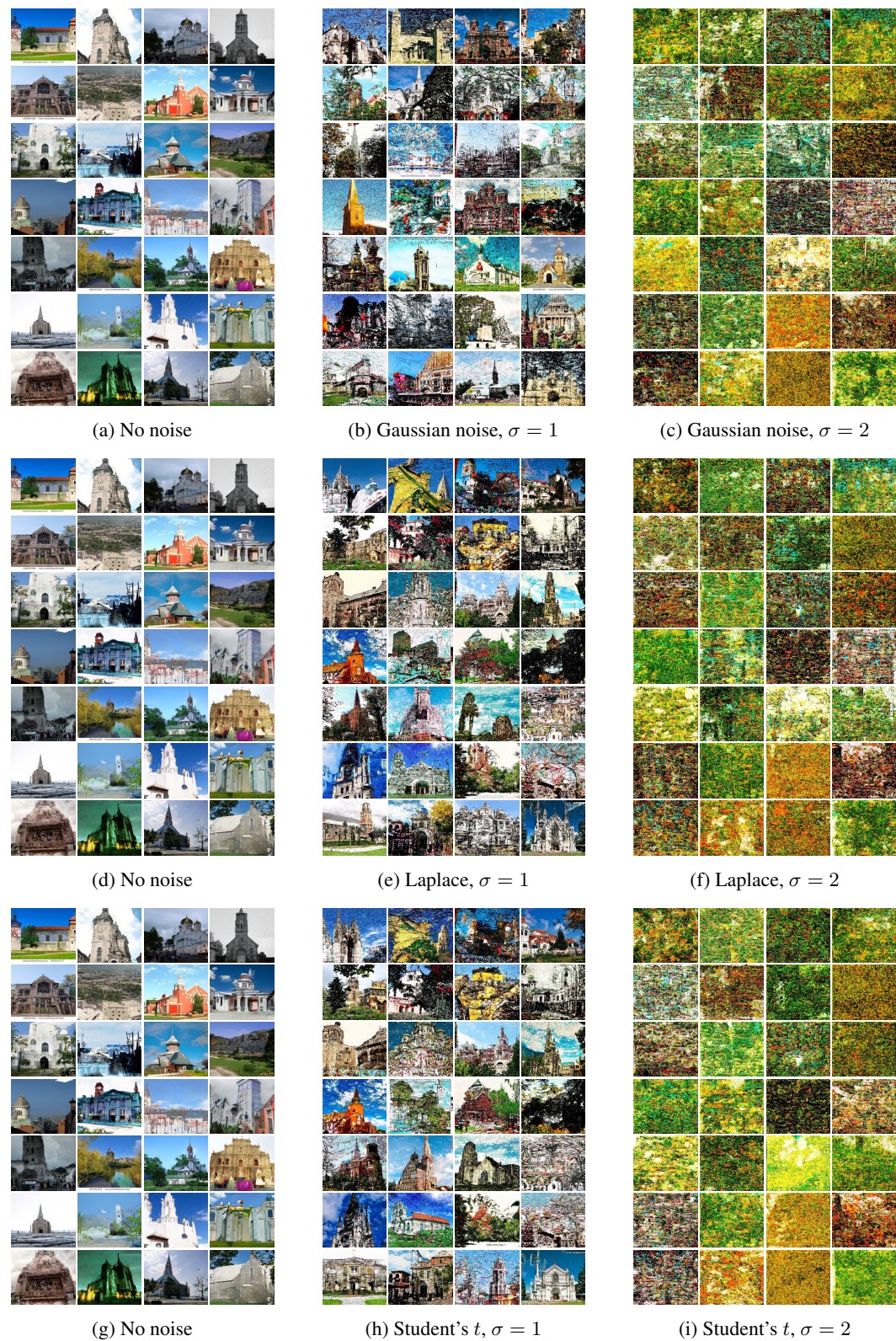

(a) No noise   (b) Gaussian noise, $\sigma = 1$   (c) Gaussian noise, $\sigma = 2$

(d) No noise   (e) Laplace, $\sigma = 1$   (f) Laplace, $\sigma = 2$

(g) No noise   (h) Student's $t$, $\sigma = 1$   (i) Student's $t$, $\sigma = 2$

Figure 8: Additional LSUN-Church generations for 3 noise families (rows) and 2 noise levels (columns).

