# OpenReview forum: "Assessing the quality of denoising diffusion models in Wasserstein distance: noisy score and optimal bounds"
_NeurIPS.cc/2025/Conference — NeurIPS 2025 poster_

### Official Review · Reviewer_KSvS · 2025-06-29

**Clarity:** 3
**Significance:** 3
**Originality:** 3
**Rating:** 4
**Confidence:** 2

**Summary:**

This paper investigate the theoretical and empirical behaviors of Denoising Diffusion Probabilistic Models (DDPMs) along the 2-Wasserstein distance. It is shown empirically by the authors that DDPMs function even if the estimated score function is additive noise corrupted. It comes as a surprise as the score holds a pivotal places in sample generation. The paper provides new non-asymptotic upper bounds on the Wasserstein-2 distance between target and generated distributions. Tightly capture estimation noise (bias and variance) effects on convergence. Establish the optimal rate for for the Gaussian case, but under much milder assumptions, such as semi-log-concavity and bounded support. The achieved rates are shown optimal with respect to the current state of results and include previously unexamined cases (e.g., mixtures of strongly log-concave and compactly supported components). Experiments on CIFAR-10 and CelebA-HQ demonstrate visual robustness of DDPMs to different types of noise, and Fréchet Inception Distance (FID) scores verify theoretical predictions.

**Questions:**

1. How realistic is it to assume a uniform bound on score estimation error for all $x,t$ ? Is the analysis transferable to expected bounds (e.g., across the data distribution)? Removing this assumption would make the results more practically applicable.

2. The bound contains exponential terms like $e^{2bM}$. Can you tell us whether they are inherent to the problem or remnants of your proof techniques? Justifying or motivating these constants would make the theoretical results more comprehensible and tighter.

3. Kinetic Langevin DDPMs is suggested as future work by authors. Do you believe your methods (e.g., error decomposition or variance decay) could be ported directly? Even a brief mention may imply the universality and extensibility of your model.

**Ethical Concerns:**

["NO or VERY MINOR ethics concerns only"]

**Final Justification:**

The authors addressed all my questions.

**Limitations:**

Yes

**Quality:**

3

**Strengths And Weaknesses:**

# Strengths:

The theory is accurate, complete, and commensurate with the state of the art. The paper introduces a novel non-asymptotic analysis of DDPMs in Wasserstein distance, which is more applicable to most real-world problems than KL/TV. In addition, it introduce new analysis framework that neatly separates bias and variance effects on score estimation, and establish convergence to a broader class than the previously studied strongly log-concave or smooth ones The results close an important theoretical gap in the robustness understanding of diffusion models. The boundaries are properly motivated, neatly defined, and claimed to be minimax optimal in some cases. Assumptions and mathematical machinery are well described i.e., key assumptions and contributions are clearly stated. Experimental setup and plots are appropriately described.

# Weaknesses

Some technical assumptions (e.g., uniform error bounds over $x$ and $t$ ) may be difficult to satisfy in real applications. Empirical analysis, although considered, is still small scale and not necessarily representative of failure modes. The results' implications may  limited unless augmented by working practical algorithms that directly make use of the insights (e.g., noise-robust training or compression techniques).  Empirical part are incremental, equivalent visual robustness results are available, but not necessarily rigorously justified.

---

> ### Author Rebuttal · Authors · 2025-07-30
>
> We would like to thank the reviewer for the detailed and thoughtful assessment of our manuscript. It is rewarding to see that the reviewer appreciated the novel analysis framework we introduced, which allowed us to close a theoretical gap left open in previous work on DDPMs. Below, we respond to the weaknesses highlighted in the review and provide answers to the questions raised.
>
> ---
>
> ### Weaknesses
>
> > *Some technical assumptions (e.g., uniform error bounds) may be difficult to satisfy in real applications.*
>
> We agree that weakening this assumption would be desirable. However, we do not believe it is a major limitation in practice. To the best of our knowledge, there are no results showing that standard score estimators (such as those based on score matching or kernel methods) fail in the sup-norm. Furthermore, as explained in the response to Question 1 below, theoretical results from nonparametric statistics support the idea that this condition is not overly restrictive.
>
> > *Empirical analysis, although considered, is still small scale and not necessarily representative of failure modes.*
>
> As correctly noted in the summary of our contributions, our primary focus is on theoretical insights. In addition, as members of an academic laboratory, we have limited access to large-scale computing resources (e.g., GPUs), which makes it difficult to run more extensive empirical evaluations. We hope these two considerations justify the relatively modest empirical part of our submission.
>
> > *The results' implications may be limited unless augmented by working practical algorithms that directly make use of the insights (e.g., noise-robust training or compression techniques).*
>
> Our main goal is to advance theoretical understanding of DDPMs by analyzing their robustness to noise. While the practical significance of this property remains an open question, the increasing interest in compression methods across the AI community confirms the timeliness of our study. For instance, the dedicated Workshop on Machine Learning and Compression at NeurIPS 2024 and the growing number of papers on this topic highlight the relevance of our direction.
>
> > *Empirical part are incremental, equivalent visual robustness results are available, but not necessarily rigorously justified.*
>
> We agree that the main originality of our paper lies in its theoretical contributions rather than its experiments, a point we explicitly acknowledge in the manuscript. However, we are not aware of prior works presenting equivalent robustness results, either visually or quantitatively. If the reviewer has specific references in mind, we would be very grateful to receive them.
>
> ---
>
> ### Questions
>
> 1. **Question** *How realistic is it to assume a uniform bound on score estimation error for all $x$ and $t$? Is the analysis transferable to expected bounds (e.g., across the data distribution)? Removing this assumption would make the results more practically applicable.*
>
>     * First, we emphasize that assuming a uniform bound on the score estimator may not be as restrictive as it seems. In nonparametric statistics, replacing $L_2$-norm error with sup-norm error often leads to only a logarithmic degradation in convergence rates. For example, Theorem 1.8 in [Tsybakov 2008] shows this for local polynomial regression. Similar results exist for density estimation; see [Lepski 2013] and references therein.
>
>     * Second, the expression on line 1076 of the manuscript shows that the relevant quantity in our proofs is actually an $L_2$ error of the score estimator with respect to the distribution of the discretized diffusion process $Z_k$ (see also $\zeta_k$ on line 1063 and $Z_k$ on line 190). This $L_2$ error is upper bounded by the sup-norm error, which is also more convenient for technical derivations. This is the main reason Assumption 2 is formulated in sup-norm terms. Nonetheless, the proof remains valid under a weaker $L_2$-type assumption with respect to the distribution of $Z_k$; the same assumption referred to as H2 in [Silveri & Ocello 2025].
>
> 2. **Question** *The bound contains exponential terms like $e^{2bM}$. Can you tell us whether they are inherent to the problem or remnants of your proof techniques? Justifying or motivating these constants would make the theoretical results more comprehensible and tighter.*
>
>     **Answer** The exponential terms in the bounds depend on the parameters of the target distribution. We conjecture that this behavior is intrinsic to bounding the Wasserstein distance in DDPMs. This belief is reinforced by the independent work [Silveri & Ocello 2025], which uses a different proof technique yet yields similar exponential dependence.
>
>     That said, we do not have a counterexample proving that the exponential factor is unavoidable; i.e., a case where the $W_2$ error must necessarily involve this term. Constructing such an example is challenging for two reasons: (1) for distributions where the DDPM law is known in closed form, our upper bound does not contain the exponential term (as either $b = 0$ or $M = 0$); (2) the $W_2$ distance is defined via an infimum over all couplings, while our upper bound is based on synchronous coupling. Demonstrating the necessity of the exponential term under synchronous coupling would not automatically imply its presence in the actual $W_2$ error.
>
> 3. **Question** *Kinetic Langevin DDPMs is suggested as future work by authors. Do you believe your methods (e.g., error decomposition or variance decay) could be ported directly? Even a brief mention may imply the universality and extensibility of your model.*
>
>     **Answer** While there are similarities between the analysis of overdamped (Langevin) and underdamped (kinetic Langevin) algorithms—as explored in [Chen et al. 2023; Yu et al. 2024; Conforti et al. 2025]; a direct transfer of methods is generally not feasible. A careful examination of these papers shows that the two settings require distinct computations, and those for the kinetic process are much more tedious. For this reason, we have chosen to focus exclusively on the overdamped (Ornstein-Uhlenbeck) setting in the present work, leaving the kinetic case for future investigations.
>
>  ---
>
> ### Cited References
>
> [Tsybakov 2008] Alexandre B. Tsybakov (2008). *Introduction to Nonparametric Estimation*. Springer.
>
> [Lepski 2013] Lepski, Oleg. “Multivariate Density Estimation under Sup-Norm Loss: Oracle Approach, Adaptation and Independence Structure.” *Annals of Statistics* 41(2): 1005–1034, 2013.
>
> [Silveri & Ocello 2025] Marta Silveri and Antonio Ocello. “Beyond Log-Concavity and Score Regularity: Improved Convergence Bounds for Score-Based Generative Models in $W_2$ Distance.” *Proceedings of ICML 2025*.
>
> [Chen et al. 2023] Sitan Chen, Sinho Chewi, Jerry Li, Yuanzhi Li, Adil Salim, Anru Zhang. “Sampling is as Easy as Learning the Score: Theory for Diffusion Models with Minimal Data Assumptions.” *ICLR 2023*.
>
> [Yu et al. 2024] Lu Yu, Avetik Karagulyan, Arnak Dalalyan. “Langevin Monte Carlo for Strongly Log-Concave Distributions: Randomized Midpoint Revisited.” *ICLR 2024*.
>
> [Conforti et al. 2025] Giovanni Conforti, Alain Durmus, Marta Silveri. “KL Convergence Guarantees for Score Diffusion Models under Minimal Data Assumptions.” *SIAM J. Math. Data Sci.*, 7(1): 86–109, 2025.

---

> > ### Comment · Reviewer_KSvS · 2025-08-02
> >
> > I would like to thank the authors for the response. I will keep my positive score.

---

> > > ### Author Response · Authors · 2025-08-05
> > >
> > > We thank the reviewer for reaffirming their positive assessment of our work. However, given the current scores, there is a risk that our manuscript may not be accepted to NeurIPS. Should that be the case, we plan to revise the paper and submit it to another venue. Since we believe the results are of genuine interest to the machine learning community (offering significant improvements over recent work published in top-tier conferences) we aim to strengthen the manuscript to maximize its chances of acceptance elsewhere.
> > >
> > > We made every effort to address the concerns raised in the initial review, and the reviewer’s response suggests that our clarifications were satisfactory. In particular, we explained that the assumption of a uniform bound on the score estimation error is not particularly restrictive, and that the exponential terms appearing in our bounds are likely intrinsic to the problem. Similar terms appear in other recent papers on related topics published in leading venues.
> > >
> > > With this in mind, we would be sincerely grateful if the reviewer could specify which aspects of the paper still fall short of warranting a higher score. Any further feedback would be extremely helpful as we revise the manuscript.
> > >
> > > In particular, the reviewer mentioned that a larger-scale experiment would strengthen the work, and that some prior papers contain "equivalent visual robustness results." We would very much appreciate it if the reviewer could elaborate on these two points. Specifically:
> > >
> > > * If the experiment on the CelebA dataset is considered small in scale, could the reviewer suggest what kind of experiment would be more appropriate or convincing for a NeurIPS-level submission?
> > >
> > > * If there are prior works presenting visual robustness to noise comparable to ours, we would be grateful for references to those papers, so that we can appropriately cite and position our contributions in relation to them.
> > >
> > > We appreciate the opportunity provided by the NeurIPS review process for this dialogue and thank the reviewer in advance for any additional guidance they are willing to provide.

---

### Official Review · Reviewer_hpXx · 2025-07-01

**Clarity:** 3
**Significance:** 3
**Originality:** 3
**Rating:** 5
**Confidence:** 2

**Summary:**

This paper studies the theoretical and empirical behavior of Denoising Diffusion Probabilistic Models (DDPMs) when the estimated score function is perturbed with noise. Specifically, the authors analyze the quality of generated samples under noisy score evaluations using the 2-Wasserstein distance as a metric of generative quality—an alternative to the more commonly used total variation or KL divergence.

On the theoretical side, the authors:

* Formalize a noisy DDPM setting where each score query returns a randomized estimate with bounded bias and variance (Assumption 2).
* Introduce conditions on the target distribution (Assumption 1), capturing both strongly log-concave distributions and semi-log-concave distributions with bounded support.
* Establish non-asymptotic, finite-sample bounds on the 2-Wasserstein distance between the target and generated distributions under noisy score estimates.
* Show that their convergence bounds match the optimal rate of $\sqrt{D}/\epsilon$ (in $D$ dimensions), previously known only for the Gaussian case.

On the experimental side, they validate their theory using pretrained DDPM models on CIFAR-10 and CelebA-HQ datasets. They inject different types of zero-mean noise (Gaussian, Laplace, Uniform, and Student-t) into the score function at test time and:

* Show visually indistinguishable generations even under substantial noise (up to σ = 1 or 2), demonstrating robustness.
* Report FID scores that increase only mildly with noise scale, confirming the theoretical insight that variance in score estimation has limited impact on sample quality.
* Observe that the shape of the noise distribution (e.g., Gaussian vs Laplace) has negligible influence, reinforcing the focus on noise scale rather than distributional assumptions.

**Questions:**

* In Algorithm 2, you use a hybrid arithmetic-geometric time grid. Could you clarify the rationale behind this specific choice? In particular, is there a theoretical justification for its optimality over alternative schemes such as purely geometric or uniform grids, and how would these alternatives affect the convergence guarantees?

* Many real datasets (especially images) lie on low-dimensional manifolds. How sensitive is your analysis to the intrinsic vs. ambient dimensionality of the data?

* In your experiments, did you consider adding noise with a non-zero mean (i.e., introducing bias) to the score function? If not, do you expect the empirical degradation in sample quality to be significantly more severe compared to zero-mean noise?

**Ethical Concerns:**

["NO or VERY MINOR ethics concerns only"]

**Final Justification:**

After carefully reviewing the authors' rebuttal, I believe they have adequately addressed all of my concerns. In particular, the authors have answered my questions about the rationale behind the use of a hybrid arithmetic-geometric time grid, the sensitivity to the intrinsic vs. ambient dimensionality of the data, and an intuition on the expected output when adding noise with a non-zero mean to the scores.

**Limitations:**

Yes.

**Paper Formatting Concerns:**

None.

**Quality:**

3

**Strengths And Weaknesses:**

**Strengths**
* The paper proves that DDPMs are not only robust to score noise (under some assumptions on the target distribution and the noise distribution) but also achieve near-optimal convergence rates in 2-Wasserstein distance.
* The theory developed in this paper is supported by experiments

**Weaknesses**
* While broader than some prior works, the main theoretical results (e.g., Theorems 2 and 3) still rely on strong assumptions like bounded support or semi-log-concavity.

---

> ### Author Rebuttal · Authors · 2025-07-29
>
> We would like to thank the reviewer for the positive evaluation and the fair summary of our contributions, both theoretical and experimental. We particularly appreciate that the strengths and limitations mentioned in the review are fully aligned with those we have transparently discussed in the manuscript. Below, we briefly comment on the limitations and respond point-by-point to the questions raised in the review.
>
> ---
>
> ### **Weakness**
>
> >*While broader than some prior works, the main theoretical results (e.g., Theorems 2 and 3) still rely on strong assumptions like bounded support or semi-log-concavity.*
>
> We agree that it would be very interesting to relax the assumptions we make, while retaining the fast convergence rates we obtain. Nevertheless, we would like to emphasize that our assumptions are *weaker* than those used in previous works that yield only suboptimal rates; see, for instance, [Silveri & Ocello 2025; Yu & Yu 2025]. In particular, these works require the target distribution to be absolutely continuous with respect to the Lebesgue measure, whereas our results apply to certain distributions that do not satisfy this assumption. It is worth noting that the widely cited work [Chen et al. 2023] requires the target distribution to have a Lipschitz-continuous score function (Assumption A1). This assumption is stronger than the semi-log-concavity, and therefore, more restrictive than the assumptions of our work, but is still considered as not restrictive since the title of the mentioned paper refers to it as “minimal data assumptions”.
>
> Furthermore, compared to prior work in the sampling literature, assuming semi-log-concavity or bounded support does not appear overly stringent. Many commonly used distributions satisfy these assumptions.
>
> Finally, as discussed in Section A (page 21), the existing results that rely on a different set of assumptions, which may arguably be considered weaker (though not strictly comparable), lead to worse dependence not only on the discretization step $h$, but also, more problematically, on the error of the estimated score. This suggests that even when using an optimal score estimator, those approaches may not achieve optimal sampling rates (in the $W_2$-distance), unlike ours.
>
> ---
>
> ### **Questions**
>
> 1. **Question** *In Algorithm 2, you use a hybrid arithmetic-geometric time grid. Could you clarify the rationale behind this specific choice? In particular, is there a theoretical justification for its optimality over alternative schemes such as purely geometric or uniform grids, and how would these alternatives affect the convergence guarantees?*
>
>     **Answer** In the second part of the time interval shown in Figure 3 (page 30), the discretization error includes a term of the form $\sum_k \frac{h_k}{T - t_{k+1}}$, as noted in lines 1141–1143. Here, $h_k$ denotes the step size, and $t_k$ the time grid points. Using a uniform grid would cause the last terms in this sum to become large, on the order of a constant, thus preventing the error from vanishing as the number of steps increases.
>
>     Conversely, applying a geometric grid throughout results in a large discretization error during the first part of the interval. According to the equation following line 1134 and Eq. (30), this error scales roughly as $\sum_k h_k^2$. Given the constraint $\sum_k h_k = C$, the sequence $h_k = C/K$ minimizes the sum of squared step sizes. Therefore, a uniform grid is optimal over the first part. Combining this with a geometric grid on the second part, where uniform discretization would fail, yields our hybrid scheme, which is both theoretically justified and empirically effective.
>
> ---
>
> 2. **Question** *Many real datasets (especially images) lie on low-dimensional manifolds. How sensitive is your analysis to the intrinsic vs. ambient dimensionality of the data?*
>
>     **Answer** The DDPM algorithm adds isotropic noise to the data without accounting for the underlying manifold structure. In other words, the injected noise is equally strong in all directions of the ambient space. Therefore, it is natural, and theoretically expected, that the resulting error scales with the square root of the ambient dimension. Reducing this dependence to the intrinsic dimension would require a careful exploitation of the low-dimensional structure in the score estimation step, not in the DDPM itself. As in many nonparametric estimation problems, the error of the score estimator (denoted $\varepsilon$ in our paper) is sensitive to the intrinsic dimension. Hence, the choice of score estimator is a critical component of the DDPM pipeline, even though it is decoupled from the convergence analysis we conduct in this work.
>
> ---
>
> 3. **Question** *In your experiments, did you consider adding noise with a non-zero mean (i.e., introducing bias) to the score function? If not, do you expect the empirical degradation in sample quality to be significantly more severe compared to zero-mean noise?*
>
>     **Answer**  We did not consider noise with a non-zero mean in our experiments. However, we expect this type of noise to have a much more detrimental impact on the sampling performance. When the noise mean is non-zero, the iterates of the algorithm are no longer decorrelated from this bias. As a result, the mean bias behaves analogously to the term $\varepsilon^b$, which negatively affects convergence guarantees. While we cannot include empirical results due to rebuttal format constraints, we would be happy to discuss this further during the discussion period.
>
> ---
>
> ### Cited References
>
>  [Silveri & Ocello 2025] Silveri, M. and Ocello, A. “Beyond Log-Concavity and Score Regularity: Improved Convergence Bounds for Score-Based Generative Models in W2-Distance.” *Proceedings of ICML 2025*.
>
>  [Yu & Yu 2025] Yifeng Yu and Lu Yu. *Advancing Wasserstein Convergence Analysis of Score-Based Models: Insights from Discretization and Second-Order Acceleration*. *arXiv preprint*, 2025.

---

> > ### Comment · Reviewer_hpXx · 2025-08-01
> >
> > I thank the authors for their complete and thoughtful response. All my concerns and questions have been resolved, and I have increased my score.

---

### Official Review · Reviewer_nFJ3 · 2025-07-01

**Clarity:** 3
**Significance:** 3
**Originality:** 3
**Rating:** 4
**Confidence:** 1

**Summary:**

This paper investigates the robustness and convergence guarantees of denoising diffusion probabilistic models (DDPMs) under noisy score evaluations. The authors empirically demonstrate that DDPMs remain effective when additive noise with constant variance is introduced into the score function during sampling. Theoretically, the paper derives non-asymptotic upper bounds on the Wasserstein distance between the true data distribution and the distribution induced by the noisy DDPM. These bounds match the optimal rate known in the Gaussian case and extend to a broad class of distributions, including compactly supported semi-log-concave measures. The analysis highlights the limited impact of noise variance compared to bias and establishes conditions under which fast convergence is achieved. The results are compared to numerical experiments on benchmark datasets.

**Questions:**

**1.** Do the authors consider it feasible and useful to extend this type of analysis to high-dimensional settings?

**2.** In the conclusion, the authors mention the possibility of incorporating estimators of the Hessian of the log-density into the DDPM framework. Could they elaborate on what they mean by this, and how such estimators would be used in practice?

**3.** Perhaps I missed it, but are there any assumptions on the score estimator $\hat{\mathbf{s}}$? For instance, can it be a neural network with arbitrary activation functions, or are there regularity or smoothness requirements needed for the theoretical results to hold?

**Ethical Concerns:**

["NO or VERY MINOR ethics concerns only"]

**Final Justification:**

I thank the authors for their careful and detailed rebuttal addressing all questions and remarks. I will maintain my positive score, though I stress that I am not familiar with the literature or the techniques used in the paper.

**Limitations:**

Yes.

**Paper Formatting Concerns:**

The paper follows the NeurIPS 2025 Paper Formatting Instructions

**Quality:**

3

**Strengths And Weaknesses:**

**Quality**

The paper presents a non-asymptotic analysis of DDPMs under noisy score evaluations, with claims of improved convergence rates in Wasserstein-2 distance. The technical arguments appear carefully constructed, and the analysis is detailed. However, I am not familiar with the details of the extensive literature on the bounds mentioned and cited in the paper.


**Clarity**

The exposition is generally well structured and the motivation is clear. I particularly appreciate the sections where the authors discuss the relevance and the challenges of the Wasserstein distance. Assumption 1, which seems very technical, is extensively discussed in the appendix.


**Significance**

The paper addresses a relevant question, particularly in settings where score evaluations may be noisy due to communication or compression constraints. If correct, the claimed convergence rates would represent a theoretical advance, given the references cited by the paper.


**Originality**

I believe it's an original paper. The focus on robustness to noisy scores in the context of DDPMs, and the use of Wasserstein distance as the evaluation metric, marks a departure from much of the existing literature, which concentrates on KL or total variation distances. The consideration of different forms of randomized score estimators is also less common.

---

> ### Author Rebuttal · Authors · 2025-07-29
>
> We thank the reviewer for their positive evaluation of our contributions. We are especially pleased that they appreciated the discussion on the relevance of the Wasserstein distance in a theoretical setting. We considered it important to include this discussion, given the number of questions we received on this point when presenting our results in various seminars. Below, we provide point-by-point responses to the reviewer’s questions.
>
> ---
>
> 1. **Question**: *Do the authors consider it feasible and useful to extend this type of analysis to high-dimensional settings?*
>    **Answer**: The setting we study in the paper is, in fact, commonly referred to as high-dimensional. Our results exhibit a square-root dependence of the Wasserstein error on the dimension, which appears to be optimal under the assumptions we consider, or even under assumptions frequently adopted in the existing literature. To develop some intuition for why this $\sqrt{D}$ scaling is unavoidable, consider the case where the target distribution is a product of identical univariate distributions. Given that the $W_2$ distance is based on the Euclidean norm, it naturally scales as $\sqrt{D}$ times the error incurred when sampling from a one-dimensional distribution. Extending our results to infinite-dimensional sampling problems is an intriguing direction for future research.
>
> ---
>
> 2. **Question**: *In the conclusion, the authors mention the possibility of incorporating estimators of the Hessian of the log-density into the DDPM framework. Could they elaborate on what they mean by this, and how such estimators would be used in practice?*
>    **Answer**: The use of the Hessian of the log-density is a well-established strategy in the sampling literature for improving convergence. For example, the Ozaki discretization [Ozaki 1992; Stramer & Tweedie 1999] leverages the Hessian to enhance the convergence properties of Langevin-based algorithms. Similarly, the KLMC2 algorithm introduced in [Dalalyan & Riou-Durand 2020] achieves fast rates for discretized kinetic Langevin diffusion, a method that Stability AI has successfully implemented.
>
>    Our remark on incorporating estimated Hessians follows this tradition. More precisely, one could use the Itô decomposition presented just above line 1202 to define a second-order discretization of the continuous-time diffusion process $Y_t$. This would enable the algorithm to more accurately capture the local curvature of the log-density and, potentially, enhance its efficiency. We hope this explanation clarifies our intention, and we would be glad to provide further details during the discussion phase if the reviewer finds it helpful.
>
> ---
>
> 3. **Question**: *Perhaps I missed it, but are there any assumptions on the score estimator? For instance, can it be a neural network with arbitrary activation functions, or are there regularity or smoothness requirements needed for the theoretical results to hold?*
>    **Answer**: The only conditions imposed on the estimated score are those stated in Assumption 2. It is entirely possible to use a neural network with non-smooth activation functions, such as ReLU or Heaviside, even though they yield non-differentiable or discontinuous score estimates. This flexibility stems from the structure of our analysis, which decouples the error in score estimation from the discrepancy between the discrete-time process using the true score and its continuous-time counterpart. As a result, no additional smoothness or regularity assumptions are needed for the estimated score.
>
> ---
>
> ### Cited References
>
> [Ozaki 1992] T. Ozaki. *A bridge between nonlinear time series models and nonlinear stochastic dynamical systems: a local linearization approach*. Statistica Sinica, 2(1):113–135, 1992.
>
> [Stramer & Tweedie 1999] O. Stramer and R. L. Tweedie. *Langevin-type models. II. Self-targeting candidates for MCMC algorithms*. Methodology and Computing in Applied Probability, 1(3):307–328, 1999.
>
> [Dalalyan & Riou-Durand 2020] A. Dalalyan and L. Riou-Durand. *On sampling from a log-concave density using kinetic Langevin diffusions*. Bernoulli, 26(3):1956–1988, 2020.

---

> > ### Comment · Reviewer_nFJ3 · 2025-08-04
> >
> > I thank the authors for their careful and detailed rebuttal addressing all questions and remarks. I will maintain my positive score, though I stress that I am not familiar with the literature or the techniques used in the paper.

---

> > > ### Author Response · Authors · 2025-08-05
> > >
> > > We thank the reviewer for reaffirming their positive assessment of our work. However, given the current scores, there is a risk that our manuscript may not be accepted to NeurIPS. Should that happen, we intend to revise the paper and submit it to another venue. Since we believe the results are of interest to the machine learning community (providing a significant improvement over recent work published in top-tier venues) we aim to strengthen the manuscript to maximize its chances of acceptance elsewhere.
> > >
> > > We did our best to address the concerns raised in the initial review, and the reviewer’s response suggests that our clarifications were found satisfactory. In particular, we emphasized that our results apply to the high-dimensional setting, where a linear dependence on the dimension is unavoidable, and that we impose no smoothness or structural assumptions on the score estimate.
> > >
> > > With this in mind, we would be sincerely grateful if the reviewer could indicate which aspects of the paper still fall short of justifying a higher score. Any additional feedback would be extremely valuable as we work on revising the manuscript.

---

### Official Review · Reviewer_YmpH · 2025-07-03

**Clarity:** 3
**Significance:** 2
**Originality:** 3
**Rating:** 4
**Confidence:** 2

**Summary:**

The paper studies how robust DDPM sampling is when the score function is noisy, using Wasserstein distance to measure the distance from the true distribution. Experiments on CIFAR-10 and CelebA-HQ show that images remain visually unchanged until the added noise becomes quite large. Motivated by this, the authors prove that under some assumptions on the score estimates and the data distribution, the Wasserstein-2 distance between the generated and true distributions decreases at a near-optimal rate of $\sqrt{D} / \epsilon$, matching the known lower bound in the Gaussian case and improving over the previous best $D / \epsilon^2$ rate.

**Questions:**

1. How critical is Assumption 1? Are there reasonably well-behaved distributions that fail to satisfy it, for which a $\sqrt{D} / \epsilon$ bound provably doesn't hold?
2. The paper says the uniformity in Assumption 2 is essential to the proof. Is this just a technical condition, or is there a concrete example where dropping it would break the $\sqrt{D} / \epsilon$ bound?

**Ethical Concerns:**

["NO or VERY MINOR ethics concerns only"]

**Final Justification:**

The authors addressed my questions in the rebuttal. I think the techniques introduced in this paper used would offer value to the NeurIPS community. I would like to increase my recommendation score to 4.

**Limitations:**

yes

**Quality:**

2

**Strengths And Weaknesses:**

Strengths:

1. The paper is fairly well written. The theory is clearly presented, with helpful examples.
2. The main result is rigorously proved, and the bounds are non-asymptotic.

Weaknesses:

I'm not fully convinced by the motivation. Assumption 2 requires a uniform error bound over all x, which feels quite strong. The examples of randomized score estimators aren't very compelling — it's hard to think of real-world cases where we truly need a compressed score network or where scores suffer from communication noise.

---

> ### Author Rebuttal · Authors · 2025-07-29
>
> We thank the reviewer for their positive comments on the clarity of the writing and the rigour of the mathematical derivations. We devoted considerable effort to these aspects and are pleased that they have been acknowledged. Below, we address the questions and concerns raised in the review.
>
> ---
>
> ### Weaknesses
>
> * **Uniform error in Assumption 2**:
>     * First, we would like to emphasize that requiring a uniform bound on the score estimator is not as restrictive as it might initially seem. In many nonparametric estimation problems, using the sup-norm (i.e., uniform error) instead of the $L_2$-norm (i.e., integrated error) leads to only a logarithmic loss in the convergence rate. For example, Theorem 1.8 in [Tsybakov 2008] demonstrates this for the local polynomial estimator of a regression function. Similar results are available in multivariate density estimation; see [Lepski 2013] and references therein.
>
>     * Second, the formula on line 1076 of the manuscript shows that the quantity appearing in our proof is actually an $L_2$ error of the score estimator with respect to the distribution of the discretized diffusion process $Z_k$ (see also the definitions of $\zeta_k$ on line 1063 and $Z_k$ on line 190). This $L_2$ error is clearly upper bounded by the sup-norm error, which is also more convenient for technical derivations. This is the main reason we stated Assumption 2 in terms of a uniform error. However, the proof remains valid under a weaker $L_2$-type assumption with respect to the distribution of $Z_k$; ; the same assumption referred to as H2 in [Silveri & Ocello 2025]..
>
> * **Compellingness of the randomized score**:
>     The main objective of our paper is to provide a theoretical analysis of a widely used algorithm, DDPM. From a theoretical perspective, we believe it is valuable to understand the robustness of this method to noise. While the practical impact of this property remains to be fully understood, the growing interest in compression within the AI community strongly supports its relevance. For example, the Workshop on Machine Learning and Compression at NeurIPS 2024, along with a large number of submissions on neural network compression and its effect on predictive performance, highlight the timeliness and importance of this direction.
>
> ---
>
> ### Questions
>
> 1. **Question**: *How critical is Assumption 1? Are there reasonably well-behaved distributions that fail to satisfy it, for which a $\sqrt{D}/\varepsilon$ bound provably doesn't hold?*
>    **Answer**: This is a difficult and subtle question. As discussed in the paper, most standard distributions satisfy Assumption 1 and therefore cannot serve as counterexamples to this assumption. Furthermore, as explained in Section A (page 21), there are essentially three main strategies for proving the convergence of diffusion samplers. Only the approach developed in our work currently yields the optimal rate of $\sqrt{D}/\varepsilon$. Even if one could identify a distribution that fails to meet certain assumptions used in our proof, this would not necessarily imply that the desired bound cannot be achieved using one of the other techniques. Finally, proving a lower bound on the $W_2$ distance between the sampling distribution and the target distribution is a challenging task.
>
> 2. **Question**: *The paper says the uniformity in Assumption 2 is essential to the proof. Is this just a technical condition, or is there a concrete example where dropping it would break the bound?*
>    **Answer**: We acknowledge that this sentence does not accurately reflect the underlying mathematical reasoning, and we will revise it in the paper. For a more complete response, please refer to our answer to the first weakness above.
>
> ---
>
> ### Concluding Remark
>
> We would like to emphasize that even if one disregards our contribution concerning the robustness of the score estimator to noise, we believe that establishing the first mathematical proof of the optimality of DDPM for a reasonably broad class of distributions is in itself a significant result. The review identifies the restrictiveness of our assumptions as a key weakness. Still, we note that most papers on DDPM, such as the recent work by [Silveri & Ocello, 2025], employ more restrictive assumptions while only achieving suboptimal rates.
>
> ---
>
> ### Cited References
>
> [Tsybakov 2008] Alexandre B. Tsybakov (2008). *Introduction to Nonparametric Estimation*. Springer Publishing Company.
>
> [Lepski 2013] Lepski, Oleg. “Multivariate Density Estimation under Sup-Norm Loss: Oracle Approach, Adaptation and Independence Structure.” *The Annals of Statistics* 41, no. 2 (2013): 1005–1034.
>
> [Silveri & Ocello 2025] Silveri, Marta and Antonio Ocello. “Beyond Log-Concavity and Score Regularity: Improved Convergence Bounds for Score-Based Generative Models in W2-Distance.” *Proceedings of ICML 2025*.

---

> > ### Comment · Reviewer_YmpH · 2025-08-06
> >
> > I would like to thank the authors for the detailed response. While I remain somewhat unconvinced about the setting of the randomized score, I think the techniques introduced in this paper used would offer value to the NeurIPS community. I will raise my score.

---

### Decision · Program_Chairs · 2025-09-17

**Decision:**

Accept (poster)

**Comment:**

This paper analyses robustness of DDPM in terms of Wasserstein distance. The authors showed that under the assumption of log-concavity and semi-log-concavity, DDPM is robust against noise injection to the estimated model and thus achieves the near optimal rate $\sqrt{D}\epsilon$.

This paper gives an interesting and solid theoretical justification of the robustness of DDPM. Under the log-concavity condition of the objective, the injected noise does not expand exponentially against the number of steps. This is somehow predictable. However, this paper successfully gives better rate than existing work, which is novel.

The numerical experiments effectively show plausibility of the insight obtained by the theoretical analysis.

Overall, this paper is well written giving a solid theoretical contribution. Hence, I recommend acceptance.